# Mirror Descent Policy Optimisation for Robust Constrained Markov Decision Processes

**David M. Bossens**                                    *david_bossens@a-star.edu.sg*
*Centre for Frontier AI Research, Agency for Science, Technology and Research (A*STAR), Singapore*
*Institute of High Performance Computing, Agency for Science, Technology and Research (A*STAR), Singapore*

**Atsushi Nitanda**                                    *atsushi_nitanda@a-star.edu.sg*
*Centre for Frontier AI Research, Agency for Science, Technology and Research (A*STAR), Singapore*
*Institute of High Performance Computing, Agency for Science, Technology and Research (A*STAR), Singapore*
*College of Computing and Data Science, Nanyang Technological University, Singapore*

**Reviewed on OpenReview:** *https://openreview.net/forum?id=tmfdqtFUqO*

## Abstract

Safety is an essential requirement for reinforcement learning systems. The newly emerging framework of robust constrained Markov decision processes allows learning policies that satisfy long-term constraints while providing guarantees under epistemic uncertainty. This paper presents mirror descent policy optimisation for robust constrained Markov decision processes, making use of policy gradient techniques to optimise both the policy (as a maximiser) and the transition kernel (as an adversarial minimiser) on the Lagrangian representing a constrained Markov decision process. Our proposed algorithm obtains an $\tilde{\mathcal{O}}\left(1/T^{1/3}\right)$ convergence rate in the sample-based robust constrained Markov decision process setting. The paper also contributes an algorithm for approximate gradient descent in the space of transition kernels, which is of independent interest for designing adversarial environments in general Markov decision processes. Experiments confirm the benefits of mirror descent policy optimisation in constrained and unconstrained optimisation, and significant improvements are observed in robustness tests when compared to baseline policy optimisation algorithms.

## 1 Introduction

Reinforcement learning (RL) traditionally forms policies that maximise the total reward, within the Markov decision process (MDP) framework. Two important aspects, often overlooked by RL algorithms, are the safety of the policy, in terms of satisfying behavioral constraints, or the robustness of the policy to environment mismatches. In continuous control applications, for instance, there are often constraints on the spaces the agent may go and on the objects it may not touch, and there is the presence of the simulation-reality gap.

Safe RL techniques have been primarily proposed within the constrained Markov decision process (CMDP) formalism (Altman, 1999). In infinite CMDPs, which are characterised by large or continuous state and action spaces, traditional solution techniques include linear programming and dynamic programming with Lagrangian relaxation. Following their empirical performance in high-dimensional control tasks (Lillicrap et al., 2015; Mnih et al., 2016), policy gradient algorithms have become a particularly popular option for safe RL in control applications (Achiam et al., 2017; Yang et al., 2020; Ding et al., 2020; Liu et al., 2021b; Xu et al., 2021; Paternain et al., 2023; Ying et al., 2023; Wu et al., 2024).

The robustness to mismatches in training and test transition dynamics has been the main subject of interest in the robust MDP (RMDP) framework (Iyengar, 2005; Nilim & Ghaoui, 2005), which formulates robust policies by training them on the worst-case dynamics model within a plausible set called the uncertainty set. The framework has led to a variety of theoretically sound policy gradient approaches (Ho et al., 2021; Zhang et al., 2021; Kumar et al., 2023; Kuang et al., 2022; Wang et al., 2024; Li et al., 2023; Zhou et al., 2023; Wang et al., 2023). Recently, the approach has also branched out into the robust constrained MDP (RCMDP) framework (Russel et al., 2020; 2023; Wang et al., 2022; Bossens, 2024; Zhang et al., 2024; Sun et al., 2024), which seeks to provide robustness guarantees to CMDPs.

The framework of RCMDPs is newly emerging and hence the theoretical development lags behind compared to CMDPs and RMDPs. At present, a few challenges have hampered the progress of RCMDP policy optimisation. First, since the framework is based on a worst-case cost function and one or more additional worst-case constraint-cost functions, it is challenging to implement in a single simulator since each of the robust cost functions may turn out to be based on a different transition kernel. This has lead to practical objectives that do not meet the original objective (Russel et al., 2020) or which only work for a discrete set of transition kernels (Mankowitz et al., 2020). Second, the state-of-the-art CMDP techniques for policy optimisation and RMDP techniques for adversarially optimising the transition dynamics have not yet widely made their way into the RCMDP literature, which has lead to a lack of theoretical guarantees. In RMDPs, state-of-the-art results are obtained for mirror descent based policy gradient algorithm (Wang et al., 2024), which applies mirror descent in the policy space and mirror ascent in the transition kernel space. In CMDPs, the policy mirror descent primal-dual (PMD-PD) algorithm is a related mirror descent based policy gradient algorithm which provides $\mathcal{O}(\log(T)/T)$ convergence rate is obtained in the oracle-based setting and an $\tilde{\mathcal{O}}(1/T^{1/3})$ convergence rate in the sample-based setting (Liu et al., 2021b). We then pose the question: using mirror descent based policy optimisation, is it intrinsically more difficult to optimise RCMDPs, or is it possible to obtain similar convergence rates?

With the aim of solving these challenges and providing an affirmative answer to the above question, we draw inspiration from the robust Lagrangian objective (Bossens, 2024) and the success of mirror descent optimisation in RMDPs and CMDPs (Wang et al., 2024; Liu et al., 2021b). We show that it is indeed possible to obtain convergence rates similar to Sample-based PMD-PD in the sample-based RCMDP setting. For a theoretical analysis, we contribute Robust Sample-based PMD-PD, which extends the theory of Liu et al. (2021b) and Wang et al. (2024) to the RCMDP setting, while focusing on the softmax parametrisation within the sample-based setting. Doing so, we achieve the $\tilde{\mathcal{O}}(1/T^{1/3})$ convergence rate as also obtained by (non-robust) Sample-based PMD-PD. We then also extend the analysis to non-rectangular RCMDPs, albeit with a weaker sample-based convergence rate. For a practical implementation, we formulate MDPO-Robust-Lagrangian, which introduces a robust Lagrangian to Mirror Descent Policy Optimisation (MDPO) (Tomar et al., 2022). We evaluate MDPO-Robust-Lagrangian empirically by comparing it to robust-constrained variants of PPO-Lagrangian (Ray et al., 2019) and RMCPMD (Wang et al., 2024), and find significant improvements in the penalised return in worst-case and average-case test performance on dynamics in the uncertainty set.

In addition to techniques drawn from Liu et al. (2021b), Wang et al. (2024), and Tomar et al. (2022), our work designs novel techniques for theoretical analysis and contributes tools and results for practical experiments. In theoretical development, our work designs the following techniques:

- **Bounding the error due to sub-optimal dual variables:** the proof of Theorem 4.2 of Wang et al. (2024) provides a result for bounding the robust regret in RMDPs by adding the tolerance of TMA and the regret on the current transition kernel. To extend this reasoning to RCMDPs, we provide an upper bound on the robust Lagrangian regret (Lemma 8) which in addition to these two terms also considers the suboptimality of the dual variables. This additional error is then shown to vanish in an average regret sense due to the constraint-costs averaging to $\mathcal{O}(\epsilon)$ across iterations, where $\epsilon > 0$ is the error tolerance (Lemma 29).

- **Bounding errors due to transition kernel changes:** a challenge in applying the telescoping analysis of Liu et al. (2021b) to RCMDPs is that some of the terms no longer cancel since the transition kernel changes after the policy update. To ensure that this analysis is still possible, Lemma 24 and Lemma 25 bound the error introduced by changes in the transition kernel using different telescoping sums.

- **Approximate TMA:** previously, the TMA algorithm of Wang et al. (2024) relied on oracle information. To extend the algorithm to the sample-based setting, we have designed the Approximate TMA algorithm, which approximates the value function to perform mirror descent on transition kernels, and we derive its sample complexity as $\tilde{\mathcal{O}}(1/\epsilon^2)$.

- **Continuous pseudo-KL divergence of occupancy:** we present a pseudo-KL divergence for continuous state-action spaces and show it is a Bregman divergence, making PMD-PD techniques applicable to continuous state-action spaces.

- **Non-rectangular RCMDPs:** while Wang et al. (2024) provides an analysis of TMA only for $s$-rectangular RMDPs and derives oracle-based convergence rates for the policy optimiser, our work applies the Appproximate TMA algorithm to non-rectangular RCMDPs and then derives a full sample-based analysis of the full algorithm optimising all three parameters $(\pi, p, \lambda)$.

From an experimental perspective, we highlight the following novelties:

- **Comprehensive ablation study:** our study compares MDPO, PPO, and MCPMD algorithms for MDPs, CMDPs, RMDPs, and RCMDPs across three problem domains, including tests on the uncertainty set where the return and different penalised returns are evaluated.

- **Scalable implementation:** the practical algorithm is highly scalable, making use of function approximators and other deep RL tools, as opposed to the pure Monte Carlo considered in Wang et al. (2024).

- **Learning rate schedules:** we also present an empirical comparison of learning rate schedules, including traditional schedules fixed an linear schedules (Tomar et al., 2022), geometric schedules (Xiao, 2022; Wang et al., 2024), and a newly designed restart schedule. We show that for RCMDPs, the best performance is obtained by the MDPO-Robust-Augmented-Lag variant under batch-based restart schedules.

## 2 Related work

### 2.1 Robust constrained objectives

Russel et al. (2020) proposed the robust constrained policy gradient (RCPG) algorithm, which solves the inner problem based on either the robust cost or the robust constraint-cost, which ignores at least one part of the robustness problem in RCMDPs. Solving the RCMDP objective directly, the R3C objective (Mankowitz et al., 2020) combines the worst-case costs and worst-case constraint-costs by running distinct simulators with different parametrisations, each being run for one step from the current state. This unfortunately cannot be realised within a single simulator and is only applicable for discrete parametrisations. Additionally, it is also computationally expensive, even more so when there are multiple constraints. Alternatively, the robust Lagrangian objective trades off the cost and different constraint-cost according to whether the constraints are active while requiring only a single simulator with differentiable parameters (Bossens, 2024).

In this paper, we use the robust Lagrangian objective with multiple constraints and rather than performing a single-step transition kernel update, we perform multi-step updates to the transition kernel at each macro-iteration, in a double-loop manner, to ensure that a near-optimal solution to the inner problem has been found, which in turn allows to provide guarantees on the near-optimality of the policy in solving the robust Lagrangian objective. Assuming that there exists at least one policy that is strictly feasible for all the constraints under all dynamics in the uncertainty set, the policy that optimises the robust Lagrangian will satisfy the worst-case constraint-costs and will, due to complementary slackness, optimise the worst-case value.

### 2.2 Adversarial transition kernels

Techniques for the design of adversarial transition kernels are of general interest for reinforcement learning. They represent different trade-offs between realism, scalability, computational cost, and sample efficiency. In the context of RMDPs and RCMDPs, they are used in the inner problem, which computes the worst-case transition kernel for the current policy. In general, convex uncertainty and rectangular uncertainty sets are required to make the problem tractable, since non-convex sets and non-rectangular set are in general NP-Hard (Wiesemann et al., 2013).

The scalability of these techniques is a key challenge. Linear programming based techniques (Russel et al., 2020) and techniques based on a small subset of simulators (Mankowitz et al., 2020) are not scalable. Instead, gradient-based methods can potentially be performed at scale. In RMDPs, gradient descent techniques for computing the inner problem have been proposed. Unfortunately, many previous techniques do not have known guarantees (Abdullah et al., 2019; Bossens, 2024). Recent work has proposed transition mirror ascent (TMA), which comes with an $\mathcal{O}(1/T)$ convergence rate in the oracle-based setting under convex rectangular uncertainty sets when using fixed learning rates, and $\mathcal{O}((1 - \frac{1-\gamma}{M})^T)$ convergence rates when using geometric schedules for the learning rate (Wang et al., 2024). A similar guarantee was earlier provided with a projected subgradient ascent algorithm, under slightly less general conditions, i.e. when the uncertainty set is a rectangular ball (Kumar et al., 2023).

In RCMDPs, the Adversarial RCPG algorithm (Bossens, 2024) provides a neural representation for transition kernels. Unfortunately, the algorithm does not have known theoretical guarantees. Our work takes note of the oracle-based convergence guarantees of the TMA algorithm of Wang et al. (2024), and extends the approach to design an algorithm for approximate TMA with $\mathcal{O}(1/T^{1/2})$ convergence rate in the sample-based setting. The result corresponds to a slightly improved oracle-based rate $\mathcal{O}((1 - \frac{1}{M})^T)$ compared to the analysis of TMA in Wang et al. (2024) and further provides the additional sample complexity and estimation error from the estimation of the Q-values.

Despite the NP-Hardness result, recent work has also explored non-rectangular sets albeit with much weaker convergence guarantees (Li et al., 2023; Kumar et al., 2025). In addition to a non-tractable but globally optimal solution, Li et al. (2023) provides a conservative policy iteration algorithm for transition kernels that converges to a sub-optimal solution of an RMDP, with an error increasing based on the degree of non-rectangularity. While the rates in the paper hide the oracle for the value or gradient computation, it is the first to provide an approximate solution to general non-rectangular uncertainty sets. We exploit a similar technique and note its relation to the TMA algorithm, and then further analyse the overall algorithm with policy optimisation in the context of the non-rectangular RCMDP setting.

Obtaining tight and statistically based uncertainty sets is important for RMDP and RCMDP algorithms. The $\ell_1$ norm over the probability space is arguably the most widely used uncertainty set in the RMDP literature. For our purposes, it is convenient to use and arises naturally in our regret bounds. First, many other distance measures and statistics can be directly related to the $\ell_1$ norm, including $\ell_q$, total variation distances, Chi-squared sets, student-t statistics, Hoeffding or Chebyshev inequalities, the total variation distance, and KL-divergences. Beyond their statistical derivation and simplicity, $\ell_q$ sets are also fairly easy to project back onto the feasible set. The KL divergence is another distance metric with known analytical form for the transition kernel (see Table 7 in AppendixA) and with a direct connection to the norm. Second, as shown in Lemma 24, general uncertainty sets can be related to the $\ell_1$ norm, which appears naturally in the cumulative regret computation, and in this sense our main theorem is flexible to different divergences. In our convergence rate analysis, $\ell_q$ sets with small uncertainty budgets can potentially reduce one of the constants, although otherwise leading to the same convergence rate as in general uncertainty sets. Specialised algorithms have also been designed for the construction of tight weighted $\ell_q$ uncertainty sets while maintaining Bayesian confidence guarantees (e.g. Behzadian et al. (2019)).

Alternative models of uncertainty, suitable for indirect parametrisation and compatible with continuous state-action spaces, include examples such as the double sampling uncertainty sets, integral probability metrics uncertainty sets, and Wasserstein uncertainty sets (Zhou et al., 2023; Abdullah et al., 2019; Hou et al., 2007). Similarly, $\ell_q$ sets can also be applied to indirect parametrisations, which is compatible with the TMA algorithm (Wang et al., 2024). As supported by our experiments, Lagrangian TMA methods can similarly be applied with indirect parametrisations, such as the Gaussian Mixture PTK and the Entropy PTK (Wang et al., 2024).

## 2.3 Policy optimisation algorithms

Different policy opimisation algorithms have been proposed for RCMDPs and the related framework of CMDPs. As summarised in Table 1, the algorithms proposed in the literature work in the primal space, dual space, or a combination thereof, and they provide different guarantees under varying parametrisations.

In the CMDP literature on policy optimisation, many of the algorithms do not come with formal guarantees apart from local optima or improvement in the feasible space (Achiam et al., 2017; Tessler et al., 2019; Yang et al., 2020). Ding et al. (2020) present a primal-dual technique, NPG-PD, which obtains an $\mathcal{O}(1/T^{1/2})$ rate in the oracle-based setting for the softmax parametrisation, which extrapolates to an $\tilde{\mathcal{O}}(1/T^{1/4})$ bound in the sample-based setting. The work also presents a rate for a general parametrisation although it is considerably worse for the constraint-cost. CRPO applies NPG in the primal space with backtracking steps to ensure the constraint-cost remains satisfied, and is able to achieve comparable rates to NPG-PD (Xu et al., 2021). State-of-the-art results of $\tilde{\mathcal{O}}(1/T^{1/3})$ in the sample-based setting and $\tilde{\mathcal{O}}(1/T)$ in the oracle-based setting are obtained by PMD-PD, which applies a primal-dual mirror descent algorithm specifically with KL-divergence as a Bregman divergence (Liu et al., 2021b). Unlike other techniques using regularisation (Cen et al., 2022), the algorithm provides unbiased convergence results (i.e. they are expressed as the solution to the unregularised Lagrangian). PMD-PD does not account for variability in the transition kernel, and therefore we extend it to the RCMDP setting, obtaining comparable rates in the sample-based setting.

From the RMDP literature, the techniques by Wang et al. (2024) apply to the oracle-based rectangular RMDP setting, where for some $b \in (0, 1)$ they achieve a state-of-the-art $\mathcal{O}(b^T)$ convergence to the global optimum under a direct parametrisation of the policy and the transition kernel. For the softmax policy parametrisation, they report an $\mathcal{O}(1/T)$ oracle-based convergence rate in Theorem 4.5, a theorem which formulates the regret on the RMDP of interest based on the regret of an MDP optimiser passing through the same policy at the same iteration and with the transition kernel fixed to the nominal transition kernel at that iteration. However, their supporting Lemma B.5 incorrectly relies on a specific, favourable policy initialisation, which cannot be done in practice, rather than a general policy initialisation. The lemma also holds for any policy parameters at any iteration, such that no inference can be made about the regret of a particular policy in the RMDP optimisation simply by virtue of passing through the same parameters. Li et al. (2023) provides an $\mathcal{O}(1/T^{1/4})$ oracle-based result for non-rectangular RMDPs, the first convergence result for the global optimum of non-rectangular RMDPs considering the full algorithm solving both policy optimisation and the inner problem, while only requiring an approximate oracle for the gradient. The challenge of non-rectangularity shows up not only in the added complexity but also in additional error based on the degree of non-rectangularity. These works do not account for constraints or the samples to estimate the gradient; our work derives convergence results for the more challenging sample-based rectangular and non-rectangular RCMDP settings while maintaining similar global optimality properties.

In the RCMDP literature, policy optimisers either come with no policy convergence guarantees (Russel et al., 2020; Mankowitz et al., 2020; Bossens, 2024), basic feasibility and improvement results (Sun et al., 2024), or sub-optimal convergence points (Russel et al., 2023; Wang et al., 2022). Our sample-based theory for rectangular uncertainty sets provides results on par with CMDPs, namely an $\tilde{\mathcal{O}}(1/T^{1/3})$ convergence rate to the robust Lagrangian optimum, a result which is on par with the state-of-the-art PMD-PD algorithm (Liu et al., 2021b). This compares favourably to the $\mathcal{O}(1/T^{1/4})$ oracle-based rate of Wang et al. (2022) which comes only with local convergence guarantees. We also provide a sample-based theory for non-rectangular RCMDPs, where an $\tilde{\mathcal{O}}(1/T^{1/5})$ rate is achieved. Both of our mentioned convergence rates account for all the samples required, including those for gradient estimation, transition kernel updates, policy updates, and dual variable updates.

Our algorithm is also practical in the sense that it can be implemented with deep RL techniques. In particular, we use MDPO (Tomar et al., 2022) as the backbone of our algorithm's implementation. The technique has been previously studied in traditional MDPs as a highly performing alternative to trust region based deep policy optimisers (e.g. PPO, Schulman et al. (2015), and TRPO, Schulman et al. (2017)). While the clipping implemented in PPO also allows a KL-constrained property, it often loses this property since due to averaging effects the policy is updated even if its clipping ratio is out of bound for some state-action pairs. TRPO computes the Fisher information, which can be expensive, and performs a KL constraint with a reversed order (i.e. $D_{\mathrm{KL}}(\pi_k, \pi)$ instead of $D_{\mathrm{KL}}(\pi, \pi_k)$ as used in mirror descent). The MDPO backbone leads to a novel algorithm for RCMDPs as well as for CMDPs, which turns out to be highly performant as an alternative to PPO-Lagrangian (Ray et al., 2019).

Table 1: Overview of theoretical guarantees for policy optimisation algorithms in CMDPs and RCMDPs. **MDP** indicates the type of MDP, with prefixes C, R, and RC having their obvious meanings, while rR indicates rectangular robust and nR indicates non-rectangular robust. **Param** indicates the parametrisation. **Oracle rate** indicates the convergence rate in the oracle setting. **Sample rate** indicates the convergence rate in the sample-based setting. Other notations: $\tilde{\mathcal{O}}$ hides logarithmic factors from the big O notation; $b \in (0, 1)$.

| Algorithm | Algorithm type | MDP | Param | Convergence point | Oracle rate | Sample rate |
|---|---|---|---|---|---|---|
| (Achiam et al., 2017) | trust region dual program with backtracking | C | general | feasible improvement | / | / |
| (Tessler et al., 2019) | primal-dual with projection | C | general | local optimum | / | / |
| (Yang et al., 2020) | trust region with projection | C | general | local optimum | / | / |
| (Ding et al., 2020) | primal-dual NPG with projected subgradient descent | C | softmax | global optimum | $\mathcal{O}(1/T^{1/2})$ | $\tilde{\mathcal{O}}(1/T^{1/4})$ |
| (Xu et al., 2021) | primal unconstrained NPG with backtracking | C | softmax | global optimum | $\mathcal{O}(1/T^{1/2})$ | $\tilde{\mathcal{O}}(1/T^{1/4})$ |
| (Liu et al., 2021b) | primal-dual mirror descent | C | softmax | global optimum | $\tilde{\mathcal{O}}(1/T)$ | $\tilde{\mathcal{O}}(1/T^{1/3})$ |
| (Wang et al., 2024) | double-loop mirror descent/ascent | rR | direct | global optimum | $\mathcal{O}(b^T)$ | / |
| (Li et al., 2023) | projected descent and CPI ascent | nR | direct | global with error $\delta_{\mathcal{P}}$ | $\mathcal{O}(1/T^{1/4})$ | / |
| (Russel et al., 2023) | primal-dual, linear programming for inner problem | rRC | general | local optimum | / | / |
| (Wang et al., 2022) | primal-dual, projected descent with robust TD | rRC | general | stationary point | $\mathcal{O}(1/T^{1/4})$ | / |
| (Sun et al., 2024) | trust region with projection | rRC | general | feasible improvement | / | / |
| **Ours** | double-loop primal-dual mirror descent/ascent | rRC | softmax | global optimum | / | $\tilde{\mathcal{O}}(1/T^{1/3})$ |
| **Ours** | double-loop primal-dual mirror descent/ascent | nRC | softmax | global with error $\delta_{\mathcal{P}}$ | / | $\tilde{\mathcal{O}}(1/T^{1/5})$ |

## 3 Preliminaries

The setting of the paper is based on RCMDPs. Formally, an RCMDP is given by a tuple $(\mathcal{S}, \mathcal{A}, \rho, c_0, c_{1:m}, \mathcal{P}, \gamma)$, where $\mathcal{S}$ is the state space, $\mathcal{A}$ is the action space, $\rho \in \Delta(\mathcal{S})$ is the starting distribution, $c_0 : \mathcal{S} \times \mathcal{A} \to \mathbb{R}$ is the cost function, $c_i : \mathcal{S} \times \mathcal{A} \to \mathbb{R}, i \in [m]$ are constraint-cost functions, $\mathcal{P}$ is the uncertainty set, and $\gamma \in [0, 1)$ is a discount factor. To denote the cardinality of sets $\mathcal{S}$ and $\mathcal{A}$, we use the convention $S := |\mathcal{S}|$ and $A := |\mathcal{A}|$. In RCMDPs, at any time step $l = 0, \ldots, \infty$, an agent observes a state $s_l \in \mathcal{S}$, takes an action $a_l \in \mathcal{A}$ based on its policy $\pi \in \Pi \subseteq (\Delta(\mathcal{A}))^{\mathcal{S}}$, and receives cost $c_0(s_l, a_l)$ and constraint-cost signals $c_j(s_l, a_l)$ for all $j \in [m] = \{1, \ldots, m\}$. Following the action, the next state is sampled from a transition dynamics model $p \in \mathcal{P}$ according to $s_{l+1} \sim p(\cdot|s_l, a_l)$. The transition dynamics model $p$ is chosen from the uncertainty set $\mathcal{P}$ to solve a minimax problem. In particular, denoting

$$V_{\pi,p}^j(\rho) := \mathbb{E}_{s_0 \sim \rho}\left[\sum_{l=0}^{\infty} \gamma^l c_j(s_l, a_l)|\pi, p\right] \tag{1}$$

and

$$V_{\pi,p}(\rho) = \mathbb{E}_{s_0 \sim \rho}\left[\sum_{l=0}^{\infty} \gamma^l c_0(s_l, a_l)|\pi, p\right], \tag{2}$$

the goal of the agent in a traditional RCMDP is to optimise a minimax objective of the form

$$\min_{\pi} \sup_{p \in \mathcal{P}} V_{\pi,p}(\rho) \quad \text{s.t.} \quad \sup_{p_j \in \mathcal{P}} V_{\pi,p_j}^j(\rho) \leq 0, \ \forall j \in [m]. \tag{3}$$

The objective in Eq. 3 does not represent a single plausible transition kernel for the CMDP since the different suprema are usually met by different transition kernels. In practice, a choice is often made to take the supremum of the traditional cost only (Russel et al., 2020). Since this ignores the constraint-costs, one may instead optimise a related problem, namely the robust Lagrangian RCMDP objective (Bossens, 2024),

$$\min_{\pi} \left\{ \Phi(\pi) := \sup_{p \in \mathcal{P}} \max_{\lambda \geq 0} \left( V_{\pi,p}(\rho) + \sum_{j=1}^{m} \lambda_j V_{\pi,p}^j(\rho) \right) \right\}, \tag{4}$$

which only requires a single simulator and will allow optimisation to focus on constraint-costs if they are violated and on the traditional cost otherwise. If the inner optimisation problem is solved as $p_k$ at iteration

$k$, this in turn corresponds to an equivalent value function over a traditional MDP (Taleghan & Dietterich, 2018; Bossens, 2024) according to the Lagrangian value function,

$$\mathbf{V}_{\pi,p_k}(\rho) = \mathbb{E}\left[\sum_{l=0}^{\infty} \gamma^l \left(c_0(s_l, a_l) + \sum_{j=1}^{m} \lambda_{k,j} c_j(s_l, a_l)\right) \Big| s_0 \sim \rho, \pi, p_k, \right], \tag{5}$$

which has a single cost function defined by $\mathbf{c}_k(s, a) = c_0(s, a) + \sum_{j=1}^{m} \lambda_{k,j} c_j(s, a)$ and no constraint-costs. The multiplier $\lambda_k$ denotes the vector of dual variables at iteration $k$ and $\lambda_{k,j}$ denotes the multiplier for constraint $j \in [m]$ at iteration $k$. Occasionally, in proofs where the iteration is not needed, the iteration index will be omitted. In the context of dynamic updates to $\lambda$, we often emphasise the value of $\lambda$ by including it as a pararameter in the notation $\mathbf{V}_{\pi,p_k}(\rho; \lambda)$.

A convenient definition used in the following is the discounted state visitation distribution

$$d_\rho^{\pi_k, p_k}(s) = (1 - \gamma) \mathbb{E}_{s_0 \sim \rho}\left[\sum_{l=0}^{\infty} \gamma^l \mathbb{P}(s_l = s | s_0, \pi_k, p_k)\right], \tag{6}$$

and the discounted state-action visitation distribution

$$d_\rho^{\pi_k, p_k}(s, a) = (1 - \gamma) \mathbb{E}_{s_0 \sim \rho}\left[\sum_{l=0}^{\infty} \gamma^l \mathbb{P}(s_l = s, a_l = a | s_0, \pi_k, p_k)\right]. \tag{7}$$

For any optimisation problem $\min_{x \in X} f(x)$, we define three notions of regret:

$$f(x_k) - f(x^*) \tag{regret}$$

$$\sum_{k=1}^{K} f(x_k) - f(x^*) \tag{cumulative regret}$$

$$\frac{1}{K}\sum_{k=1}^{K} f(x_k) - f(x^*) \tag{average regret}$$

where $\{x_k\}_{k \geq 1}$ represent the iterates of the algorithm and $x^*$ is the optimum. Based on these regret notions, the paper provides a sample-based analysis in terms of how many samples are required until the regret is below some small threshold $\epsilon > 0$.

A first objective of interest is the maximisation over transition kernels within Eq. 3, i.e.

$$\sup_{p \in \mathcal{P}} \mathbf{V}_{\pi,p}(\rho; \lambda), \tag{8}$$

where $\lambda \in \mathbb{R}^m$ and $\pi \in \Pi$ are fixed. Our analysis aims to show that it is possible to obtain, within a limited number of samples using only approximate evaluations of the Lagrangian w.r.t. the current parameters, a near-optimal maximiser, i.e. a transition kernel with a limited regret,

$$\sup_{p \in \mathcal{P}} \mathbf{V}_{\pi,p}(\rho; \lambda) - \mathbf{V}_{\pi,p^t}(\rho; \lambda) \leq \epsilon, \tag{9}$$

where $p^t$ is the transition kernel being optimised at iteration $t \geq 1$.

A second objective of interest is the minimax optimisation problem in Eq. 3 itself. To design an optimal algorithm for the full RCMDP problem, the main goal of the sample-based analysis is to minimise the average regret w.r.t. Eq. 4,

$$\frac{1}{K}\sum_{k=1}^{K} \Phi(\pi_k) - \Phi(\pi^*) \leq \epsilon, \tag{10}$$

within a limited number of samples. To count the samples, we account for those used for the evaluation and optimisation of the policy and the transition kernel, for a full sample-based analysis.

## 4  PMD-PD for CMDPs

We briefly review the PMD-PD algorithm for CMDPs (Liu et al., 2021b). PMD-PD uses natural policy gradient with parametrised softmax policies, which is equivalent to the mirror descent.

At iteration $k \in \{0, \dots, K-1\}$ and policy update $t \in \{0, \dots, t_k\}$ within the iteration, PMD-PD defines the regularised state value function,

$$\tilde{\mathbf{V}}^{\alpha}_{\pi^t_k, p}(s) = \mathbb{E}\left[\sum_{l=0}^{\infty} \gamma^l \left(\tilde{\mathbf{c}}_k(s_l, a_l) + \alpha \log\left(\frac{\pi^t_k(a_l|s_l)}{\pi_k(a_l|s_l)}\right)\right) \Big| s_0 = s, \pi^t_k, p\right], \tag{11}$$

where $p$ is a fixed transition kernel, $\tilde{\mathbf{c}}_k(s_l, a_l) = c_0(s_l, a_l) + \sum_{j=1}^{m}\left(\lambda_{k,j} + \eta_\lambda V^j_{\pi_k, p_k}\right) c_j(s_l, a_l)$ and $\alpha \geq 0$ is the regularisation coeffient. The regularisation implements a weighted Bregman divergence which is similar to the KL-divergence. The augmented Lagrangian multiplier $\tilde{\lambda}_{k,j} = \lambda_{k,j} + \eta_\lambda V^j_{\pi_k, p_k}$ is used to improve convergence results by providing a more smooth Lagrangian compared to techniques that clip the Lagrangian multiplier. At the end of each iteration $\lambda_{k,j}$ is updated according to $\lambda_{k,j} = \max\left\{\lambda_{k,j} + \eta_\lambda V^j_{\pi_{k+1}, p_{k+1}}, -\eta_\lambda V^j_{\pi_{k+1}, p_{k+1}}\right\}$.

Analogous to 11, the regularised state-action value function is given by

$$\tilde{\mathbf{Q}}^{\alpha}_{\pi^t_k, p}(s, a) = \tilde{\mathbf{c}}_k(s, a) + \alpha \log\left(\frac{1}{\pi_k(a|s)}\right) + \gamma \mathbb{E}_{s' \sim p(\cdot|s, a)}\left[\mathbf{V}^{\alpha}_{\pi^t_k, p}(s')\right]. \tag{12}$$

While the bold indicates the summation over $1 + m$ terms (i.e. the costs and constraint-costs) as in Eq. 5, the tilde notations will be used throughout the text to indicate the use of the augmented Lagrangian multipliers $\tilde{\lambda}_k$. Note that since all the costs and constraint-costs range in $[-1, 1]$,

$$|\tilde{\mathbf{c}}_k(s, a)| \leq 1 + \sum_{j=1}^{m} \lambda_{k,j} + \frac{m\eta_\lambda}{1-\gamma}. \tag{13}$$

Consequently, for any set of multipliers $\lambda \in \mathbb{R}^m$, the Lagrangian is bounded by $[-F_\lambda/(1-\gamma), F_\lambda/(1-\gamma)]$ based on the constant

$$F_\lambda := 1 + \sum_{j=1}^{m} \lambda_j + \frac{\eta_\lambda m}{(1-\gamma)} \tag{14}$$

for the augmented Lagrangian, and according to

$$F_\lambda := 1 + \sum_{j=1}^{m} \lambda_j \tag{15}$$

for the traditional Lagrangian.

Note that this can be equivalently written based on the weighted Bregman divergence (equivalent to a pseudo KL-divergence over occupancy distributions),

$$B_{d^{\pi^t_k, p}_\rho}(\pi^t_k, \pi_k) = \sum_{s \in \mathcal{S}} d^{\pi^t_k, p}_\rho(s) \sum_{a \in \mathcal{A}} \pi^t_k(a|s) \frac{\log(\pi^t_k(a|s))}{\log(\pi_k(a|s))}, \tag{16}$$

according to

$$\tilde{\mathbf{Q}}^{\alpha}_{\pi^t_k, p}(s, a) = \tilde{\mathbf{Q}}_{\pi^t_k}(s, a) + \frac{\alpha}{1-\gamma} B_{d^{\pi^t_k}_\rho}(\pi^t_k, \pi_k). \tag{17}$$

In this way, PMD-PD applies entropy regularisation with respect to the previous policy as opposed to the uniformly randomized policy used in Cen et al. (2022), allowing PMD-PD to converge to the optimal unregularised policy as opposed to the optimal regularised (and sub-optimal unregularised) policy.

With softmax parametrisation, the policy is defined as

$$\pi_\theta(a|s) = \frac{e^{\theta_{s,a}}}{\sum_{a'} e^{\theta_{s,a'}}} \ . \tag{18}$$

Under this parametrisation, the mirror descent update over the unregularised augmented Lagrangian with the KL-divergence as the Bregman divergence can be formulated for each state independently according to

$$
\begin{aligned}
\pi_k^{t+1}(\cdot|s) &= \arg\min_{\pi \in \Pi} \left\{ \left\langle \frac{\partial \tilde{\mathbf{V}}_{\pi_k^t, p}(\rho)}{\partial \theta(s, \cdot)}, \pi(\cdot|s) \right\rangle + \frac{1}{\eta'} D_{\mathrm{KL}}(\pi(\cdot|s), \pi_k^t(\cdot|s)) \right\} \\
&= \arg\min_{\pi \in \Pi} \left\{ \frac{1}{1-\gamma} d_\rho^{\pi_k^t, p}(s) \left\langle \tilde{\mathbf{Q}}_{\pi_k^t, p}(s, \cdot), \pi(\cdot|s) \right\rangle + \frac{1}{\eta'} D_{\mathrm{KL}}(\pi(\cdot|s), \pi_k^t(\cdot|s)) \right\} \\
&= \arg\min_{\pi \in \Pi} \left\{ \left\langle \tilde{\mathbf{Q}}_{\pi_k^t, p}(s, \cdot), \pi(\cdot|s) \right\rangle + \frac{1}{\eta} D_{\mathrm{KL}}(\pi(\cdot|s), \pi_k^t(\cdot|s)) \right\} ,
\end{aligned}
\tag{19}
$$

where the last step defines $\eta = \eta'(1-\gamma)/d_\rho^{\pi_k^t, p}(s)$ and subsequently removes the constant $\frac{1}{1-\gamma} d_\rho^{\pi_k^t, p}(s)$ from the minimisation problem. Treating $\tilde{\mathbf{Q}}_{\pi_k^t, p}$ as any other value function, Eq. 19 is shown to be equivalent to the natural policy gradient update, as discussed by works in traditional MDPs (Zhan et al., 2023; Cen et al., 2022). In particular, based on Lemma 6 in Cen et al. (2022), the update rule derives from the properties of the softmax parametrisation in Eq. 18,

$$\pi_k^{t+1}(a|s) = \frac{1}{Z_k^t(s)} (\pi_k^t(a|s))^{1 - \frac{\eta\alpha}{1-\gamma}} e^{\frac{-\eta \tilde{\mathbf{Q}}_{\pi_k^t, p}^\alpha(s,a)}{1-\gamma}} , \tag{20}$$

where $\pi_k^{t+1}$ is the $t+1$'th policy at the $k$'th iteration of the algorithm and $Z_k^t(s)$ is the normalisation constant.

Denoting $T = \sum_{k=1}^K t_k$, the oracle version (with perfect gradient info) converges at rate $\mathcal{O}(\log(T)/T)$ in value and constraint violation. PMP-PD Zero (oracle version aiming for zero constraint violation) achieves 0 violation with the same convergence rate for the value. Finally, the sample-based version, which has imperfect gradient information, was shown to converge at a rate $\tilde{\mathcal{O}}(1/T^{1/3})$, where $T$ denotes the total number of samples.

## 5 Robust Sample-based PMD-PD for RCMDPs

Having reviewed the performance guarantees of PMD-PD, we now derive the Robust Sample-based PMD-PD algorithm for RCMDPs. The algorithm performs robust training based on Transition Mirror Ascent (TMA) (Wang et al., 2024) over the robust Lagrangian objective (Bossens, 2024). Using TMA provides a principled way to compute adversarial dynamics while maintaining the transition dynamics within the bounds of the uncertainty sets. Such uncertainty sets allow for rich parametrisation such as Gaussian mixtures, which can estimate rich probability distributions, and entropy-based parametrisations, which are motivated by the optimal form of KL-divergence based uncertainty sets (see Table 7 for these examples). The algorithm takes place in the sample-based setting, where the state(-action) values are approximated from a limited number of finite trajectories. With an average regret bound $\tilde{\mathcal{O}}(1/T^{1/3})$, results are in line with the Sample-based PMD-PD algorithm. The resulting algorithm is given in Algorithm 3. While the first subsection develops the theory of TMA in the oracle setting (Section 5.2), where exact values and policy gradients are given, the subsequent subsections derive convergence results for the sample-based setting, where values and policy gradients are estimates. In particular, Section 5.3 proposes and analyses Approximate TMA, an algorithm for TMA within the sample-based setting, and Section 5.4) proposes and analyses Robust Sample-based PMD-PD in the full sample-based RCMDP setting. Further extensions are then considered, namely non-rectangular uncertainty sets (Section 5.5) and continuous state-action spaces (Section 5.6). The last subsection (Section 5.7) then proposes a practical implementation.

### 5.1 General assumptions

As shown below, the assumptions of our analysis are common and straightforward.

**Assumption 1** (Bounded costs and constraint-costs). *Costs and constraint-costs are bounded in* $[-1, 1]$.

The boundedness assumption is ubiquitous in the MDP literature although some authors prefer the range $[0, 1]$ or some other constant. For our purposes, it implies that all $1 + m$ value functions in the Lagrangian are bounded by $\frac{1}{1-\gamma}$.

**Assumption 2** (Bounded range of the dual variables). *The dual variables are upper bounded by* $B_\lambda > 0$.

We assume that the dual variables are bounded by a constant, such that also $F_\lambda = \mathcal{O}(1)$, which allows bounding the regret of a policy with respect to the optimal Lagrangian in Eq. 4 to be bounded. The variables can take a wider range compared to other primal-dual algorithms due to the use of the augmented Lagrangian instead of clipping. The parameter is expected to be large for domains with highly active, restrictive constraints. The assumption is not particularly strong since it may be that $B_\lambda \gg \frac{2}{(1-\gamma)\zeta}$, the clipping bound usually taken for primal-dual variables (Ding et al., 2020). While with the modified Lagrangian update, the iterates obtained in the algorithm are always bounder, Assumption 2 additionally allows bounding the maximising dual variables of each iterate. In the sample complexity analysis of Robust Sample-based PMD-PD, the assumption is only required to limit the average regret induced by a mismatch of the dual variable with the optimal ones for that macro-iteration (see Lemma 29. There, the assumption could be removed when the dual variables are, on average, close to their optima of their respective inner problems.

**Assumption 3** (Sufficient exploration). *The initial state distribution satisfies* $\mu(s) > 0$ *for all* $s \in \mathcal{S}$.

This assumption is common in MDPs. It ensures that when sampling from some $\mu$ instead of the initial state distribution $\rho$, any importance sampling corrections with $\mu$ and $d_\mu^\pi$ are finite. In particular, it ensures that for any transition kernel $p$ and any policy $\pi$ that the mismatch coefficient $M_p(\pi) := \left\| \frac{d_\mu^{\pi,p}}{\mu} \right\|_\infty$ is finite (Mei et al., 2020). This is also useful for RMDPs (Wang et al., 2024) since

$$M := \sup_{p \in \mathcal{P}, \pi \in \Pi} M_p(\pi) \tag{21}$$

is also finite. Similarly, via Lemma 9 of Mei et al. (2020), it ensures $U_p = \inf_{s \in \mathcal{S}, k \geq 1} \pi_k(a_p^*(s)|s) > 0$ when following softmax policy gradient, where $a_p^*(s)$ is the action for state $s$ under the optimal deterministic policy under the transition dynamics $p$. Again, this also finds its use in RMDPs (Wang et al., 2024) where it follows that

$$U := \inf_{p \in \mathcal{P}} U_p > 0. \tag{22}$$

We use Eq. 22 to upper bound the KL divergence (Lemma 25), by further noting the relation to the global optimum $(\pi^*, p^*)$:

$$0 < U = \inf_{p \in \mathcal{P}, s \in \mathcal{S}, k \geq 1} \pi_k(a_p^*(s)|s) \leq \inf_{s \in \mathcal{S}, k \geq 1} \pi_k(a_{p^*}^*(s)|s) = \inf_{s \in \mathcal{S}, k \geq 1} \pi_k(a^*(s)|s). \tag{23}$$

**Assumption 4** (Slater's condition). *There exists a slack variable* $\zeta > 0$ *and (at least one) policy* $\bar{\pi}$ *with* $V_{\bar{\pi},p}^j(\rho) \leq -\zeta$ *for all* $j \in [m]$ *and all* $p \in \mathcal{P}$.

The condition is an extension of the traditional Slater's condition for CMDPs. The assumption of Slater points is not strong: since the constraints are based on expected values, they take on real values. Therefore, if it is possible to obtain the constraint-cost of zero, it is almost surely true that one can also obtain a negative constraint-cost (however small the slack variable). The possible non-negativity of instantaneous constraint-costs does not matter since at least a very small positive budget would be allowed.

In traditional CMDPs, Slater's condition implies strict feasibility of the constraints, zero duality gap, and complementary slackness for any convex optimisation problem, of which the CMDP is an instance due to its equivalence to a linear programming formulation (Altman, 1999). The assumption of Slater points is common in CMDPs, and it is inherited from (Liu et al., 2021a). It is a ubiquitous but often hidden assumption when using dual and primal-dual algorithms, since it is needed to satisfy strong duality, such that $\min_{\pi \in \Pi} \max_{\lambda \geq 0} \mathbf{V}_{\pi,p}(\rho; \lambda) = \max_{\lambda \geq 0} \min_{\pi \in \Pi} \mathbf{V}_{\pi,p}(\rho; \lambda)$. Many constrained RL algorithms do mention the assumption explicitly (Achiam et al., 2017; Liu et al., 2021b; Ding et al., 2020; 2021). Note that we do

not require knowledge of the parameter $\zeta$ while other primal-dual algorithms (Ding et al., 2020; 2021) need to know the parameter to clip the Lagrangian. Another approach, which assumes a Slater point but also does not require knowledge of $\zeta$ (Germano et al., 2024) bounds the dual variables by showing a strong no-regret property for both the primal and the dual regret minimizers. Approaches without Slater's condition are possible using primal linear programming (Altman, 1999; Efroni et al., 2020). Unfortunately, such techniques are less practical due to requiring estimates of the occupancy measure.

Assumption 4 is slightly more restrictive than in traditional CMDPs due to the requirement of the condition to hold for all $p \in \mathcal{P}$. The condition of having a subset of the policies being able to strictly satisfy all the constraints for all the transition dynamics reflects the RCMDP objective in Eq. 3 since it is presumed that the worst-case dynamic with respect to the constraints is solvable. For our purposes, the assumption ensures each of the individual CMDPs in the uncertainty set can be individually solved without duality gap similar to Liu et al. (2021b) and that the dual variable optimum and iterates (see Lemma 26 and Lemma 28) can be bounded by a constant that can be significantly smaller than the $B_\lambda$ that bounds the space of dual variables.

Due to the occupancy being a function of an induced transition kernel which is potentially different for each policy, a zero duality gap may be hard to guarantee for RCMDPs even with Slater points (Wang et al., 2022). However, our sample-based analysis provides an average regret upper bound with respect to the (primal) global optimum of the RCMDP $(\pi^*, p^*, \lambda^*)$ using Lemma 8, which is derived based on information from the iteration of the algorithm, namely the error tolerance of LTMA, the suboptimality of the dual variables, and the Lagrangian at $(\pi^*, p_k, \lambda_{k-1})$. Such errors are then shown to vanish in an average regret analysis.

Before stating the next assumption, we require the definition of rectangular uncertainty sets.

**Definition 1** (Rectangularity). *An uncertainty set is $(s,a)$-rectangular if it can be decomposed as $\mathcal{P} = \bigtimes_{(s,a) \in \mathcal{S} \times \mathcal{A}} \mathcal{P}_{s,a}$ where $\mathcal{P}_{s,a} \subseteq \Delta(\mathcal{S})$. Similarly, it is $s$-rectangular if it can be decomposed as $\mathcal{P} = \bigtimes_{s \in \mathcal{S}} \mathcal{P}_s$ where $\mathcal{P}_s \subseteq \Delta(\mathcal{S})$.*

With the above definition, our assumption on the uncertainty set is as follows.

**Assumption 5** (Convex and rectangular uncertainty set). *The uncertainty set is $s$-rectangular or $(s,a)$-rectangular (Def. 1) and is convex, such that for any pair of transition kernel parameters $\xi, \xi' \in \mathcal{U}_\xi$, and any $\alpha \in [0,1]$, if $p_\xi, p_{\xi'} \in \mathcal{P}$ then also $p_{\alpha\xi + (1-\alpha)\xi'} \in \mathcal{P}$.*

The convex set assumption ensures updates to the transition kernel will remain in the interior so long as it is on a line between two interior points, a common assumption for mirror descent which is also needed for TMA (Wang et al., 2024). The assumption is a standard (often hidden) assumption in gradient descent/ascent type algorithms to ensure the parameter remains in the feasible set.

The rectangular uncertainty set assumption is satisfied for most common uncertainty sets, e.g. state-action conditioned $\ell_1$ and $\ell_2$ sets centred around nominal probability distributions. In our main theorem, a tight rectangular $\ell_q$ uncertainty set is used in Lemma 24 to provide an improved upper bound on the average regret, although the same convergence rate is achieved for general uncertainty sets albeit with a larger constant. While Wang et al. (2024) mentions the policy optimisation part of their algorithm works with non-rectangular uncertainty sets in principle, the guarantees of the TMA algorithm only apply to rectangular uncertainty sets, where it can provide an ascent property to guarantee an increase in the objective. Consequently, Lagrangian TMA algorithms inherit this property.

Proposals for solving the inner problem of non-rectangular uncertainty sets do exist but most results are not encouraging. As noted by Wiesemann et al. (2013), the inner loops of non-rectangular and non-convex uncertainty sets are NP-Hard problems. Other alternatives include $r$-rectangular uncertainty sets, which could help reduce the number of distinct factors included to compose the uncertainty set. Recent work shows that $r$-rectangular and non-rectangular frameworks are only tractable if one assumes costs with no next-state dependencies, an assumption that may undermine their usefulness (Grand-Clément et al., 2024).

However, recent research has also highlighted some benefits of non-rectangular uncertainty sets, such as allowing statistically optimal confidence regions (Li et al., 2023) and covering the maximum likelihood areas rather than the edges (Kumar et al., 2025). These works also introduce techniques for approximately solving the inner and outer problems in the context of RMDPs. To supplement our analysis of rectangular sets, we

also consider non-rectangular RCMDPs in Section 5.5. The analysis uses the conservative policy iteration algorithm as defined for transition kernels in Li et al. (2023), which treats policy iteration as a special case of Approximate TMA.

Note that when using parametrised transition kernels, it is convenient to formulate an uncertainty set in the parameter space, which for a parameter $\xi$ we denote as $\mathcal{U}_\xi$. A few examples of parametrised transition kernels (PTKs) and their associated parametrised uncertainty sets can be found in Table 7 (see Appendix A), including the Entropy PTK (Wang et al., 2024), which presents a parametrisation related to the optimal parametrisation for KL divergence uncertainty sets of the form $\mathcal{P}_{s,a} = \{p : D_{\mathrm{KL}}(p, \bar{p}(\cdot|s, a)) \leq \kappa\}$ (Nilim & Ghaoui, 2005), and the Gaussian mixture PTK (Wang et al., 2024), which is able to capture a rich set of distributions for each state-action pair. Such uncertainty sets can be considered as general uncertainty sets although they are often also contained within an $\ell_q$ set.

## 5.2 Global optimality of Lagrangian TMA

As shown below, Lagrangian TMA (LTMA) demonstrably solves the maximisation problem in the robust objective (see Eq. 4) by performing mirror ascent on a suitable PTK. Note that the analysis follows closely that of Wang et al. (2024), where the Lagrangian and the weighted Bregman divergence are explicitly separated in this subsection.

The first step of our analysis of LTMA is to derive the gradient of the transition dynamics.

**Lemma 1** (Lagrangian transition gradient theorem). *Let $\mathbf{V}_{\pi,p}(\rho; \lambda)$ be the Lagrangian for a policy $\pi \in \Pi$, a transition kernel $p \in \mathcal{P}$ parametrised by $\xi$, and dual variables $\lambda \in \mathbb{R}^m$. Then the transition kernel has the gradient*

$$\nabla_\xi \mathbf{V}_{\pi,p}(\rho; \lambda) = \frac{1}{1-\gamma} \mathbb{E}_{s \sim d_\rho^{\pi,p}}[\nabla_\xi \log p(s'|s, a)(\mathbf{c}(s, a, s') + \gamma \mathbf{V}_{\pi,p}(s'; \lambda))|\pi, p]. \tag{24}$$

*Proof.* The result follows from Theorem 5.7 in Wang et al. (2024) by considering the Lagrangian as a value function (see Eq. 5). □

Note that the Lagrangian adversarial policy gradient in Bossens (2024) follows similar reasoning but there are two differences in the setting. First, due to the formulation of costs as $c_0(s, a, s')$ rather than $c_0(s, a)$, the state-action value of every time step appears rather than the value of every next time step. Second, we show the proof for a discounted Lagrangian of a potentially infinite trajectory rather than an undiscounted $T$-step trajectory. Third, in the following, we apply it not only to the traditional Lagrangian but also to the augmented Lagrangian and the augmented regularised Lagrangian.

The second step of our analysis of LTMA to show that optimising the transition dynamics $p_\xi$ while fixing the policy $\pi$ and the Lagrangian multipliers yields a bounded regret w.r.t. the global optimum.

**Lemma 2** (Regret of LTMA). *Let $\epsilon' > 0$, let $\pi$ be a fixed policy, let $\lambda \in \mathbb{R}^m$ be a fixed vector of multipliers for each constraint, let $p^t$ be the transition kernel at iteration $t$, and let $p^0$ be the transition kernel at the start of LTMA. Moreover, let $p^* = \arg\min_{p \in \mathcal{P}} \mathbf{V}_{\pi,p}(\rho; \lambda)$, let $M := \sup_{p \in \mathcal{P}, \pi \in \Pi} M_p(\pi)$ be an upper bound on the mismatch coefficient as in Eq. 21, and let $\eta_p > 0$ be the learning rate and $\alpha_p > 0$ be the penalty parameter for LTMA. Then for any starting distribution $\rho \in \Delta(\mathcal{S})$ and any $s \in \mathcal{S}$, the regret of LTMA is given by*

$$\mathbf{V}_{\pi,p^*}(\rho; \lambda) - \mathbf{V}_{\pi,p^t}(\rho; \lambda) \leq \frac{2F_\lambda}{t}\left(\frac{M}{(1-\gamma)^2} + \frac{\alpha_p}{\eta_p(1-\gamma)}B_{d_\rho^{\pi,p^*}}(p^*, p^0)\right), \tag{25}$$

*where $F_\lambda$ is defined according to Eq. 15 for the traditional Lagrangian and on Eq. 14 for the augmented Lagrangian. Moreover, if $\eta_p/\alpha_p \geq (1-\gamma)B_{d_\rho^{\pi,p^*}}(p_*, p_0)$ and $t \geq 2F_\lambda \frac{M+1}{\epsilon'(1-\gamma)^2}$, an $\epsilon'$-precise transition kernel is found such that*

$$\mathbf{V}_{\pi,p^*}(\rho; \lambda) - \mathbf{V}_{\pi,p^t}(\rho; \lambda) \leq \epsilon'.$$

*Proof.* Following Theorem 5.5 in Wang et al. (2024), it follows for any value function with cost in $[0, 1]$ that

$$V_{\pi,p^*}(\rho) - V_{\pi,p^t}(\rho) \leq \frac{1}{t}\left(\frac{M}{(1-\gamma)^2} + \frac{\alpha_p}{\eta_p(1-\gamma)}B_{d_\rho^{\pi,p^*}}(p^*, p^0)\right). \tag{26}$$

Reformulating the Lagrangian as a value function (see Eq. 5), considering the bounds on the absolute value of the constraint-cost Eq. 13, and observing that the bounds on the (un-)augmented Lagrangian lie in $[-F_\lambda, F_\lambda]$ (see Eq. 14 and 15), the result can be scaled to obtain

$$\mathbf{V}_{\pi,p^*}(\rho; \lambda) - \mathbf{V}_{\pi,p^t}(\rho; \lambda) \leq \frac{2F_\lambda}{t} \left( \frac{M}{(1-\gamma)^2} + \frac{\alpha_p}{\eta_p(1-\gamma)} B_{d_\rho^{\pi,p^*}}(p^*, p_0) \right).$$

From the settings of $\eta_p/\alpha_p \geq (1-\gamma)B_{d_\rho^{\pi,p^*}}(p^*, p_0)$ and $t \geq 2F_\lambda \frac{M+1}{\epsilon'(1-\gamma)^2}$, we obtain

$$\mathbf{V}_{\pi,p^*}(\rho; \lambda) - \mathbf{V}_{\pi,p^t}(\rho; \lambda) \leq \frac{2F_\lambda}{t} \left( \frac{M+1}{(1-\gamma)^2} \right) \leq \epsilon'.$$

$\square$

### 5.3 Analysis of Approximate TMA

While the above-mentioned LTMA algorithm was formulated in terms of exact policy gradient evaluations, such computations are intractable in practice. Now we turn to the design of an algorithm for TMA based on approximate policy gradient evaluations, which we call Approximate TMA (see Algorithm 1). Since it has not been proposed in earlier work, the algorithm will be presented in a generic manner, i.e. for general value functions. To maintain consistency across different value functions (including the Lagrangian and traditional value functions), we deal with their difference through a factor $C > 0$ that scales the maximal cost at each time step.

To construct the Approximate TMA algorithm, the definition of the action next-state value function provides a useful short hand that is an essential part of the transition gradient for a directly parametrised transition kernel.

**Definition 2** (Action next-state value function (Definition 3.2 in Li et al. (2023))). *For any cost function $c : \mathcal{S} \times \mathcal{A} \times \mathcal{S} \to \mathbb{R}$ and starting distribution $\rho$, the action next-state value function for transition kernel $p$ and policy $\pi$ is defined as*

$$G_{\pi,p}(s, a, s') = \mathbb{E} \left[ \sum_{l=0}^{\infty} \gamma^l c(s_l, a_l, s_{l+1}) | s_0 = s, a_0 = a, s_1 = s', \pi, p \right]. \tag{27}$$

*It can also be written as $G_{\pi,p}(s, a, s') = c(s, a, s') + \gamma V_{\pi,p}(s')$ (Wang et al., 2024).*

The directly parametrised transition kernel can now be written in terms of Definition 2 based on the lemma below.

**Lemma 3** (Partial derivative for approximate TMA). *Under the direct parametrisation, the partial derivative of $\hat{V}_{\pi,p}(\rho)$ has the form*

$$\frac{\partial}{\partial_{p_{s,a,s'}}} \hat{V}_{\pi,p}(\rho) = \frac{1}{1-\gamma} d_\rho^{\pi,p}(s)\pi(a|s)\hat{G}_{\pi,p}(s, a, s'). \tag{28}$$

*Proof.* The proof follows directly by applying Lemma 5.1 of Wang et al. (2024) (or Lemma 3.4 of Li et al. (2023) with direct parametrisation) to the approximate value function. $\square$

The resulting update rule is then given by

$$p^{t+1}(\cdot|s, a) = \arg \sup_{p \in \mathcal{P}} \eta_p(t)\frac{1}{1-\gamma}d_\rho^{\pi,p^t}(s, a)\hat{G}_{\pi,p^t}(s, a, \cdot) - \alpha_p B_{d_\rho^{\pi,p^t}}(p, p^t) \quad \text{for all } \mathcal{S} \times \mathcal{A}. \tag{29}$$

As a consequence of this update rule, the value difference in $p^{t+1}$ and $p^t$ can be bounded. In particular, the ascent property of TMA (Lemma 5.4 of Wang et al. (2024)) is a key result which provides a proof based on rectangularity (Definition 1) to show that TMA increases the objective over iterations. For our purposes, it is used not only in the oracle-based LTMA but also to prove a related ascent property for the Approximate

TMA algorithm. The property, which is given below, notes that conditioning on the state or state-action, a pushback property can be achieved, and this can be used to show that (under perfect gradient information) TMA updates always result in increasing value.

**Lemma 4** (Ascent property of TMA (Lemma 5.4 of Wang et al. (2024))). *Denoting* $G_{\pi,p}(s,\cdot,\cdot) = (\pi(a|s)G_{\pi,p}(s,a,\cdot))_{a\in\mathcal{A}}$, *for any* $p \in \mathcal{P}_s$ *and* $\pi \in \Pi$, *we have that*

$$\langle G_{\pi,p^t}(s,\cdot,\cdot), p - p^{t+1}(\cdot|s,\cdot)\rangle + B(p^{t+1}(\cdot|s,\cdot), p^t(\cdot|s,\cdot)) \leq B(p, p^t(\cdot|s,\cdot)) - B(p, p^{t+1}(\cdot|s,\cdot)), \tag{30}$$

*where* $B(p,q)$ *is the unweighted Bregman divergence between* $p$ *and* $q$. *Moreover,*

$$V_{\pi,p^t}(\rho) - V_{\pi,p^{t+1}}(\rho) \leq 0. \tag{31}$$

With imperfect information on the gradient, a monotonic increase in value cannot be guaranteed. However, it is possible to bound the potential decrease, resulting in the following ascent property of Approximate TMA.

**Lemma 5** (Ascent property of Approximate TMA). *Let* $t \geq 0$ *be the current iteration, let* $V : \mathcal{S} \to \mathbb{R}$ *be a value function with instantaneous cost bounded in* $[-C, C]$, *and let* $\pi \in \Pi$. *Suppose we have an estimate* $\hat{G} : \mathcal{S} \times \mathcal{A} \times \mathcal{S} \to \mathbb{R}$ *of the action next-state value function such that*

$$\left\|G_{\pi,p^t} - \hat{G}_{\pi,p^t}\right\|_\infty \leq \epsilon'. \tag{32}$$

*Then at iteration* $t + 1$, *Approximate TMA satisfies*

$$V_{\pi,p^t}(\rho) - V_{\pi,p^{t+1}}(\rho) \leq \frac{2\epsilon'}{1-\gamma}. \tag{33}$$

*Proof.* Using the first performance difference lemma (Lemma 23) and the ascent property (Eq. 31), we have that

$$V_{\pi,p^t}(\rho) - V_{\pi,p^{t+1}}(\rho) = \frac{1}{1-\gamma}\sum_{s\in\mathcal{S}}d_\rho^{\pi,p^{t+1}}(s)\sum_{a\in\mathcal{A}}\pi(a|s)\langle p^t(\cdot|s,a) - p^{t+1}(\cdot|s,a), G_{\pi,p^t}(s,a,\cdot)\rangle \qquad \text{(Lemma 22)}$$

$$= \frac{1}{1-\gamma}\sum_{s\in\mathcal{S}}d_\rho^{\pi,p^{t+1}}(s)\sum_{a\in\mathcal{A}}\pi(a|s)\Big(\langle p^t(\cdot|s,a) - p^{t+1}(\cdot|s,a), \hat{G}_{\pi,p^t}(s,a,\cdot)\rangle +$$

$$\langle p^t(\cdot|s,a) - p^{t+1}(\cdot|s,a), G_{\pi,p^t}(s,a,\cdot) - \hat{G}_{\pi,p^t}(s,a,\cdot)\rangle\Big)$$

$$\text{(decomposing)}$$

$$\leq \frac{1}{1-\gamma}\sum_{s\in\mathcal{S}}d_\rho^{\pi,p^{t+1}}(s)\sum_{a\in\mathcal{A}}\pi(a|s)\langle p^t(\cdot|s,a) - p^{t+1}(\cdot|s,a), G_{\pi,p^t}(s,a,\cdot) - \hat{G}_{\pi,p^t}(s,a,\cdot)\rangle$$

$$\text{(Eq. 31)}$$

$$\leq \frac{1}{1-\gamma}\sum_{s\in\mathcal{S}}d_\rho^{\pi,p^{t+1}}(s)\sum_{a\in\mathcal{A}}\pi(a|s)\left\|G_{\pi,p^t}(s,a,\cdot) - \hat{G}_{\pi,p^t}(s,a,\cdot)\right\|_\infty\left\|p^t(\cdot|s,a) - p^{t+1}(\cdot|s,a)\right\|_1$$

$$\text{(Hölder's inequality)}$$

$$\leq \frac{2\epsilon'}{1-\gamma}. \tag{Eq. 32}$$

$\square$

With the ascent property established, the theorem below bounds the regret of Approximate TMA. In the context of Robust Sample-based PMD-PD, the following algorithm can be used to obtain the updated transition kernel $p_{k+1}$ such that it closely matches $p_k^*$, the worst-case transition kernel for a particular macro-iteration, using the Lagrangian as a value function. However, since it is also of independent interest, it is formulated more generally here with respect to any policy and independent of the macro-iteration.

**Theorem 1** (Regret bound for Approximate TMA). *Let $V : \mathcal{S} \to \mathbb{R}$ be a value function with instantaneous cost bounded in $[-C, C]$, fix $\pi \in \Pi$, and let $p^* \in \arg\sup_{p \in \mathcal{P}} V_{\pi,p}(\rho)$. Suppose we have an estimate $\hat{G} : \mathcal{S} \times \mathcal{A} \times \mathcal{S} \to \mathbb{R}$ of the action next-state value function which is $\epsilon'$-precise as in Eq. 32, and the step sizes follow $\eta_p(0) \geq \alpha_p(0) \frac{\gamma}{C(1-\gamma)} B_{d_\rho^\pi, p^0}(p^*, p^0)$, $\eta_p(t) \geq \eta_p(t-1)/\gamma$ and $\alpha_p(t) = \alpha_p(0) > 0$ for all $t \geq 1$. It follows that approximate TMA (Algorithm 1) yields a regret bounded by*

$$V_{\pi,p^*}(\rho) - V_{\pi,p^t}(\rho) \leq \left(1 - \frac{1}{M}\right)^t \frac{3C}{1-\gamma} + \frac{4M}{1-\gamma}\epsilon'. \tag{34}$$

*for all $t \geq 1$, where $M$ is given by Eq. 21.*

*Proof.* The proof decomposes the approximate G-value into the true G-value and the estimation error, which allows using the ascent property of TMA and a limited factor $\mathcal{O}(\epsilon')$, followed by further derivations based on pushback properties. The full proof is given in Appendix C.2. □

---

**Algorithm 1** Approximate TMA.

---

1: Inputs: Discount factor $\gamma \in [0, 1)$, sample size $M_{G,t}$, episode lengths $N_{G,t}$, and learning rate schedule $\eta_p(t)$ for all $t \in \{0, \ldots, t'_k - 1\}$, penalty coefficient $\alpha_p > 0$, cost function $c : \mathcal{S} \times \mathcal{A} \to \mathbb{R}$, initial transition kernel $p^0$, and fixed policy $\pi$.
2: **for** $t = 0, \ldots, t'_k - 1$ **do**
3:     Generate $M_{G,t}$ samples of length $N_{G,t}$ based on $\pi$ and $p^t$.
4:     **if** using function approximation **then**
5:         $\hat{G}_{\pi,p^t} \leftarrow \arg\min_{G \in \mathcal{F}_G} \frac{1}{M_{G,t}} \sum_{j=1}^{M_{G,t}} \left(G(s_0^j, a_0^j, s_1^j) - \sum_{l=0}^{N_{G,t}-1} \gamma^l \left[c(s_l^j, a_l^j, s_{l+1}^j)\right]\right)^2$.
6:     **else**
7:         **for** $(s, a, s') \in \mathcal{S} \times \mathcal{A} \times \mathcal{S}$ **do**
8:             $\hat{G}_{\pi,p^t}(s, a, s') = \frac{1}{M_{G,t}} \sum_{j=1}^{M_{Q,k}} \sum_{l=1}^{N_{G,t}-1} \gamma^l c(s_l^j, a_l^j, s_{l+1}^j)$ where $s_0 = s, a_0 = a, s_1 = s'$.
9:         **end for**
10:    **end if**
11:    **if** using direct parametrisation **then**
12:        $p^{t+1}(\cdot|s, a) = \arg\sup_{p \in \mathcal{P}} \eta_p(t) \left\langle p, \hat{G}_{\pi,p^t}(s, a, \cdot) \right\rangle - \alpha_p B_{d_\rho^{\pi,p^t}}(p, p^t)$ for all $\mathcal{S} \times \mathcal{A}$.
13:    **else**
14:        $\xi^{t+1} \leftarrow \arg\max_{\xi \in \mathcal{U}} \eta_p(t) \left\langle \nabla_\xi \hat{V}_{\pi,p^t}(\rho), \xi \right\rangle - \alpha_p B_{d_\rho^{\pi,p^t}}(\xi, \xi^t)$.     ▷ Generic form
15:        Define $p^{t+1} := p_{\xi^{t+1}}$.
16:    **end if**
17: **end for**

---

Theorem 1 relies on an $\epsilon$-precise estimate of the action next-state value, which can be established within $\mathcal{O}(\epsilon^2)$, as shown in the lemma below.

**Lemma 6** (Approximate action next-state value). *Let $k \geq 0$, let $\delta \in (0, 1)$ and let $t \in \{0, \ldots, t'_k - 1\}$. Moreover, let $C > 0$ be the maximal absolute value of the instantaneous cost. Then provided the number of trajectories for the estimator $\hat{G}$ (l.7–9 of Algorithm 1) satisfies*

$$M_{G,t} \geq \frac{2\gamma^{-2N_{G,t}}}{(1-\gamma)^2} \log\left(\frac{2S^2 A t'_k}{\delta}\right),$$

*Algorithm 1 provides an approximately correct estimate such that with probability at least $1 - \delta$,*

$$\left\|\hat{G}_{\pi,p^t} - G_{\pi,p^t}\right\|_\infty \leq 2C\frac{\gamma^{N_{G,t}}}{1-\gamma} \quad \text{for all } t \in \{0, \ldots, t'_k - 1\}. \tag{35}$$

*Proof.* The proof of this lemma is based on Hoeffding's inequality and is given in Appendix C.3. □

Due to the above results, the sample complexity of Approximate TMA can be bounded by $\tilde{\mathcal{O}}\left(\epsilon^{-2}\right)$.

**Lemma 7** (Approximate TMA sample complexity). *Let $k \geq 0$, Let $\epsilon' > 0$, $\delta \in (0,1)$, and define $M$ according to Eq. 21. Moreover, let $C > 0$ be the maximal absolute value of the instantaneous cost, and let*

$$t'_k \geq \log_{\frac{M}{M-1}}\left(\frac{6C}{\epsilon'(1-\gamma)}\right).$$

*Define step sizes $\eta_p(0) \geq \frac{\gamma}{1-\gamma} B_{d_\rho^{\pi,p^*}}(p^*, p^0)$, $\eta_p(t) = \eta_p(t-1)/\gamma$ and $\alpha_p(t) = \alpha_p(0)$ for all $t \in [t'_k]$. Moreover, let*

$$N_{G,t} = H \geq \log_{\frac{1}{\gamma}}\left(\frac{16MC}{(1-\gamma)^2\epsilon'}\right)$$

*and*

$$M_{G,t} = M_G = \frac{2\gamma^{-2H}}{(1-\gamma)^2}\log\left(\frac{2S^2At'_k}{\delta}\right)$$

*for all $t \in [t'_k]$. Then Algorithm 1 yields a transition kernel $p^{t'_k}$ such that*

$$V_{\pi,p^*}(\rho) - V_{\pi,p^{t'_k}}(\rho) \leq \epsilon' \tag{36}$$

*within at most*

$$n = \tilde{\mathcal{O}}\left(\frac{S^2AM^3}{(1-\gamma)^4\epsilon'^2}\right) \tag{37}$$

*samples.*

*Proof.* The proof bounds both parts in the RHS of Eq. 34 with $\epsilon'/2$ and then counts the number of state-action-state triples, the number of iterations, the horizon, and the number of trajectories per mini-batch. The full proof is given in Appendix C.4. □

## 5.4 Analysis of Robust Sample-based PMD-PD

Having shown results Lagrangian TMA in oracle-based and sample-based settings, we now turn to proving the sample complexity of Robust Sample-based PMD-PD (Algorithm 3). The algorithm modifies Sample-based PMD-PD (Liu et al., 2021b) by accounting for the mismatch between the nominal transition kernel and the optimal transition kernel in the min-max problem in Eq. 4. The complexity analysis below is slightly simplified by dropping purely problem-dependent constants (e.g. $\gamma$, $\eta$, $\zeta$) from the big-O statements.

As a first auxiliary result, we note that the robust Lagrangian objective in Eq. 4 can be upper bounded based on the Lagrangian at macro-iteration $k$, obtained after the policy update and the LTMA update, and two error terms, namely the LTMA error tolerance and the suboptimality of the dual variable. These terms can then be bounded using Lemma 2, Lemma 34, and Lemma 29.

**Lemma 8** (Upper bound on robust Lagrangian regret). *Let $\Phi$ be the objective in Eq. 4 and let $\pi^* \in \arg\inf_{\pi \in \Pi} \Phi(\pi)$. Then for any macro-step $k \geq 1$,*

$$\Phi(\pi_k) - \Phi(\pi^*) \leq \mathbf{V}_{\pi_k,p_k}(\rho; \lambda_{k-1}) + \epsilon'_k + \Delta_{\lambda_k} - \mathbf{V}_{\pi^*,p_k}(\rho; \lambda_{k-1}), \tag{38}$$

*where $\epsilon'_k > 0$ is the error tolerance for LTMA, and $\Delta_{\lambda_k} = \sum_{j=1}^m (\lambda_{k-1,j}^* - \lambda_{k-1,j}) V_{\pi_k, p_{k-1}^*}^j(\rho)$ where $(p_{k-1}^*, \lambda_{k-1}^*) = \arg\max_{p \in \mathcal{P}, \lambda \geq 0} \mathbf{V}_{\pi_k,p}(\rho; \lambda)$.*

*Proof.* Note that

$$\Phi(\pi_k) = \mathbf{V}_{\pi_k, p_{k-1}^*}(\rho; \lambda_{k-1}^*)$$

$$= \sup_{p \in \mathcal{P}} \mathbf{V}_{\pi_k, p}(\rho; \lambda_{k-1}) + \left( \mathbf{V}_{\pi_k, p_{k-1}^*}(\rho; \lambda_{k-1}^*) - \sup_{p \in \mathcal{P}} \mathbf{V}_{\pi_k, p}(\rho; \lambda_{k-1}) \right) \qquad \text{(decomposing)}$$

$$\leq \sup_{p \in \mathcal{P}} \mathbf{V}_{\pi_k, p}(\rho; \lambda_{k-1}) + \left( \mathbf{V}_{\pi_k, p_{k-1}^*}(\rho; \lambda_{k-1}^*) - \mathbf{V}_{\pi_k, p_{k-1}^*}(\rho; \lambda_{k-1}) \right)$$

$$(p_{k-1}^* \text{ is optimal for } \lambda_{k-1}^* \text{ but not for } \lambda_{k-1})$$

$$= \sup_{p \in \mathcal{P}} \mathbf{V}_{\pi_k, p}(\rho; \lambda_{k-1}) + \left( V_{\pi_k, p_{k-1}^*}(\rho) - V_{\pi_k, p_{k-1}^*}(\rho) + \sum_{j=1}^{m} \left( \lambda_{k-1,j}^* - \lambda_{k-1,j} \right) V_{\pi_k, p_{k-1}^*}^j(\rho) \right)$$

$$\text{(definition of Lagrangian, rearranging)}$$

$$\leq \mathbf{V}_{\pi_k, p_k}(\rho; \lambda_{k-1}) + \epsilon_k + \Delta_{\lambda_k} \,,$$

where the last step follows from the error tolerance of LTMA and the definition of $\Delta_{\lambda_k}$.
Eq. 38 then follows from $\Phi(\pi^*) = \sup_{p \in \mathcal{P}} \max_{\lambda \geq 0} \mathbf{V}_{\pi^*, p}(\rho; \lambda) \geq \mathbf{V}_{\pi^*, p_k}(\rho; \lambda_{k-1})$. □

As sample-based algorithms work with approximate value functions, Lemma 9 describes the conditions under which one can obtain an $\epsilon$-approximation of the cost functions and the regularised augmented Lagrangian. The same settings as sample-based PMD in Lemma 15 of Liu et al. (2021b) transfer to the Robust Sample-based PMD algorithm. We rephrase the lemma to include updated transition dynamics.

**Lemma 9** (Value function approximation, Lemma 15 of Liu et al. (2021b) rephrased)**.** *Let $k \geq 0$ be the macro-step of the PMD-PD algorithm. With parameter settings*

$$K = \Theta\left(\frac{1}{\epsilon}\right), \quad t_k = \Theta\left(\log(\max\{1, \|\lambda_k\|_1\})\right), \quad \delta_k = \Theta\left(\frac{\delta}{K t_k}\right),$$

$$M_{V,k} = \Theta\left(\frac{\log(1/\delta_k)}{\epsilon^2}\right), \quad N_{V,k} = \Theta\left(\log_{1/\gamma}\left(\frac{1}{\epsilon}\right)\right),$$

$$M_{Q,k} = \Theta\left(\frac{(\max\{1, \|\lambda_k\|_1\} + \epsilon t_k)^2 \log(1/\delta_k)}{\epsilon^2}\right), \quad N_{Q,k} = \Theta\left(\log_{1/\gamma}\left(\frac{(\max\{1, \|\lambda_k\|_1\}}{\epsilon}\right)\right),$$

*the approximation is $\epsilon$-optimal, i.e.*

$$\textbf{a)} \quad |\hat{V}_{\pi_k^t, p_k}^i(\rho) - V_{\pi_k^t, p_k}^i(\rho)| \leq \epsilon \qquad \forall i = 1 \in [k]$$

$$\textbf{b)} \quad |\hat{\mathbf{Q}}_{\pi_k^t, p_k}^\alpha(s, a) - \tilde{\mathbf{Q}}_{\pi_k^t, p_k}^\alpha(s, a)| \leq \epsilon \qquad \forall (s, a) \in \mathcal{S} \times \mathcal{A} \,,$$

*with probability $1 - \delta$.*

*Proof.* The full proof can be found in Appendix D.5. □

Denoting $\tilde{\mathbf{V}}$ as the unregularised augmented Lagrangian, we can show a relation between two otherwise unrelated policies based on their divergence to two consecutive policy iterates.

**Lemma 10** (Bound on regularised augmented Lagrangian under approximate entropy-regularised NPG, Lemma 18 in Liu et al. (2021b))**.** *Assume the same settings as in Lemma 9. Then the regularised augmented Lagrangian is bounded by*

$$\tilde{\mathbf{V}}_{\pi_{k+1}, p_k}(s) + \frac{\alpha}{1-\gamma} B_{d_\rho^{\pi_{k+1}, p_k}}(\pi_{k+1}, \pi_k) \leq \tilde{\mathbf{V}}_{\pi, p_k}(s) + \frac{\alpha}{1-\gamma}\left(B_{d_\rho^{\pi, p_k}}(\pi, \pi_k) - B_{d_\rho^{\pi, p_k}}(\pi, \pi_{k+1})\right) + \Theta(\epsilon) \quad (39)$$

*Proof.* The proof makes use of Lemma 20. In particular, it notes that $K = \Theta(1/\epsilon)$ and the value difference to $\pi_k^*$ is bounded by $1/K$ after $t_k$ steps. After applying the pushback property (Lemma 17) to $\pi$, $\pi_k$, and $\pi_k^*$, and derives an upper bound on the Bregman divergence w.r.t. $\pi_k^*$ based on $B_{d_\rho^{\pi, p_k}}(\pi, \pi_{k+1}) + \left\|\log(\pi_k^*) - \log(\pi_k^{t+1})\right\|_\infty$, where the second term is $\Theta(1/K)$. We refer to Lemma 18 of Liu et al. (2021b) for the detailed proof. □

The theorem below proves the sample complexity of Robust Sample-based PMD-PD, using Lemma 9 to compute estimates of the Q-values for policy updates and using Lemma 6 to compute estimates of the G-values for transition kernel updates. This then allows to demonstrate the overall sample complexity for obtaining an average regret in the value, constraint-cost, and Lagrangian bounded by $\mathcal{O}(\epsilon)$.

**Theorem 2** (Sample complexity of Robust Sample-based PMD-PD). *Choose parameter settings according to Lemma 9, let $\alpha = \frac{2\gamma^2 m \eta_\lambda}{(1-\gamma)^3}$, $\eta_\lambda = 1$, and $\eta = \frac{1-\gamma}{\alpha}$. Moreover, let $\pi^* \in \arg\inf_{\pi \in \Pi} \Phi(\pi)$, let $\epsilon'_k > 0$ be the upper bound on the LTMA error tolerance at macro-iteration $k$ such that $\epsilon'_k = \Theta(\epsilon)$, and let $\mathcal{P}$ be a rectangular uncertainty set according to Lemma 24**a)** or **b)**. Then within a total number of queries to the generative model equal to $T = \tilde{\mathcal{O}}(\epsilon^{-3})$, Robust Sample-based PMD-PD provides the following guarantees:*
*(a) average regret upper bound:*

$$\frac{1}{K} \sum_{k=1}^{K} (V_{\pi_k, p_k}(\rho) - V_{\pi^*, p_k}(\rho)) = \mathcal{O}(\epsilon).$$ (40)

*(b) average constraint-cost upper bound:*

$$\max_{j \in [m]} \frac{1}{K} \sum_{k=1}^{K} V_{\pi_k, p_k}^j(\rho) = \mathcal{O}(\epsilon).$$ (41)

*c) robust Lagrangian average regret upper bound under (oracle-based) LTMA: under the conditions of Lemma 2, with a number of LTMA iterations $t'_k = \Theta\left(\frac{F_{\lambda_k} M}{\epsilon'_k (1-\gamma)}\right)$, and with $\mathcal{P}$ defined according to the preconditions of Lemma 24**b)**, we have*

$$\frac{1}{K} \sum_{k=1}^{K} \Phi(\pi_k) - \Phi(\pi^*) = \mathcal{O}(\epsilon).$$

*d) robust Lagrangian average regret upper bound under Approximate LTMA: under the conditions of Lemma 7 as applied to the Lagrangian at each macro-iteration, with $C = F_\lambda$, and with $\mathcal{P}$ defined according to the preconditions of Lemma 24**b)**, we have*

$$\frac{1}{K} \sum_{k=1}^{K} \Phi(\pi_k) - \Phi(\pi^*) = \mathcal{O}(\epsilon).$$

*Proof.* A proof sketch is given below for convenience while the full deriviation is given in Appendix D.6.2.
**(a)** The proof uses Lemma 10 and Lemma 15 to obtain a telescoping sum similar to the analysis of Theorem 3 in Liu et al. (2021a). To account for transition kernel changes, it then upper bounds the cumulative regret difference using Lemma 24) and the Bregman divergence terms in the RHS of Eq. 39 using Lemma 25. This then results in a telescoping sum.
**(b):** The proof uses the update rule of the dual variable and its relation to the approximate constraint-cost to obtain a telescoping sum and an additional term $\mathcal{O}(\epsilon)$.
**c):** The proof uses Lemma 8 to upper bound the robust Lagrangian regret with respect to $(\pi^*, p^*, \lambda^*)$ by using the regret bounds with respect obtained in **a)** and **b)**. Noting that the regret from oracle-based LTMA (Lemma 2) is at most $\epsilon'_k = \mathcal{O}(\epsilon)$ for each $k$, and that the average regret induced by the dual variable is bounded by $\mathcal{O}(\epsilon)$ (via Lemma 29).
**d):** The proof is analogous to part **c)** but now applies Approximate TMA using the settings of Lemma 7.
**Sample complexity:** To obtain the sample complexity under Algorithm 3 with Approximate TMA, plug in the settings from Lemma 9 and Lemma 7 to obtain

$$T = \sum_{k=1}^{K} \left( M_{V,k} N_{V,k} + \sum_{t=0}^{t_k - 1} M_{Q,k} N_{Q,k} + \sum_{t=0}^{t'_k - 1} M_{G,t} N_{G,t} \right) = \tilde{\mathcal{O}}(\epsilon^{-3}).$$ (42)

$\square$

## 5.5 Extension to non-rectangular uncertainty sets

Similar to arguments made by Xiao (2022) for Policy Mirror Descent, oracle-based TMA has a direct relation to the policy iteration algorithm.

**Lemma 11** (Relation between TMA and Policy Iteration). *Let $\eta_p(t) \to \infty$ for all $t \in \{0, \ldots, t'_k - 1\}$. Then under the direct parametrisation, TMA is equivalent to Policy Iteration as defined for transition kernels in Li et al. (2023).*

*Proof.* Note that for $\eta_p(t) \to \infty$, applying the TMA update rule, analogous to Eq. 29,

$$
\begin{aligned}
p^{t+1}(\cdot|s,a) &= \arg\sup_{p \in \mathcal{P}} \eta_p(t) \frac{1}{1-\gamma} d_\rho^{\pi,p^t}(s,a) G_{\pi,p^t}(s,a,\cdot) - \alpha_p B_{d_\rho^{\pi,p^t}}(p,p^t) \\
&= \arg\sup_{p \in \mathcal{P}} \langle p, G_{\pi,p^t}(s,a,\cdot)\rangle - \frac{\alpha_p}{\eta_p(t)} B(p,p^t) \qquad \text{(reweighting similar to Eq. 19)} \\
&= \arg\sup_{p \in \mathcal{P}} \langle p, G_{\pi,p^t}(s,a,\cdot)\rangle \,,
\end{aligned}
$$

which is equivalent to policy iteration. The latter can also be written as $\arg\sup_{p \in \mathcal{P}} \langle p - p^t, G_{\pi,p^t}(s,a,\cdot)\rangle$, which based on Lemma 3 is equivalent to

$$
\arg\sup_{p \in \mathcal{P}} \langle p - p^t, \nabla V_{\pi,p^t}(\rho)\rangle \tag{43}
$$

as in Eq. 3.4 of Li et al. (2023). $\qquad\square$

---

**Algorithm 2** Conservative Policy Iteration over transition kernels using Approximate TMA

---

1: Inputs: $\epsilon' > 0$, policy $\pi \in \Pi$, value function $V : \mathcal{S} \to \mathbb{R}$, initial transition kernel $p^0$, compact and convex uncertainty set $\mathcal{P}$, learning rate $\eta_p(t) \to \infty$ for all $t \geq 0$.
2: **while** True **do**
3:      One step of Approximate TMA with learning rate $\eta_p(t)$ to obtain $p_{\epsilon'}$ as $\epsilon'$-optimal solution to Eq. 43.
4:      Compute Frank-Wolfe gap $\mathbb{G}_t = \langle \nabla V_{\pi,p^t}(\rho), p_{\epsilon'} - p^t \rangle$.
5:      **if** $\mathbb{G}_t \leq \epsilon'$ **then**
6:          **return** $\hat{p} = p^t$.
7:      **else**
8:          Let $\beta_t = \mathbb{G}_t \frac{(1-\gamma)^3}{4\gamma^2}$
9:          Let $p^{t+1} = (1 - \beta_t)p^t + \beta_t p_{\epsilon'}$.
10:          $t \leftarrow t + 1$.
11:      **end if**
12: **end while**

---

Exploiting the above relation, and using Conservative Policy Iteration to transition kernels according to Algorithm 2, we can obtain the sample complexity under non-rectangular RCMDPs using arguments by Li et al. (2023). In particular, while Wang et al. (2024) provide an analysis of TMA only for $s$-rectangular RMDPs and derives oracle-based convergence rates for the policy optimiser, it is possible to extend the analysis to non-rectangular uncertainty set by relating it to the smallest $s$-rectangular uncertainty set that contains it. With this reasoning, the result below considers the Appproximate TMA algorithm for non-rectangular RCMDPs and then derives a full sample-based analysis of the full algorithm optimising all three parameters $(\pi, p, \lambda)$. The Algorithm 2 requires $t'_k = \mathcal{O}(\epsilon^{-2})$ iterations of the Approximate TMA algorithm. Therefore, with derivations as in Theorem 2, a sample complexity of $\tilde{\mathcal{O}}(\epsilon^{-5})$ is obtained. In addition to the added complexity, the non-rectangularity also comes with an additional error term based on the degree of non-rectangularity $\delta_\mathcal{P}$, defined as

$$
\delta_\mathcal{P} = \max_{p' \in \mathcal{P}} \left( \max_{p_s \in \mathcal{P}_s} \langle \nabla \mathbf{V}_{\pi,p'}, p_s - p' \rangle - \max_{p \in \mathcal{P}} \langle \nabla \mathbf{V}_{\pi,p'}, p - p' \rangle \right) \,, \tag{44}
$$

where $\mathcal{P}_s$ is the smallest $s$-rectangular uncertainty set that contains $\mathcal{P}$. In the algorithm, the supremum and maximum are equivalent since $\mathcal{P}$ and $\mathcal{P}_s$ are assumed to be compact. Using compactness and convexity of these sets, Remark 3 of Li et al. (2023) shows that $\delta_{\mathcal{P}}$ is finite using Berge's maximum theorem.

**Theorem 3** (Sample complexity for non-rectangular RCMDPs). *Let $\mathcal{P}$ be a non-rectangular uncertainty set according to Lemma 24 **a),b)** or **c)**. Further, assume the preconditions of Theorem 2 and Lemma 30. Then applying Algorithm 2 to solve the inner problem on the Lagrangian, Robust Sample-based PMD-PD obtains an average regret of*

$$\frac{1}{K}\sum_{k=1}^{K}\Phi(\pi_k) - \Phi(\pi^*) = \mathcal{O}(\epsilon + \delta_{\mathcal{P}})$$

*within $\tilde{\mathcal{O}}(1/\epsilon^5)$ samples.*

*Proof.* Let $k \in [K]$. Applying Lemma 30 to $\pi_k$, Algorithm 2 obtains a solution $p_k = \hat{p}$ such that

$$\mathbf{V}_{\pi_k, p^*}(\rho; \lambda_{k-1}) - \mathbf{V}_{\pi_k, p_k}(\rho; \lambda_{k-1}) \le M(2\epsilon'_k + \delta_{\mathcal{P}}) = \mathcal{O}(\epsilon + \delta_{\mathcal{P}})$$

within $t'_k = \tilde{\mathcal{O}}(n/\epsilon^2)$ samples, where $n$ indicates the number of samples to evaluate Eq. 43. Following the same argumentation as in Theorem 2**d)** and adding the additional term $\mathcal{O}(\delta_{\mathcal{P}})$,

$$\frac{1}{K}\sum_{k=1}^{K}\left(\Phi(\pi_k) - \Phi(\pi^*)\right) \le \frac{1}{K}\sum_{k=1}^{K}\left(\mathbf{V}_{\pi_k, p_k}(\rho; \lambda_{k-1}) + \mathcal{O}(\epsilon + \delta_{\mathcal{P}}) + \Delta_{\lambda_k} - \mathbf{V}_{\pi^*, p_k; \lambda_{k-1}}(\rho)\right) = \mathcal{O}(\epsilon + \delta_{\mathcal{P}}).$$

Since $n = \tilde{\mathcal{O}}(1/\epsilon^2)$ in Approximate TMA, substituting $\sum_{k=1}^{K} t'_k n = \tilde{\mathcal{O}}(1/\epsilon^5)$ into Eq. 42, the term dominates the sum, resulting in a $T = \tilde{\mathcal{O}}(1/\epsilon^5)$ sample complexity. $\square$

### 5.6 KL-based Mirror Descent in continuous state-action spaces

To generalise the algorithm to continuous state-action spaces, we follow a similar algorithm but replace the tabular approximation with function approximation (see Algorithm 4). A further required change is related to the weighted Bregman divergence term formulated in Eq. 16. While the weighted Bregman divergence is not a proper Bregman divergence over policies, it is over the occupancy distribution, i.e. $B_{d_\rho^{\pi, p}}(\pi, \pi') = B(d_\rho^{\pi, p}, d_\rho^{\pi', p})$, as shown in the pseudo-KL divergence in Lemma 10 of Liu et al. (2021b). To maintain similar theoretical results in continuous state-action spaces, we extend the discrete pseudo-KL divergence of occupancy to a continuous pseudo-KL divergence of occupancy.

**Definition 3.** *Continuous pseudo-KL divergence of occupancy. Define the generating function $h : \Delta^A \to \mathbb{R}$ according to*

$$h(d_\rho^{\pi, p}) = \int_{\mathcal{S} \times \mathcal{A}} d_\rho^{\pi, p}(s, a) \log(d_\rho^{\pi, p}(s, a)) \mathrm{d}s \mathrm{d}a - \int_{\mathcal{S}} d_\rho^{\pi, p}(s) \log(d_\rho^{\pi, p}(s)) \mathrm{d}s.$$

*Then its Bregman divergence is given by the continuous pseudo KL-divergence, defined as*

$$B(d_\rho^{\pi, p}, d_\rho^{\pi', p}; h) := \int_{\mathcal{S} \times \mathcal{A}} d_\rho^{\pi, p}(s, a) \log\left(\frac{d_\rho^{\pi, p}(s, a)/d_\rho^{\pi, p}(s)}{d_\rho^{\pi', p}(s, a)/d_\rho^{\pi', p}(s)}\right) \mathrm{d}s. \tag{45}$$

*Moreover, it is equivalent to a weighted Bregman divergence over policies $B_{d_\rho^{\pi, p}}(\pi, \pi')$ over continuous state-action spaces.*

*Proof.* From the definition in Eq.45 and $\nabla h(d_\rho^{\pi,p})|_{s,a} = \log(d_\rho^{\pi,p}(s,a)) - \log(d_\rho^{\pi,p}(s))$ it follows that

$$
\begin{aligned}
B(d_\rho^{\pi,p}, d_\rho^{\pi',p}; h) &= h(d_\rho^{\pi,p}) - h(d_\rho^{\pi'}) - \left\langle \nabla h(d_\rho^{\pi',p}), d_\rho^{\pi,p} - d_\rho^{\pi',p} \right\rangle \\
&= \int_{\mathcal{S} \times \mathcal{A}} d_\rho^{\pi,p}(s,a) \log(d_\rho^{\pi,p}(s,a)) \mathrm{d}s\mathrm{d}a - \int_{\mathcal{S}} d_\rho^{\pi,p}(s) \log(d_\rho^{\pi,p}(s)) \mathrm{d}s \\
&\quad - \int_{\mathcal{S} \times \mathcal{A}} d_\rho^{\pi'}(s,a) \log(d_\rho^{\pi',p}(s,a)) \mathrm{d}s\mathrm{d}a + \int_{\mathcal{S}} d_\rho^{\pi',p}(s) \log(d_\rho^{\pi',p}(s)) \mathrm{d}s \\
&\quad - \int \left( d_\rho^{\pi,p}(s,a) - d_\rho^{\pi'}(s,a) \right) \left( \log(d_\rho^{\pi',p}(s,a)) - \log(d_\rho^{\pi',p}(s)) \right) \mathrm{d}s\mathrm{d}a \\
&= \int_{\mathcal{S} \times \mathcal{A}} d_\rho^{\pi,p}(s,a) \log(d_\rho^{\pi,p}(s,a)/d_\rho^{\pi',p}(s,a)) \mathrm{d}s\mathrm{d}a - \int_{\mathcal{S}} d_\rho^{\pi,p}(s) \log(d_\rho^{\pi,p}(s)/d_\rho^{\pi',p}(s)) \mathrm{d}s \\
&= \int_{\mathcal{S} \times \mathcal{A}} d_\rho^{\pi,p}(s,a) \log \left( \frac{d_\rho^{\pi,p}(s,a)/d_\rho^{\pi,p}(s)}{d_\rho^{\pi',p}(s,a)/d_\rho^{\pi',p}(s)} \right) \mathrm{d}s\mathrm{d}a
\end{aligned}
$$

The equivalence to the weighted Bregman divergence over policies is shown by extending Eq. 16 to continuous state-action spaces and noting that $d_\rho^{\pi,p}(s,a) = d_\rho^{\pi,p}(s)\pi(a|s)$. $\qquad\square$

### 5.7 MDPO-Robust-Lagrangian: a practical implementation

To implement the algorithm in practice, we use MDPO (Tomar et al., 2022) as the policy optimiser. On-policy MDPO optimises the objective

$$
\pi_{k+1} \leftarrow \arg\max_\theta J_{\mathrm{MDPO}} := \mathbb{E}_{s \sim d_\rho^{\pi_k, p_k}(s)} \left[ \mathbb{E}_{a \sim \pi_\theta} [\hat{A}_{\pi_k, p_k}(s,a)] - \alpha D_{\mathrm{KL}}(\pi_\theta(\cdot|s), \pi_k(\cdot|s)) \right] \tag{46}
$$

based on $t_k$ SGD steps per batch $k$, which corresponds to the macro-iteration. For macro-iteration $k$, it defines the policy gradient for $\theta = \theta_k^t$ at any iteration $t \in [t_k]$ as

$$
\nabla J_{\mathrm{MDPO}}(\theta, \theta_k) = \mathbb{E}_{s \sim d_\rho^{\pi_{\theta_k}, p_k}(s)} \left[ \mathbb{E}_{a \sim \pi_{\theta_k}} \left[ \frac{\pi_\theta(a|s)}{\pi_{\theta_k}(a|s)} \nabla_\theta \log(\pi_\theta(a|s)) \hat{A}_{\pi_{\theta_k}, p_k}(s,a) \right] - \alpha \nabla_\theta D_{\mathrm{KL}}(\pi_\theta(\cdot|s), \pi_{\theta_k}(\cdot|s)) \right], \tag{47}
$$

where in our case the advantage $\hat{A}_{\pi_{\theta_k}, p_k}(s,a)$ is implemented based on the Lagrangian. The use of the importance weight $\frac{\pi_\theta}{\pi_{\theta_k}}$ is used in MDPO to correct for the deviation from mirror descent theory, where it is assumed that the data at each epoch is generated from $\pi_\theta$ rather than $\pi_{\theta_k}$. In line with our theory, our implementation performs the critic updates together with policy updates, which deviates from the original MDPO implementation. There still remains some gap of the practical implementation to our theoretical algorithm in that the state occupancy of MDPO is given by the old policy whereas in theory it is based on the current. The implementation of advantage values is based on generalised advantage estimation (Schulman et al., 2018) and the critic estimates the value and the constraint-costs based on an MLP architecture with $1 + m$ outputs. The update of the multipliers is based on the observed constraint-costs in separate samples.

To form a robust-constrained variant of MDPO, we use Lagrangian relaxation similar to the above (i.e. Algorithm 3). The algorithm considers multiple constraints and preserves the structure of the PMD-PD algorithm (Liu et al., 2021b). The LTMA algorithm implementation follows the TMA implementation from Wang et al. (2024), which applies a projected gradient descent based on clipping to the uncertainty set bounds. Due to its similarity to PPO-Lagrangian (or PPO-Lag for short) (Ray et al., 2019) and the use of updates in the robust Lagrangian, we name the algorithm **MDPO-Robust-Lagrangian** (or MDPO-Robust-Lag for short). While MDPO-Robust-Lag is based on traditionally clipping the multiplier, we also implement **MDPO-Robust-Augmented-Lag**, which uses the augmented Lagrangian according to the theory (as summarised in Algorithm 3 and 4). The practical implementation of the KL estimate follows the standard implementation in StableBaselines3, i.e. $D_{\mathrm{KL}}(\pi(\cdot|s), \pi'(\cdot|s)) \approx \mathbb{E}\left[ e^{\log(\pi(a|s)) - \log(\pi'(a|s))} - 1 - (\log(\pi(a|s)) - \log(\pi'(a|s))) \right]$.

# 6 Experiments

To demonstrate the use of MDPO for RCMDPs, we assess MDPO-Robust-Lag on three RCMDP domains. We compare MDPO-Robust-Lag to robust-constrained algorithms from related algorithm families, namely Monte Carlo based mirror descent (**RMCPMD** (Wang et al., 2024) in particular) and proximal policy optimisation with function approximation (**PPO-Lag** (Achiam et al., 2017) in particular). All the involved algorithm families (MDPO, MCPMD, and PPO) are included with four variants, namely with and without Lagrangian (indicated by the -Lag suffix) and with and without robustness training (indicated by the R- prefix or the -Robust- infix). Additionally, the MDPO-Lag implementations also have Augmented-Lag variants which use the augmented Lagrangian as proposed in the theory section. The implementation details of the algorithms included in the study can be found in Appendix E. The code for all experiments can be found at `https://github.com/bossdm/MDPO-RCMDP`.

After training with the above-mentioned algorithms, the agents are subjected to a test which presents transition dynamics based on the distortion levels setup in Wang et al. (2024). The test procedure follows the same general procedure for all domains. For distortion level $x \in [0, 1]$, the agent is subjected to a perturbation such that if the nominal model is $\xi_c$ and the uncertainty set varies the parameter in $[\underline{\xi}, \overline{\xi}]$, the transition dynamics kernel parameter of the test with distortion level $x$ is given for each dimension by either $\xi_i = \xi_c + x^2(\overline{\xi} - \xi_c)$ or $\xi_i = \xi_c + x^2(\underline{\xi} - \xi_c)$. We slightly improve the test evaluation by considering all the $\pm$ directions rather than just a single one, amounting to a total number of test evaluations $N = n_{\text{test}} \times 2^n$, where $n$ is the dimensionality of the uncertainty set parametrisation and $n_{\text{test}}$ is the number of episodes per perturbation.

To summarise the results, we use the penalised return, a popular summary statistic for robust constrained RL (Mankowitz et al., 2020; Bossens, 2024). The statistic is defined for a maximisation problem as $R_{\text{pen}} = V(\rho) - \lambda_{\text{max}} \sum_{j=1}^{m} \max(0, C_j(\rho))$ where $V(\rho)$ denotes the value (negative of the cost) from the starting distribution and $C_j(\rho)$ denotes the $j$'th constraint-cost from the starting distribution. In addition, we also formulate the *signed penalised return* $R_{\text{pen}}^{\pm} = V(\rho) - \sum_{j=1}^{m} \lambda_{\text{max}} C_j(\rho)$. We note that the penalised return matches the objective of the constrained methods and the return matches the objective of the unconstrained methods. The signed penalised return might be especially useful for robust problems; as even our extensive testing procedure may not find the worst case in the uncertainty set, and it is also of interest to generalise even beyond the uncertainty set, a solution that has negative cost can be evaluated more positively since it is more likely not to have a positive cost in the worst-case transition kernel. The summary statistics reported included both the mean and the minimum since the minimum indicates the effectiveness in dealing with the minimax problem (the worst-case robustness). However, as a note of caution, the minimum is challenging to establish accurately because even though we test many environments it may still not contain the worst-case environment.

We report the results based on two distinct sample budgets, one with a limited number of time steps (200–500 times the maximal number of time steps in the episode) and one with a large number of time steps (2000–5000 times the maximal number of time steps in the episode). This setup allows to assess the sample-efficiency and convergence properties more clearly.

An additional set of experiments is to establish which kind of schedule for $\alpha_k$ works best for MDPO algorithms, and the robust(-constrained) variants in particular (see Apppendix H). From this set of experiments, it is confirmed that a fixed setting works reasonably well across problems. The best overall performance for robust-constrained RL is obtained by MDPO-Robust-Augmented-Lag using a batch-based linear schedule with restarts, and a geometric schedule achieves high performance with MDPO-Robust-Lag. For simplicity, the fixed schedule is used for all MDPO based algorithms throughout the remaining experiments.

## 6.1 Cartpole

To assess our algorithm on an RCMDP, we introduce the robust constrained variant of the well-known Cartpole problem (Barto et al., 1983; Brockman et al., 2016) by modifying the RMDP from Wang et al. (2024). The problem involves a mechanical simulation of a frictionless cart moving back and forth to maintain a pole, which is attached to the cart as an un-actuated joint, in the upright position. The agent observes

$x, \dot{x}, \theta, \dot{\theta}$ and takes actions in $\{\text{left}, \text{right}\}$, which apply a small force moving either left or right. The agent receives a reward of $+1$ until either the cart goes out of the $[-2.4, 2.4]$m bounds or the pole has an angle outside the range $[-12°, 12°]$. In contrast to the traditional problem, the mechanics are not deterministic, and following Wang et al. (2024), transitions dynamics models are multi-variate Gaussians of the form

$$p(s'|s,a) = \frac{1}{(2\pi)^2 |\Sigma|^{1/2}} e^{-\frac{1}{2}(s'-\mu_c(s,a))^\intercal \Sigma^{-1}(s'-\mu_c(s,a))} , \tag{48}$$

where $\mu_c(s,a)$ is the deterministic next state given $(s,a) \in \mathcal{S} \times \mathcal{A}$ based on the mechanics of the original Cartpole problem. We first train and evaluate algorithms in a CMDP problem for their ability to adhere to safety constraints. We then also show the benefits of robustness training and evaluate algorithms in an RCMDP problem.

To formulate safety constraints into the above Cartpole problem, we define the instantanous constraint-cost as $c_t = |\dot{x}_t| - d$, where $d$ is a constant, set to $d = 0.15$ in the experiments, and $\dot{x}$ is the velocity of the cart. The safety constraint of the agent is to maintain constraint $C = \mathbb{E}[\sum_t \gamma^t c_t] \leq 0$.

To incorporate robustness into our setting, we introduce delta-perturbations as in Wang et al. (2024), using an uncertainty set with parametrisation

$$p(s'|s,a) = \frac{1}{(2\pi)^{n/2} |\Sigma|^{1/2}} e^{-\frac{1}{2}(s'-(1+\delta)\mu_c(s,a))^\intercal \Sigma^{-1}(s'-(1+\delta)\mu_c(s,a))} , \tag{49}$$

where $\delta \in [-\kappa_i, \kappa_i]_{i=1}^n$. Our parameter settings differ in that we use a larger uncertainty set, with $\kappa_i$ being increased fivefold (making the problem more challenging) while the number of time steps is reduced to make the constraint satisfiable. The robust algorithms are trained on this uncertainty set, where the transition dynamics are adjusted based on the transition mirror ascent algorithm of Wang et al. (2024), i.e. by using Monte Carlo and mirror ascent. In the experiment, this amounts to a projected update, where the projection restricts the update to lie within the uncertainty set.

**Training performance** The training performance of non-robust algorithms can be found in Appendix F.1 (Figure 7). The training performance of robust methods, which use LTMA, can be found in Figure 8. A trade-off in reward vs constraint-cost can be observed; that is, the constrained algorithms put emphasis on reducing the constraint-cost while the unconstrained only maximise the reward. The plots demonstrate the rapid convergence of the MDPO-based algorithms.

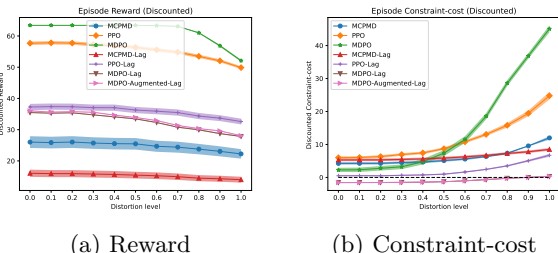

(a) Reward  (b) Constraint-cost

Figure 1: Test performance of MDP and CMDP algorithms obtained by applying the learned deterministic policy from the Cartpole domain after 20,000 time steps of training. The line and shaded area represent the mean and standard error across the perturbations for the particular distortion level.

**Test performance** Figure 1 and 2 show the test performance after 20,000 time steps depending on the distortion level. MDPO-Robust and PPO achieve the highest performance on the reward. However, since they do not optimise the Lagrangian, they perform poorly on the constraint-cost. MDPO-Robust-Augmented-Lag achieves the best overall performance by providing the lowest constraint-cost by far across all distortion levels. In the test performance after 200,000 time steps (see Figure 13 and 14 of Appendix G.1), the MCPMD algorithms are able to get to similar performance levels, and all three baselearners are competitive.

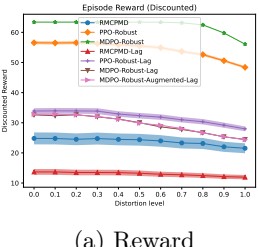
(a) Reward

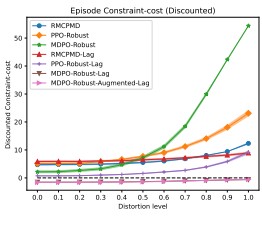
(b) Constraint-cost

Figure 2: Test performance of RMDP and RCMDP algorithms obtained by applying the learned determinstic policy from the Cartpole domain after 20,000 time steps of training. The line and shaded area represent the mean and standard error across the perturbations for the particular distortion level.

To summarise the overall performance quantitatively, Table 2a shows that MDPO-Robust is the top performer on the return after 20,000 time steps. MDPO-Robust-Lag and MDPO-Lag algorithms obtain similar levels of performance on the mean penalised return statistics, although the former obtain an improved minimum score, and both are far ahead of PPO-Lag and MCPMD-Lag algorithms. In Table 2b, the performance after 200,000 time steps can be observed. MDPO-Robust-Lag, with or without augmentation, is the top performer on the mean penalised return metrics, followed by non-robust MDPO-Lag. MCPMD-Lag and RMCPMD-Lag perform highest on the minimum performance, and are followed by MDPO-Robust-Lag algorithms. In summary, the solutions converge to relatively similar test performance levels but MDPO-based algorithms are superior in terms of sample efficiency.

Table 2: Test performance on the return and penalised return statistics in the Cartpole domain based on 11 distortion levels, 16 perturbations per level, 10 seeds, and 100 evaluations per test. The mean score is the grand average over distortion levels, perturbations, seeds, and evaluations. The standard error and minimum are computed across the different environments (i.e. distortion levels and perturbations), indicating the robustness of the solution to changes in the environment. Bold indicates the top performance and any additional algorithms that are within one pooled standard error.

(a) After 20,000 time steps of training

| | Return | | $R_{\mathrm{pen}}^{\pm}$ (signed) | | $R_{\mathrm{pen}}$ (positive) | |
|---|---|---|---|---|---|---|
| | Mean ± SE | Min | Mean ± SE | Min | Mean ± SE | Min |
| MCPMD | $26.7_{\pm 0.1}$ | 23.3 | $-196.6_{\pm 22.3}$ | -523.9 | $-209.6_{\pm 20.9}$ | -525.9 |
| PPO | $58.1_{\pm 0.1}$ | 51.3 | $-262.1_{\pm 76.8}$ | -1121.3 | $-290.4_{\pm 72.7}$ | -1122.3 |
| MDPO | $62.9_{\pm 0.1}$ | 52.4 | $-210.6_{\pm 152.3}$ | -2015.7 | $-270.1_{\pm 144.3}$ | -2016.0 |
| RMCPMD | $25.4_{\pm 0.1}$ | 22.3 | $-216.8_{\pm 22.0}$ | -538.5 | $-228.2_{\pm 20.4}$ | -539.0 |
| PPO-Robust | $57.1_{\pm 0.2}$ | 50.3 | $-225.3_{\pm 70.5}$ | -985.3 | $-250.8_{\pm 66.9}$ | -987.0 |
| MDPO-Robust | $\mathbf{63.1}_{\pm 0.1}$ | **56.4** | $-229.7_{\pm 179.6}$ | -2610.6 | $-294.1_{\pm 171.4}$ | -2610.8 |
| MCPMD-Lag | $16.6_{\pm 0.1}$ | 14.0 | $-247.3_{\pm 11.2}$ | -383.7 | $-259.3_{\pm 10.9}$ | -392.5 |
| PPO-Lag | $39.1_{\pm 0.1}$ | 33.9 | $2.4_{\pm 17.2}$ | -217.4 | $-23.6_{\pm 15.1}$ | -229.3 |
| MDPO-Lag | $37.0_{\pm 0.2}$ | 28.0 | $\mathbf{106.1}_{\pm 6.9}$ | 23.2 | $\mathbf{35.7}_{\pm 1.6}$ | 11.9 |
| MDPO-Augmented-Lag | $37.5_{\pm 0.2}$ | 28.3 | $105.6_{\pm 7.2}$ | 18.4 | $35.9_{\pm 1.7}$ | 7.2 |
| RMCPMD-Lag | $14.2_{\pm 0.1}$ | 12.0 | $-276.7_{\pm 10.8}$ | -408.6 | $-287.7_{\pm 10.9}$ | -419.5 |
| PPO-Robust-Lag | $35.5_{\pm 0.2}$ | 28.8 | $-12.2_{\pm 22.0}$ | -376.9 | $-39.4_{\pm 19.8}$ | -387.5 |
| MDPO-Robust-Lag | $34.4_{\pm 0.3}$ | 24.9 | $108.4_{\pm 4.4}$ | 65.9 | $34.2_{\pm 1.2}$ | **24.9** |
| MDPO-Robust-Augmented-Lag | $34.6_{\pm 0.3}$ | 24.7 | $\mathbf{110.8}_{\pm 4.9}$ | **61.6** | $\mathbf{34.6}_{\pm 1.2}$ | **24.7** |

(b) After 200,000 time steps of training

| | Return | | $R_{\mathrm{pen}}^{\pm}$ (signed) | | $R_{\mathrm{pen}}$ (positive) | |
|---|---|---|---|---|---|---|
| | Mean ± SE | Min | Mean ± SE | Min | Mean ± SE | Min |
| MCPMD | $63.0_{\pm 0.1}$ | 54.1 | $-316.3_{\pm 167.8}$ | -2372.3 | $-364.0_{\pm 161.2}$ | -2372.5 |
| PPO | $\mathbf{63.1}_{\pm 0.1}$ | **56.4** | $-252.2_{\pm 178.6}$ | -2474.0 | $-307.2_{\pm 171.1}$ | -2474.2 |
| MDPO | $62.7_{\pm 0.2}$ | 46.3 | $-96.2_{\pm 120.7}$ | -1594.5 | $-159.5_{\pm 112.4}$ | -1594.5 |
| RMCPMD | $\mathbf{63.1}_{\pm 0.1}$ | **56.5** | $-175.2_{\pm 181.9}$ | -2672.5 | $-241.0_{\pm 173.6}$ | -2672.9 |
| PPO-Robust | $62.8_{\pm 0.1}$ | 55.2 | $-191.0_{\pm 148.9}$ | -2036.3 | $-251.0_{\pm 140.9}$ | -2037.4 |
| MDPO-Robust | $\mathbf{63.2}_{\pm 0.1}$ | 56.1 | $-109.7_{\pm 176.0}$ | -2647.1 | $-185.5_{\pm 167.1}$ | -2648.3 |
| MCPMD-Lag | $29.7_{\pm 0.3}$ | 21.6 | $97.4_{\pm 3.9}$ | **72.0** | $29.7_{\pm 1.2}$ | **21.6** |
| PPO-Lag | $41.0_{\pm 0.2}$ | 33.8 | $78.4_{\pm 10.3}$ | -55.5 | $32.0_{\pm 5.8}$ | -58.2 |
| MDPO-Lag | $44.5_{\pm 0.2}$ | 37.8 | $90.9_{\pm 11.7}$ | -38.9 | $\mathbf{36.0}_{\pm 5.5}$ | -42.4 |
| MDPO-Augmented-Lag | $45.2_{\pm 0.2}$ | 38.4 | $86.0_{\pm 12.5}$ | -54.4 | $\mathbf{35.1}_{\pm 6.6}$ | -56.3 |
| RMCPMD-Lag | $29.4_{\pm 0.3}$ | 22.0 | $89.4_{\pm 3.7}$ | **70.8** | $29.1_{\pm 1.1}$ | **21.9** |
| PPO-Robust-Lag | $39.4_{\pm 0.2}$ | 32.1 | $84.2_{\pm 9.0}$ | -28.9 | $32.1_{\pm 4.5}$ | -35.6 |
| MDPO-Robust-Lag | $41.6_{\pm 0.2}$ | 32.1 | $105.1_{\pm 9.5}$ | 9.6 | $\mathbf{38.6}_{\pm 2.5}$ | 2.6 |
| MDPO-Robust-Augmented-Lag | $41.2_{\pm 0.2}$ | 31.4 | $\mathbf{106.2}_{\pm 9.1}$ | 13.7 | $\mathbf{38.6}_{\pm 2.2}$ | 5.0 |

## 6.2 Inventory Management

A second domain is the Inventory Management domain from Wang et al. (2024), which involves maintaining an inventory of resources. The resources induce a period-wise cost and the goal is to purchase and sell resources such that minimal cost is incurred over time. To form a constrained variant of this benchmark, we introduce the constraint that the discounted sum of actions should not average to higher than zero (i.e. the rate of selling should not exceed that of purchasing). We use the same radial features and clipped uncertainty set as in the implementation on github `https://github.com/JerrisonWang/JMLR-DRPMD`. The implementation uses Gaussian parametric transition dynamics

$$p(s'|s,a) = \frac{1}{(2\pi)^{1/2}\sigma} e^{-\frac{1}{2\sigma}(s'-\eta(s,a)^{\intercal}\zeta(s,a))^2}, \tag{50}$$

with $\sigma = 1$, and the feature vector for $i = 1, 2$ is given by

$$\zeta_i(s, a) = e^{-\frac{\left\|s - \mu_{\zeta_i, s}\right\|^2 + \left\|a - \mu_{\zeta_i, a}\right\|^2}{2\sigma_{\zeta_i}^2}}, \tag{51}$$

where $\mu_{\zeta_1} = (-4, 5)$, $\mu_{\zeta_2} = (-2, 8)$. The uncertainty set is given by $\{\eta : \|\eta - \eta_c\|_\infty \leq \kappa\}$ where $\eta_c = (-2, 3.5)$. With the exception of the constraint, and the number of steps per episode for the adversary, the setting matches Wang et al. (2024). The constraint at time $t$ is given by

$$c(s_t, a_t) = K_{s,a}(a_t^2 - s_t) - d, \tag{52}$$

with constants set as $K_{s,a} = 0.01$ and $d = 0.0$ in the experiments. The constraint indicates a preference for actions with modest squared transaction magnitudes and a preference for having a positive inventory.

**Training performance**   The training performance can be found in Figure 9 of Appendix F.2 for non-robust algorithms and in Figure 10 of Appendix F.2 for robust algorithms. The plots confirm the effectiveness of the constrained optimisation algorithms and the robust algorithms can also find a good solution despite the environment being adversarial. It is also clear that the MDPO based algorithms converge more rapidly to solutions of the highest quality in constrained and robust-constrained optimisation, although there appears to be a small benefit of PPO in non-robust unconstrained optimisation.

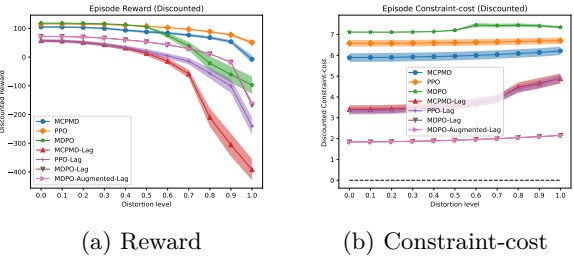

(a) Reward                  (b) Constraint-cost

Figure 3: Test performance of MDP and CMDP algorithms in the Inventory Management domain using the deterministic policy on the test distortions.

**Test performance**   Figure 5 and Figure 6 visualise the results for the test performance after 16,000 time steps. MDPO-Robust obtains the highest performance on the test reward, followed closely by MDPO and PPO. The constraint is challenging to satisfy for all algorithms, but MDPO-Lag based algorithms obtain the best solutions with cost between 2 and 3 across the distortion levels. Algorithms based on PPO-Lag and MCPMD-Lag perform poorly with worst starting points and stronger performance degrading across distortion levels. After 400,000 time steps, the differences between the algorithms is relatively small in the test cases (see Figure 15 and  16 of Appendix G.2)

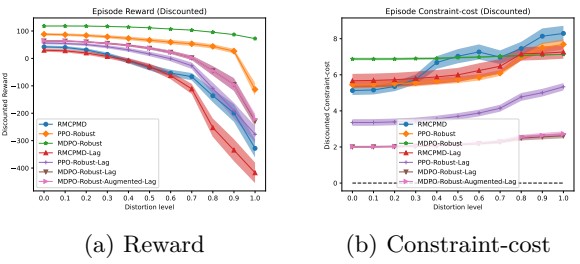

(a) Reward                  (b) Constraint-cost

Figure 4: Test performance of RMDP and RCMDP algorithms in the Inventory Management domain using the deterministic policy on the test distortions.

As can be observed in Table 3a, MDPO-Robust is the top performer on the return after 16,000 time steps. All other statistics, including the mean and minimum of the signed and positive penalised return are maximised by

MDPO-Lag based algorithms. In Table 3b, the performance after the full 400,000 time steps can be observed. With the exception of PPO-Robust-Lag, all constrained algorithms converge to a similar optimum, indicating that the nominal environment and the other environments in the uncertainty set have large overlap. Similarly, the unconstrained algorithms reach a similar optimum. In summary, the solutions converge to relatively similar test performance levels but MDPO-based algorithms are superior in terms of sample efficiency.

Table 3: Test performance on the return and penalised return statistics in the Inventory Management domain based on 11 distortion levels, 4 perturbations per level, 10 seeds, and 50 evaluations per test. The mean score is the grand average over distortion levels, perturbations, seeds, and evaluations. The standard error and minimum are computed across the different environments (i.e. distortion levels and perturbations), indicating the robustness of the solution to changes in the environment. Bold indicates the top performance and any additional algorithms that are within one pooled standard error.

(a) After 16,000 time steps of training

| | Return | | $R_{\text{pen}}^{\pm}$ (signed) | | $R_{\text{pen}}$ (positive) | |
|---|---|---|---|---|---|---|
| | Mean ± SE | Min | Mean ± SE | Min | Mean ± SE | Min |
| MCPMD | 103.6 ± 4.0 | -7.5 | -2838.3 ± 12.7 | -2946.2 | -2838.4 ± 12.6 | -2946.2 |
| PPO | 114.7 ± 2.3 | 50.7 | -3166.9 ± 6.5 | -3208.0 | -3166.9 ± 6.5 | -3208.0 |
| MDPO | 102.6 ± 8.0 | -97.4 | -3430.3 ± 21.9 | -3557.6 | -3430.3 ± 21.9 | -3557.6 |
| RMCPMD | 63.6 ± 23.1 | -328.0 | -2646.9 ± 124.6 | -3548.1 | -2648.0 ± 123.9 | -3548.1 |
| PPO-Robust | 100.7 ± 9.9 | -113.0 | -2673.0 ± 74.2 | -3397.5 | -2674.4 ± 73.4 | -3397.5 |
| MDPO-Robust | **118.0** ± 1.8 | **72.3** | -3306.4 ± 9.7 | -3350.0 | -3306.4 ± 9.7 | -3350.0 |
| MCPMD-Lag | 50.1 ± 20.2 | -392.3 | -1684.0 ± 35.8 | -2003.1 | -1686.4 ± 36.0 | -2003.1 |
| PPO-Lag | 64.7 ± 13.9 | -240.1 | -1593.3 ± 43.0 | -1878.9 | -1597.2 ± 39.5 | -1878.9 |
| MDPO-Lag | 66.8 ± 7.1 | -167.2 | **-837.3** ± 21.4 | **-939.0** | -837.9 ± 20.4 | **-939.0** |
| MDPO-Augmented-Lag | 66.9 ± 7.0 | -159.4 | **-837.5** ± 21.1 | **-938.9** | **-838.0** ± 20.4 | **-938.9** |
| RMCPMD-Lag | 50.3 ± 27.4 | -417.4 | -2635.6 ± 113.0 | -3159.5 | -2638.3 ± 111.1 | -3159.5 |
| PPO-Robust-Lag | 63.2 ± 17.1 | -277.0 | -1655.0 ± 67.5 | -2190.2 | -1660.0 ± 65.2 | -2190.2 |
| MDPO-Robust-Lag | 60.7 ± 10.7 | -228.5 | -930.9 ± 32.3 | -1118.2 | -932.2 ± 30.8 | -1118.2 |
| MDPO-Robust-Augmented-Lag | 60.9 ± 10.6 | -221.4 | -938.6 ± 34.4 | -1172.2 | -940.1 ± 32.9 | -1172.2 |

(b) After 400,000 time steps of training

| | Return | | $R_{\text{pen}}^{\pm}$ (signed) | | $R_{\text{pen}}$ (positive) | |
|---|---|---|---|---|---|---|
| | Mean ± SE | Min | Mean ± SE | Min | Mean ± SE | Min |
| MCPMD | **119.9** ± 1.8 | 72.8 | -3425.3 ± 5.5 | -3460.1 | -3425.3 ± 5.5 | -3460.1 |
| PPO | **120.0** ± 1.8 | **73.6** | -3425.4 ± 5.4 | -3459.7 | -3425.4 ± 5.4 | -3459.7 |
| MDPO | 118.8 ± 2.5 | 32.2 | -3417.0 ± 8.8 | -3459.7 | -3417.0 ± 8.8 | -3459.7 |
| RMCPMD | 109.3 ± 2.8 | 26.0 | -2907.7 ± 8.5 | -2956.1 | -2907.9 ± 8.3 | -2956.1 |
| PPO-Robust | **119.9** ± 1.8 | **73.2** | -3425.3 ± 5.4 | -3460.3 | -3425.3 ± 5.4 | -3460.3 |
| MDPO-Robust | **119.7** ± 1.8 | 72.4 | -3416.7 ± 4.1 | -3429.7 | -3416.7 ± 4.1 | -3429.7 |
| MCPMD-Lag | 67.1 ± 7.0 | -156.5 | **-837.7** ± 20.9 | **-939.0** | -838.2 ± 20.2 | **-939.0** |
| PPO-Lag | 66.7 ± 7.1 | -166.7 | **-837.2** ± 21.4 | **-939.0** | **-837.8** ± 20.5 | **-939.0** |
| MDPO-Lag | 66.8 ± 7.2 | -168.5 | **-837.3** ± 21.5 | **-939.3** | -838.0 ± 20.4 | **-939.3** |
| MDPO-Augmented-Lag | 66.9 ± 7.1 | -162.5 | -837.5 ± 21.2 | **-939.2** | -838.0 ± 20.4 | **-939.2** |
| RMCPMD-Lag | 66.8 ± 7.1 | -165.8 | **-837.4** ± 21.4 | **-939.0** | -838.1 ± 20.3 | **-939.0** |
| PPO-Robust-Lag | 60.8 ± 10.8 | -234.0 | -931.0 ± 32.5 | -1118.3 | -932.4 ± 30.9 | -1118.3 |
| MDPO-Robust-Lag | 66.9 ± 7.1 | -166.5 | **-837.4** ± 21.4 | **-939.3** | -838.0 ± 20.4 | **-939.3** |
| MDPO-Robust-Augmented-Lag | 66.8 ± 7.1 | -167.7 | **-837.2** ± 21.5 | **-939.1** | -837.9 ± 20.5 | **-939.1** |

## 6.3 3-D Inventory Management

The third (and last) domain in the experiments introduces a multi-dimensional variant of the above Inventory Management problem, with dynamics

$$p(s'|s,a) = \frac{1}{(2\pi)^{n/2}\sigma} e^{-\frac{1}{2\sigma}(s'-\eta(s,a)^\intercal \zeta(s,a))^2} , \qquad (53)$$

where the feature vectors of Eq. 51 are now given by $\mu_{\zeta_1} = (-3, -2.5, -3.5, 5.0, 2.0, 4.5)$, $\mu_{\zeta_2} = (-6.0, -2.8, -4.0, 8.0, 2.0, 2.5)$, $\sigma_\zeta = (4, 4.5)$, and the uncertainty set is given by $\{\eta : \|\eta - \eta_c\|_\infty \leq \kappa\}$ where $\eta_c = [[-2, 2.5], [-1.8, 1.5], [-1.5, 2.0]]$ and $\kappa = 0.5$. The constraint $j \in [m]$ at time $t$ is given by

$$c_j(s_t, a_t) = a_{t,j} - K_s s_{t,j} - d , \qquad (54)$$

where $a_{t,j}$ and $s_{t,j}$ denote the $j$'th dimension of the state and action, respectively, at time $t$, and constants are set as $K_s = 0.5$ and $d = 0.0$ in the experiments. The constraint indicates that the agent should not get a negative inventory and if the inventory is positive, it should not sell more than 50% of the current inventory.

**Training performance** The training performance can be found in Figure 11 and Figure 12 of Appendix F.3. It is clear that the MDPO-based algorithm have improved sample-efficiency and converge to the highest unconstrained and constrained performances.

**Test performance** Figure 5 and Figure 6 visualise the results for the test performance after 40,000 time steps. While all algorithms are sensitive to the distortion level, it can be seen that the robust algorithms have are generally shifted up for the rewards, and robust-constrained algorithms have been shifted down for the constraint-costs, indicating the effectiveness of robustness training. The MDPO-Robust algorithm is superior in the return and the the MDPO-Robust-Lag algorithms are found to be superior in the constraint-cost. While it is challenging to satisfy the constraint with limited training, after 400,000 time all algorithms have further improved, but remarkably MDPO-Robust-Augmented-Lag can satisfy all the constraints for all but the highest distortion levels (see Figure 17 and Figure 18 of Appendix G.3).

The return and penalised return statistics after 40,000 time steps can be observed in Table 4a. MDPO-Robust has the highest score on the mean and minimum of the test return, which indicates the robustness

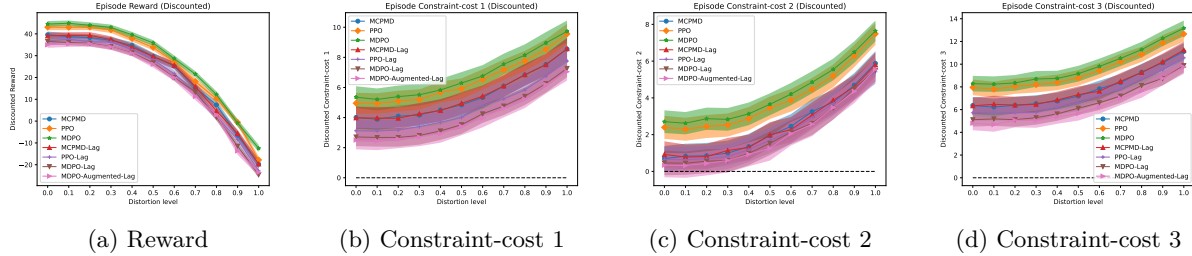

| (a) Reward | (b) Constraint-cost 1 | (c) Constraint-cost 2 | (d) Constraint-cost 3 |

Figure 5: Test performance of MDP and CMDP algorithms in the 3-D Inventory Management domain using the deterministic policy on the test distortions.

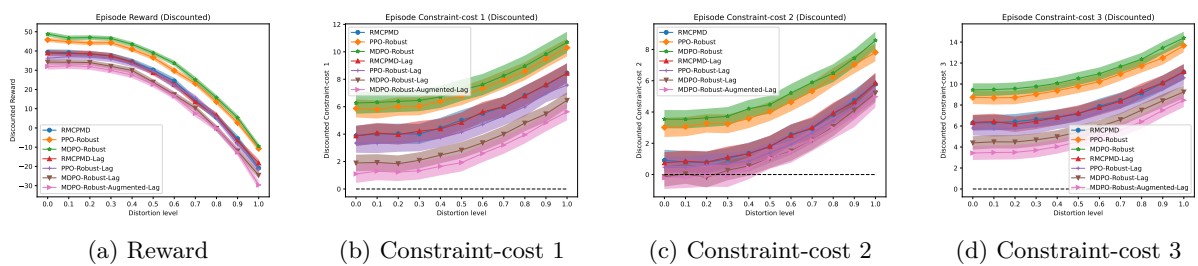

| (a) Reward | (b) Constraint-cost 1 | (c) Constraint-cost 2 | (d) Constraint-cost 3 |

Figure 6: Test performance of RMDP and RCMDP algorithms in the 3-D Inventory Management domain using the deterministic policy on the test distortions.

Table 4: Test performance on the return and penalised return statistics in the 3-D Inventory Management domain based on 11 distortion levels, 64 perturbations per level, 10 seeds, and 10 evaluations per test. The standard error and minimum are computed across the different environments (i.e. distortion levels and perturbations), indicating the robustness of the solution to changes in the environment. Bold indicates the top performance and any additional algorithms that are within one pooled standard error.

### (a) After 40,000 time steps

| | Return | | $R_{\mathrm{pen}}^{\pm}$ (signed) | | $R_{\mathrm{pen}}$ (positive) | |
|---|---|---|---|---|---|---|
| | Mean ± SE | Min | Mean ± SE | Min | Mean ± SE | Min |
| MCPMD | $46.2_{\pm 0.5}$ | -19.7 | $-3810.9_{\pm 448.7}$ | -10322.4 | $-7143.7_{\pm 250.0}$ | -11158.3 |
| PPO | $50.6_{\pm 0.4}$ | -17.8 | $-5909.8_{\pm 458.0}$ | -12737.1 | $-7957.4_{\pm 286.0}$ | -12855.0 |
| MDPO | $51.6_{\pm 0.4}$ | -12.5 | $-6415.1_{\pm 463.7}$ | -13210.2 | $-8389.6_{\pm 300.0}$ | -13384.9 |
| RMCPMD | $46.3_{\pm 0.5}$ | -20.8 | $-3817.4_{\pm 449.2}$ | -10449.0 | $-7146.4_{\pm 253.1}$ | -11228.8 |
| PPO-Robust | $53.2_{\pm 0.4}$ | -10.7 | $-7164.3_{\pm 453.6}$ | -13926.4 | $-8652.0_{\pm 316.2}$ | -14027.1 |
| MDPO-Robust | $\mathbf{54.9}_{\pm 0.4}$ | **-9.6** | $-7956.9_{\pm 457.5}$ | -14747.8 | $-9376.8_{\pm 330.2}$ | -14836.3 |
| MCPMD-Lag | $46.3_{\pm 0.5}$ | -19.6 | $-3822.8_{\pm 452.0}$ | -10390.2 | $-7149.5_{\pm 253.6}$ | -11185.8 |
| PPO-Lag | $44.7_{\pm 0.5}$ | -22.8 | $-3038.1_{\pm 448.7}$ | -9467.8 | $-6515.6_{\pm 238.5}$ | -10277.8 |
| MDPO-Lag | $43.4_{\pm 0.5}$ | -24.6 | $-2391.4_{\pm 447.4}$ | -8762.9 | $-5530.5_{\pm 237.8}$ | -9481.2 |
| MDPO-Augmented-Lag | $42.8_{\pm 0.5}$ | -23.5 | $-2126.5_{\pm 446.1}$ | -8385.7 | $-5460.9_{\pm 229.5}$ | -9142.2 |
| RMCPMD-Lag | $46.2_{\pm 0.5}$ | -18.2 | $-3811.7_{\pm 451.5}$ | -10399.7 | $-7144.5_{\pm 253.5}$ | -11239.9 |
| PPO-Robust-Lag | $44.7_{\pm 0.5}$ | -21.5 | $-3014.6_{\pm 445.4}$ | -9428.9 | $-6494.1_{\pm 238.7}$ | -10268.1 |
| MDPO-Robust-Lag | $41.4_{\pm 0.5}$ | -24.6 | $-1416.9_{\pm 446.1}$ | -7726.9 | $\mathbf{-5088.9}_{\pm 221.7}$ | -8560.6 |
| MDPO-Robust-Augmented-Lag | $39.5_{\pm 0.5}$ | -29.6 | $\mathbf{-525.0}_{\pm 441.2}$ | **-6861.9** | $\mathbf{-4957.0}_{\pm 211.2}$ | **-8289.3** |

### (b) After 400,000 time steps

| | Return | | $R_{\mathrm{pen}}^{\pm}$ (signed) | | $R_{\mathrm{pen}}$ (positive) | |
|---|---|---|---|---|---|---|
| | Mean ± SE | Min | Mean ± SE | Min | Mean ± SE | Min |
| MCPMD | $46.2_{\pm 0.5}$ | -21.3 | $-3810.9_{\pm 448.9}$ | -10358.5 | $-7147.0_{\pm 251.9}$ | -11105.7 |
| PPO | $59.5_{\pm 0.4}$ | -5.9 | $-10122.1_{\pm 462.4}$ | -16777.6 | $-10886.2_{\pm 370.4}$ | -16823.6 |
| MDPO | $81.2_{\pm 0.4}$ | 18.6 | $-20577.2_{\pm 535.4}$ | -28669.4 | $-20769.1_{\pm 504.0}$ | -28670.9 |
| RMCPMD | $46.3_{\pm 0.5}$ | -18.9 | $-3817.2_{\pm 450.6}$ | -10514.2 | $-7148.9_{\pm 252.9}$ | -11353.5 |
| PPO-Robust | $60.9_{\pm 0.4}$ | -2.4 | $-10802.7_{\pm 443.3}$ | -17484.1 | $-11429.2_{\pm 367.4}$ | -17506.1 |
| MDPO-Robust | $\mathbf{87.0}_{\pm 0.4}$ | **24.4** | $-23312.6_{\pm 525.6}$ | -31314.1 | $-23459.7_{\pm 499.3}$ | -31314.1 |
| MCPMD-Lag | $46.2_{\pm 0.5}$ | -22.0 | $-3813.8_{\pm 453.1}$ | -10509.0 | $-7147.1_{\pm 254.3}$ | -11366.5 |
| PPO-Lag | $34.2_{\pm 0.5}$ | -36.5 | $1965.9_{\pm 416.5}$ | -3963.9 | $-3126.5_{\pm 214.3}$ | -7007.9 |
| MDPO-Lag | $34.9_{\pm 0.5}$ | -32.5 | $1566.5_{\pm 420.9}$ | -4406.7 | $\mathbf{-1180.5}_{\pm 229.6}$ | -5420.4 |
| MDPO-Augmented-Lag | $24.6_{\pm 0.5}$ | -44.1 | $\mathbf{6421.1}_{\pm 369.2}$ | **1517.0** | $\mathbf{-1139.5}_{\pm 175.7}$ | -4564.8 |
| RMCPMD-Lag | $46.2_{\pm 0.5}$ | -20.8 | $-3813.1_{\pm 451.5}$ | -10594.7 | $-7146.9_{\pm 252.7}$ | -11310.6 |
| PPO-Robust-Lag | $31.0_{\pm 0.5}$ | -34.8 | $3460.0_{\pm 419.6}$ | -2398.7 | $-2285.9_{\pm 167.0}$ | -5397.1 |
| MDPO-Robust-Lag | $32.8_{\pm 0.4}$ | -32.4 | $2617.9_{\pm 378.6}$ | -2601.7 | $\mathbf{-1011.4}_{\pm 169.9}$ | **-3767.4** |
| MDPO-Robust-Augmented-Lag | $26.8_{\pm 0.4}$ | -38.6 | $5312.6_{\pm 340.2}$ | 758.7 | $-1219.2_{\pm 150.4}$ | **-3867.7** |

training was succesful in guaranteeing high levels of performance across the uncertainty set. MDPO-Robust-Augmented-Lag outperforms all other algorithms on the signed and positive penalised return statistics. MDPO-Robust-Lag follows closely in mean and minimum performance on the positive penalised return. After 400,000 time steps, MDPO-Robust-Lag algorithms have by far the highest minimum penalised return while MDPO algorithms with augmented Lagrangian score remarkably well on the signed penalised return (see Table 4b). Overall, the data indicate the effectiveness for MDPO-based algorithms, and particularly the effectiveness of MDPO-Robust-Lag algorithms for robust constrained optimisation.

## 7 Conclusion

This paper presents mirror descent policy optimisation for RCMDPs, making use of policy gradient techniques to optimise both the policy and the transition kernel (as an adversary) on the Lagrangian representing a CMDP. In the sample-based rectangular RCMDP setting, we require $\tilde{\mathcal{O}}(\epsilon^{-3})$ samples for an average regret of at most $\epsilon$, confirming an $\tilde{\mathcal{O}}\left(1/T^{1/3}\right)$ convergence rate. Our analysis also covers the case of non-rectangular RCMDPs, where our algorithm provides an $\tilde{\mathcal{O}}\left(1/T^{1/5}\right)$ sample-based rate. An approximate algorithm for transition mirror ascent is also proposed which is of independent interest for designing adversarial environments in general MDPs. Experiments confirm the performance benefits of mirror descent policy optimisation in practice, obtaining significant improvements in test performance.

**Acknowledgments**

This research/project is supported by the National Research Foundation, Singapore under its National Large Language Models Funding Initiative (AISG Award No: AISG-NMLP-2024-003). Any opinions, findings and conclusions or recommendations expressed in this material are those of the author(s) and do not reflect the views of National Research Foundation, Singapore. This research is supported by the National Research Foundation, Singapore and the Ministry of Digital Development and Information under the AI Visiting Professorship Programme (award number AIVP-2024-004). Any opinions, findings and conclusions or recommendations expressed in this material are those of the author(s) and do not reflect the views of National Research Foundation, Singapore and the Ministry of Digital Development and Information.

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

## A  Bregman divergence and associated policy definitions

**Definition 4.** *Bregman divergence. Let $h : \Delta(A) \to \mathbb{R}$ be a convex and differentiable function. We define*

$$B(x, y; h) := h(x) - h(y) - \langle \nabla h(y), x - y \rangle \tag{55}$$

*as the Bregman divergence with distance-generating function $h$.*

The Bregman divergence represents the distance between the first-order Taylor expansion and the actual function value, intuitively representing the strength of the convexity. A list of common distance-generating functions and their associated Bregman divergences is given in Table 5.

Table 5: Bregman divergence for common distance-generating functions.

| Distance-generating function ($h$) | Bregman divergence ($B(\cdot, \cdot; h)$) |
|---|---|
| $\ell_1$-norm $\|p\|_1$ | 0 for $x, y \in \mathbb{R}^d_+$. |
| Squared $\ell_2$-norm $\frac{1}{2} \|x\|_2^2$ | Squared Euclidian distance $D_{\mathrm{SE}}(x, y) = \frac{1}{2} \|x - y\|_2^2$ for $x, y \in \mathbb{R}^d$ |
| Negative entropy $\sum_i p(i) \log(p(i))$ | Kullbach-Leibler divergence $D_{\mathrm{KL}}(p, q) = \sum_i p(i) \log(p(i)/q(i))$ for $p, q \in \Delta$ |

Table 6 and Table 7 show the update rules for different policy and transition kernel parametrisations. They hold true for any value function as well as Lagrangian value functions, so the tables omit the bold for generality purposes.

Table 6: Update rules for different policy parametrisations and Bregman divergences.

| Parametrisation | Bregman divergence | Update rule |
|---|---|---|
| Direct: $\pi = \theta$ | $D_{\mathrm{SE}}(\theta, \theta_t)$ | $\theta_{t+1} \leftarrow \mathrm{proj}_\Pi(\theta_t - \eta_t \nabla_\theta V_{\pi,p}(d_0))$ |
| Softmax: $\pi(a\|s) = \frac{\exp(\theta_{s,a})}{\sum_{a'} \exp(\theta_{s,a'})}$ | $D_{\mathrm{SE}}(\theta, \theta_t)$ | $\theta_{t+1} \leftarrow \theta_t - \eta_t \nabla_\theta V_{\pi,p}(d_0)$ |
| Softmax: $\pi(a\|s) = \frac{\exp(\theta_{s,a})}{\sum_{a'} \exp(\theta_{s,a'})}$ | Occupancy-weighted KL-divergence (Eq. 16) | $\pi_k^{t+1}(a\|s) = \frac{1}{Z_k^t(s)} (\pi_k^t(a\|s))^{1 - \frac{\eta\alpha}{1-\gamma}} e^{\frac{-\eta \mathbf{Q}_{\pi_k^t,p}^\alpha(s,a)}{1-\gamma}}$ |

Table 7: Update rules and uncertainty sets for different parametric transition kernels (PTKs). The notation $D(\cdot, \cdot)$ indicates a particular distance function such as $\ell_1$ or $\ell_\infty$-norm and the $\bar{x}$ notation indicates the nominal model for any parameter $x$ (typically obtained from parameter estimates).

| Parametrisation | Update rule | Uncertainty sets |
|---|---|---|
| Entropy PTK: $p(s'\|s,a) = \frac{\bar{p}(s'\|s,a) \exp\left(\frac{\zeta^\mathsf{T} \phi(s')}{\lambda'^\mathsf{T} \varphi(s,a)}\right)}{\sum_{s''} \bar{p}(s''\|s,a) \exp\left(\frac{\zeta^\mathsf{T} \phi(s'')}{\lambda'^\mathsf{T} \varphi(s,a)}\right)}$ | $\xi_{t+1} \leftarrow \arg\max_\xi \{\langle \eta_t \nabla_\xi V_{\pi,p}(d_0), \xi \rangle - D(\xi, \xi_t)\}$ | $\mathcal{U}_\xi = \{\xi : D(\xi, \bar{\xi}) \leq \kappa_\xi\}$. |
| Gaussian mixture PTK: $p(s'\|s,a) = \sum_{m=1}^M \omega_m \mathcal{N}(\eta^\mathsf{T} \zeta(s,a))$ | $\xi_{t+1} \leftarrow \arg\max_\xi \{\langle \eta_t \nabla_\xi V_{\pi,p}(d_0), \xi \rangle - D(\xi, \xi_t)\}$ | $\mathcal{U}_\eta = \{\eta : \forall m \in [M], D(\eta, \bar{\eta}_m) \leq \kappa_\eta\}$ 
 $\mathcal{U}_\omega = \{\omega \in \Delta^M : \forall m \in [M], \omega_m \in [F_{\omega_m}^{-1}(\delta/2), F_{\omega_m}^{-1}(1-\delta/2)]\}$ |

# B Robust Sample-based PMD-PD pseudocode

## B.1 Discretised state-action space

---

**Algorithm 3** Robust Sample-based PMD-PD (discrete setting)

---

Inputs: Discount $\gamma \in [0,1)$, error tolerances $\epsilon, \epsilon'_0 > 0$, failure probability $\delta \in [0,1)$, learning rates $\eta = \frac{1-\gamma}{\alpha}$, $\eta_p > 0$, $\eta_\lambda = 1.0$, and penalty coefficients $\alpha = \frac{2\gamma^2 m \eta_\lambda}{(1-\gamma)^3}$ and $\alpha_p > 0$.

Initialise: $\pi_0$ as uniform random policy, $\lambda_{0,j} = \max\left\{0, -\eta_\lambda \hat{V}_{\pi_0, p_0}(\rho)\right\}$, $p_0 = \bar{p}$.

**for** $k \in \{0, \dots, K-1\}$ **do**

    **for** $t \in \{0, \dots, t_k - 1\}$ **do**

        **for** $(s,a) \in \mathcal{S} \times \mathcal{A}$ **do**

            Generate $M_{Q,k}$ samples of length $N_{Q,k}$ based on $\pi_k^t$ and $p_k$ (following Lemma 9).

            Estimate the value: e.g. for softmax parametrisation,

$$\hat{\mathbf{Q}}^\alpha_{\pi_k^t, p_k}(s,a) = \tilde{\mathbf{c}}_k(s,a) + \alpha \log(\tfrac{1}{\pi_k(a|s)}) + \tfrac{1}{M_{Q,k}} \sum_{j=1}^{M_{Q,k}} \sum_{l=1}^{N_{V,k}-1} \gamma^l \left[ \tilde{\mathbf{c}}_k(s_l^j, a_l^j) + \alpha \sum_{a'} \pi_k^t(a'|s_l^j) \log(\tfrac{\pi_k^t(a'|s_l^j)}{\pi_k(a'|s_l^j)}) \right].$$

        **end for**

        ▷ Policy mirror descent

        **if** Using Softmax parametrisation **then**

$$\pi_k^{t+1}(a|s) \leftarrow (\pi_k^t(a|s))^{1-\frac{\eta\alpha}{1-\gamma}} \exp\left(-\eta \frac{\hat{\mathbf{Q}}^\alpha_{\pi_k^t, p_k}(s,a)}{1-\gamma}\right) \quad \forall (s,a) \in \mathcal{S} \times \mathcal{A}.$$

        **else**

            ▷ Use general Bregman divergence

            $\theta_k^{t+1} \leftarrow \arg\min_{\theta \in \Theta} \eta \left\langle \nabla_\theta \hat{\mathbf{Q}}^\alpha_{\pi_k^t, p_k}, \theta \right\rangle + \alpha B_{d_\rho^{\pi_\theta, p_k}}(\theta, \theta_k)$

            Define $\pi_k^{t+1} := \pi_{\theta_k^{t+1}}$.

        **end if**

    **end for**

    Define $\pi_{k+1} := \pi_k^{t_k}$

    Apply Tabular Approximate TMA (Algorithm 1) with $c = \mathbf{c}_k$, $\pi = \pi_{k+1}$, and $p^0 = p_k^0$ to get $p_k^{t'_k}$.

    Define $p_{k+1} := p_k^{t'_k}$

    ▷ Augmented update of Lagrangian multipliers

    Generate $M_{V,k+1}$ samples of length $N_{V,k+1}$ based on $\pi_{k+1}$ and $p_{k+1}$ starting from $\rho$ ( Lemma 9).

    Estimate $\hat{V}^i_{\pi_{k+1}, p_{k+1}}(\rho) = \frac{1}{M_{V,k+1}} \sum_{j=1}^{M_{Q,k+1}} \sum_{l=1}^{N_{V,k+1}} \gamma^l c_i(s_l^j, a_l^j)$ for all $i \in [m]$.

    $\lambda_{k+1,i} \leftarrow \max\left\{-\eta_\lambda \hat{V}_{\pi_{k+1}, p_{k+1}}(\rho), \lambda_{k,i} + \eta_\lambda \hat{V}^i_{\pi_{k+1}, p_{k+1}}(\rho)\right\} \quad \forall i \in [m]$

**end for**

---

## B.2 Continuous state-action space

---

**Algorithm 4** Robust Sample-based PMD-PD (continuous setting)

---

Inputs: Discount factor $\gamma \in [0,1)$, sample sizes $M_{Q,k}, M_{V,k}$ and episode lengths $N_{Q,k}, N_{V,k}$ for all $k \in [K]$, learning rate $\eta$, TMA learning rate $\eta_p > 0$, dual learning rate $\eta_\lambda$, error tolerance for LTMA $\epsilon'_0 > 0$, and penalty coefficients $\alpha > 0$ and $\alpha_p > 0$.

Initialise: $\pi_0$ as uniform random policy, $\lambda_{0,j} = \max\left\{0, -\eta_\lambda \hat{V}_{\pi_0,p_0}(\rho)\right\}$, $p_0 = \bar{p}$.

**for** $k \in \{0, \dots, K-1\}$ **do**

    **for** $t \in \{0, \dots, t_k - 1\}$ **do**

        Generate $M_{Q,k}$ samples of length $N_{Q,k}$ based on $\pi_k^t$ and $p_k$.

        Update $\hat{\mathbf{Q}}_{\pi_k^t, p_k} \leftarrow \arg\min_{Q \in \mathcal{F}_Q} \frac{1}{M_{Q,k}} \sum_{j=1}^{M_{Q,k}} \left(Q(s_0^j, a_0^j) - \sum_{l=0}^{N_{Q,k}-1} \gamma^l \left[\tilde{\mathbf{c}}_k(s_l^j, a_l^j)\right]\right)^2$.

        $\triangleright$ Policy mirror descent

        $\theta_k^{t+1} \leftarrow \arg\min_{\theta \in \Theta} \eta \left\langle \nabla_\theta \hat{\mathbf{Q}}_{\pi_k^t, p_k}^\alpha, \theta \right\rangle + \alpha B_{d_\rho^{\pi_\theta, p_k}}(\theta, \theta_k)$.

        Define $\pi_k^{t+1} := \pi_{\theta_k^{t+1}}$

    **end for**

    Define $\pi_{k+1} := \pi_k^{t_k}$

    Apply Function Approximation TMA (Algorithm 1) with $c = \mathbf{c}_k$, $\pi = \pi_{k+1}$, and $p^0 = p_k^0$ to get $p_k^{t'_k}$.

    Define $p_{k+1} := p_k^{t'_k}$.

    $\triangleright$ Dual update of augmented Lagrangian multipliers

    Generate $M_{V,k+1}$ samples of length $N_{V,k+1}$ based on $\pi_{k+1}$ and $p_{k+1}$ starting from $\rho$.

    Estimate $\hat{\mathbf{V}}_{\pi_{k+1}}^i(\rho) \leftarrow \arg\min_{V \in \mathcal{F}_V} \frac{1}{M_{V,k+1}} \sum_{j=1}^{M_{V,k+1}} \left(V(s_0^j) - \sum_{l=1}^{N_{V,k+1}} \gamma^l c_i(s_l^j, a_l^j)\right)^2$ for all $i \in [m]$.

    $\lambda_{k+1,i} \leftarrow \max\left\{-\eta_\lambda \hat{V}_{\pi_{k+1}}^i(\rho), \lambda_{k,i} + \eta_\lambda \hat{V}_{\pi_{k+1}}^i(\rho)\right\}$ for all $i \in [m]$.

**end for**

---

# C Supporting lemmas and proofs for Approximate TMA

## C.1 Recursion property

The following straightforward recursion property is useful in the proof of the regret bound of approximate TMA.

**Lemma 12** (Recursion property, Lemma 13 in Xiao (2022)). *Let $\beta \in (0,1)$, and $b > 0$. If $\{a_t\}_{t \geq 0}$ is a non-negative sequence such that $x_{t+1} \leq \beta a_t + b$, then $x_t \leq \beta^t x_0 + \frac{b}{1-\beta}$.*

## C.2 Regret bound for approximate TMA

The regret bound of Approximate TMA (Theorem 1) is shown below.

*Proof.* Note that

$$\langle p^*(\cdot|s,a) - p^{t+1}(\cdot|s,a), \hat{G}_{\pi,p^t}(s,a,\cdot)\rangle = \langle p^*(\cdot|s,a) - p^t(\cdot|s,a), \hat{G}_{\pi,p^t}(s,a,\cdot)\rangle + \langle p^t(\cdot|s,a) - p^{t+1}(\cdot|s,a), \hat{G}_{\pi,p^t}(s,a,\cdot)\rangle.$$

From the above and the pushback property in Eq. 30 with $p = p^*$,

$$\langle p^*(\cdot|s,a) - p^t(\cdot|s,a), \hat{G}_{\pi,p^t}(s,a,\cdot)\rangle + \langle p^t(\cdot|s,a) - p^{t+1}(\cdot|s,a), \hat{G}_{\pi,p^t}(s,a,\cdot)\rangle \leq \frac{\alpha_p(t)}{\eta_p(t)}\left(B(p^*,p^t) - B(p^*,p^{t+1})\right)$$

Weighting both sides according to the occupancy measure, we obtain

$$
\underbrace{\frac{1}{1-\gamma}\sum_{s\in\mathcal{S}}d_\rho^{\pi,p^*}(s)\sum_{a\in\mathcal{A}}\pi(a|s)\langle p^*(\cdot|s,a)-p^t(\cdot|s,a),\hat{G}_{\pi,p^t}(s,a,\cdot)\rangle}_{A}
$$
$$
+\underbrace{\frac{1}{1-\gamma}\sum_{s\in\mathcal{S}}d_\rho^{\pi,p^*}(s)\sum_{a\in\mathcal{A}}\pi(a|s)\langle p^t(\cdot|s,a)-p^{t+1}(\cdot|s,a),\hat{G}_{\pi,p^t}(s,a,\cdot)\rangle}_{B}
$$
$$
\leq\frac{\alpha_p(t)}{\eta_p(t)}\left(B_{d_\rho^{\pi,p^*}}(p^*,p^t)-B_{d_\rho^{\pi,p^*}}(p^*,p^{t+1})\right).
$$

For part A), note that

$$
\frac{1}{1-\gamma}\sum_{s\in\mathcal{S}}d_\rho^{\pi,p^*}(s)\sum_{a\in\mathcal{A}}\pi(a|s)\langle p^*(\cdot|s,a)-p^t(\cdot|s,a),\hat{G}_{\pi,p^t}(s,a,\cdot)\rangle
$$
$$
=\frac{1}{1-\gamma}\sum_{s\in\mathcal{S}}d_\rho^{\pi,p^*}(s)\sum_{a\in\mathcal{A}}\pi(a|s)\left(\langle p^*(\cdot|s,a)-p^t(\cdot|s,a),G_{\pi,p^t}(s,a,\cdot)\rangle+\langle p^*(\cdot|s,a)-p^t(\cdot|s,a),\hat{G}_{\pi,p^t}(s,a,\cdot)-G_{\pi,p^t}(s,a,\cdot)\rangle\right)
$$
$$
\text{(decomposing)}
$$
$$
=V_{\pi,p^*}(\rho)-V_{\pi,p^t}(\rho)+\frac{1}{1-\gamma}\sum_{s\in\mathcal{S}}d_\rho^{\pi,p^*}(s)\sum_{a\in\mathcal{A}}\pi(a|s)\langle p^*(\cdot|s,a)-p^t(\cdot|s,a),\hat{G}_{\pi,p^t}(s,a,\cdot)-G_{\pi,p^t}(s,a,\cdot)\rangle
$$
$$
\text{(Lemma 23)}
$$
$$
\geq V_{\pi,p^*}(\rho)-V_{\pi,p^t}(\rho)-\frac{2\epsilon'}{1-\gamma},
$$

where the last step follows from a bound of the absolute value as in the proof of Lemma 5.

For part B), note that applying the ascent property (Eq. 31 on $\hat{G}$ results in

$$
0\geq\frac{1}{1-\gamma}\sum_{s\in\mathcal{S}}d_\rho^{\pi,p^*}(s)\sum_{a\in\mathcal{A}}\pi(a|s)\langle p^t(\cdot|s,a)-p^{t+1}(\cdot|s,a),\hat{G}_{\pi,p^t}(s,a,\cdot)\rangle
$$
$$
=\frac{1}{1-\gamma}\sum_{s\in\mathcal{S}}d_\rho^{\pi,p^*}(s)\sum_{a\in\mathcal{A}}\pi(a|s)\big(\langle p^t(\cdot|s,a)-p^{t+1}(\cdot|s,a),G_{\pi,p^t}(s,a,\cdot)\rangle+
$$
$$
\langle p^t(\cdot|s,a)-p^{t+1}(\cdot|s,a),\hat{G}_{\pi,p^t}(s,a,\cdot)-G_{\pi,p^t}(s,a,\cdot)\rangle\big)
$$
$$
\text{(decomposing)}
$$
$$
=\frac{1}{1-\gamma}\sum_{s\in\mathcal{S}}\frac{d_\rho^{\pi,p^*}(s)}{d_\rho^{\pi,p^{t+1}}(s)}d_\rho^{\pi,p^{t+1}}(s)\sum_{a\in\mathcal{A}}\pi(a|s)\big(\langle p^t(\cdot|s,a)-p^{t+1}(\cdot|s,a),G_{\pi,p^t}(s,a,\cdot)\rangle+
$$
$$
\langle p^t(\cdot|s,a)-p^{t+1}(\cdot|s,a),\hat{G}_{\pi,p^t}(s,a,\cdot)-G_{\pi,p^t}(s,a,\cdot)\rangle\big)
$$

$$\geq \left\| \frac{d_\rho^{\pi,p^*}}{d_\rho^{\pi,p^{t+1}}} \right\|_\infty \bigg( \big( V_{\pi,p^t}(\rho) - V_{\pi,p^{t+1}}(\rho) \big) +$$

$$\frac{1}{1-\gamma} \sum_{s \in \mathcal{S}} d_\rho^{\pi,p^{t+1}}(s) \sum_{a \in \mathcal{A}} \pi(a|s) \langle p^t(\cdot|s,a) - p^{t+1}(\cdot|s,a), \hat{G}_{\pi,p^t}(s,a,\cdot) - G_{\pi,p^t}(s,a,\cdot) \rangle \bigg)$$

(since the term is negative, upper bound the ratio; Lemma 23)

$$\geq M \bigg( \big( V_{\pi,p^t}(\rho) - V_{\pi,p^{t+1}}(\rho) \big) +$$

$$\frac{1}{1-\gamma} \sum_{s \in \mathcal{S}} d_\rho^{\pi,p^{t+1}}(s) \sum_{a \in \mathcal{A}} \pi(a|s) \langle p^t(\cdot|s,a) - p^{t+1}(\cdot|s,a), \hat{G}_{\pi,p^t}(s,a,\cdot) - G_{\pi,p^t}(s,a,\cdot) \rangle \bigg)$$

(definition of $M = \sup_{p \in \mathcal{P}} \left\| \frac{d_\rho^{\pi,p^*}}{d_\rho^{\pi,p}} \right\|_\infty$)

$$\geq M \left( V_{\pi,p^t}(\rho) - V_{\pi,p^{t+1}}(\rho) - \frac{2\epsilon'}{1-\gamma} \right),$$

where the last step again follows from derivations in Lemma 5.

Combining both A) and B), denoting $\delta_t = V_{\pi,p^*}(\rho) - V_{\pi,p^t}(\rho)$, and rearranging, we obtain

$$\delta_t + M \left( \delta_{t+1} - \delta_t - \frac{2\epsilon'}{1-\gamma} \right) \leq \frac{\alpha_p(t)}{\eta_p(t)(1-\gamma)} \left( B_{d_\rho^{\pi,p^*}}(p^*,p^t) - B_{d_\rho^{\pi,p^*}}(p^*,p^{t+1}) \right) + \frac{2\epsilon'}{1-\gamma}$$

$$M\delta_{t+1} + (1-M)\delta_t \leq \frac{\alpha_p(t)}{\eta_p(t)(1-\gamma)} \left( B_{d_\rho^{\pi,p^*}}(p^*,p^t) - B_{d_\rho^{\pi,p^*}}(p^*,p^{t+1}) \right) + \frac{2\epsilon'(1+M)}{1-\gamma} \qquad \text{(rearrange)}$$

$$\left( \frac{1}{M} - 1 \right) \delta_t + \delta_{t+1} \leq \frac{\alpha_p(t)}{M\eta_p(t)(1-\gamma)} \left( B_{d_\rho^{\pi,p^*}}(p^*,p^t) - B_{d_\rho^{\pi,p^*}}(p^*,p^{t+1}) \right) + \frac{4\epsilon'}{1-\gamma}$$

(divide by $M$ and note $\frac{1}{M} + 1 < 2$)

$$\delta_{t+1} + \frac{\alpha_p(t)}{M\eta_p(t)(1-\gamma)} B_{d_\rho^{\pi,p^*}}(p^*,p^{t+1}) \leq \left( 1 - \frac{1}{M} \right) \left( \delta_t + \frac{\alpha_p(t)}{(M-1)\eta_p(t)(1-\gamma)} B_{d_\rho^{\pi,p^*}}(p^*,p^t) \right) + \frac{4\epsilon'}{1-\gamma}$$

(rearrange)

$$\delta_{t+1} + \frac{\alpha_p(t+1)}{(M-1)\eta_p(t+1)(1-\gamma)} B_{d_\rho^{\pi,p^*}}(p^*,p^{t+1}) \leq \left( 1 - \frac{1}{M} \right) \left( \delta_t + \frac{\alpha_p(t)}{(M-1)\eta_p(t)(1-\gamma)} B_{d_\rho^{\pi,p^*}}(p^*,p^t) \right) + \frac{4\epsilon'}{1-\gamma}.$$

(since $M \geq \frac{1}{1-\gamma}$, it follows that $\eta_p(t+1) \geq \eta_p(t)/\gamma \geq \eta_p(t)\frac{M}{M-1}$)

Using Lemma 12 with $\beta = 1 - \frac{1}{M}$, $x_t = \delta_t + \frac{\alpha_p(t)}{(M-1)\eta_p(t)(1-\gamma)} B_{d_\rho^{\pi,p^*}}(p^*,p^t)$, and $b = \frac{4\epsilon'}{1-\gamma}$, it follows that

$$\delta_t \leq \delta_t + \frac{\alpha_p(t)}{(M-1)\eta_p(t)(1-\gamma)} B_{d_\rho^{\pi,p^*}}(p^*,p^t)$$

$$\leq \left( 1 - \frac{1}{M} \right)^t \left( \delta_0 + \frac{\alpha_p(0)}{(M-1)\eta_p(0)(1-\gamma)} B_{d_\rho^{\pi,p^*}}(p^*,p^0) \right) + \frac{4M\epsilon'}{1-\gamma}$$

$$\leq \left( 1 - \frac{1}{M} \right)^t \frac{3C}{1-\gamma} + \frac{4M}{1-\gamma}\epsilon',$$

where the last step follows from $\delta_0 \leq \frac{2C}{1-\gamma}$ and $\eta_p(0) \geq \alpha_p(0)\frac{\gamma}{C(1-\gamma)} B_{d_\rho^{\pi,p^0}}(p^*,p^0)$. $\qquad \square$

## C.3 Approximate action next-state value

The proof of Lemma 6 is given below.

*Proof.* By triangle inequality, the error $\left\| \hat{G} - G \right\|_\infty$ is upper bounded by the truncation error and the estimation error, so it suffices to bound both by half of the desired error in Eq. 35.

For the truncation error, observe that

$$\left| \mathbb{E}\left[ \sum_{t=N_{G,t}}^{\infty} \gamma^t c(s_t, a_t, s_{t+1}) \right] \right| \leq \sum_{t=N_{G,t}}^{\infty} C\gamma^t = C\frac{\gamma^{N_{G,t}}}{1-\gamma}.$$

For the estimation error of any particular $(s, a, s') \in \mathcal{S} \times \mathcal{A} \times \mathcal{S}$ and any $t \in \{0, \ldots, t'_k\}$, apply Hoeffding's inequality with the random variable $X_t(s, a, s') = \sum_{l=0}^{N_{G,t}-1} \gamma^l \left[ c(s_l^j, a_l^j) \right] \in [-\frac{C}{1-\gamma}, \frac{C}{1-\gamma}]$ conditioned on $(s_0, a_0, s_1) = (s, a, s')$, error tolerance $\zeta = \frac{\gamma^{N_{G,t}} C}{1-\gamma}$, and failure probability $\delta' = \frac{\delta}{S^2 A t'_k}$. This results in

$$P(|\bar{X}(s, a, s') - \mathbb{E}[\bar{X}(s, a, s')]| \geq \zeta) \leq 2\exp\left( -\frac{M_{G,t}\zeta^2(1-\gamma)^2}{2C^2} \right) = \delta'$$

$$M_{G,t} \leq \frac{2C^2}{\zeta^2(1-\gamma)^2}\log\left(\frac{2}{\delta'}\right) = \frac{2\gamma^{2N_{G,t}}}{(1-\gamma)^2}\log\left(\frac{2}{\delta'}\right).$$

Repeating the same procedure for all $\mathcal{S} \times \mathcal{A} \times \mathcal{S}$, and applying union bound, we obtain a failure probability

$$P\left( \bigcup_{s,a,s',t} |\bar{X}_t(s, a, s') - \mathbb{E}[\bar{X}_t(s, a, s')]| \geq \zeta \right)$$

$$\leq \sum_{s,a,s',t} P(|\bar{X}_t(s, a, s') - \mathbb{E}[\bar{X}_t(s, a, s')]| \geq \zeta)$$

$$\leq \sum_{s,a,s',t} \delta' = \delta.$$

This concludes the proof that the setting $M_{G,t} = \frac{2\gamma^{-2N_{G,t}}}{(1-\gamma)^2}\log\left(\frac{2}{\delta'}\right)$ is sufficient to obtain an estimation error of at most $C\frac{\gamma^{N_{G,t}}}{1-\gamma}$ with a failure probability of at most $\delta$ across all iterations and state-action-state triplets. $\qquad\square$

### C.4 Sample complexity of Approximate TMA

The proof of Lemma 7 is given below.

*Proof.* Note that it suffices to bound both parts in the RHS of Eq. 34 with $\epsilon'/2$.

For the first part of Eq. 34, note that requiring

$$\left(1 - \frac{1}{M}\right)^t \frac{3C}{1-\gamma} \leq \frac{\epsilon'}{2} \tag{56}$$

implies that the number of iterations is bounded by

$$t \leq \log\left(\frac{\epsilon'(1-\gamma)}{6C}\right) / \log\left(\frac{M-1}{M}\right) = \log_{\frac{M}{M-1}}\left(\frac{6C}{\epsilon'(1-\gamma)}\right).$$

This confirms that the setting $t'_k \geq \log_{\frac{M}{M-1}}\left(\frac{6C}{\epsilon'(1-\gamma)}\right)$ is sufficient to satisfy Eq. 56.

For the second part of Eq. 34, define $\epsilon' = 2C\frac{\gamma^H}{1-\gamma}$ consistent with Eq. 32. This amounts to requiring that

$$\frac{4M}{1-\gamma}\epsilon' = \frac{8MC}{(1-\gamma)^2}\gamma^H \leq \epsilon'/2 \tag{57}$$

This implies a requirement of at most

$$H \leq \log\left(\frac{\epsilon'(1-\gamma)^2}{16MC}\right) / \log(\gamma) = \log_{\frac{1}{\gamma}}\left(\frac{16MC}{\epsilon'(1-\gamma)^2}\right).$$

Therefore the setting of $H \geq \log_{\frac{1}{\gamma}} \left( \frac{16MC}{\epsilon'(1-\gamma)^2} \right)$ suffices to obtain Eq. 57.

With the settings of $M_G$, $t_k'$, and $H$, note that $\gamma^{-2H} = \left( \frac{1}{\gamma} \right)^{2 \log_{\frac{1}{\gamma}} \left( \frac{16MC}{\epsilon'(1-\gamma)^2} \right)} = \left( \frac{16MC}{\epsilon'(1-\gamma)^2} \right)^2$. Therefore

$$
\begin{aligned}
M_G &= \frac{2\gamma^{-2H}}{(1-\gamma)^2} \log \left( \frac{2S^2 A t_k'}{\delta} \right) \\
&= 2 \left( \frac{16M}{(1-\gamma)^2 \epsilon'} \right)^2 \log \left( \frac{2S^2 A t_k'}{\delta} \right) .
\end{aligned}
$$

The total number of samples is determined by the number of state-action-state triplets, the number of iterations, the horizon, and the number of trajectories per mini-batch:

$$
\begin{aligned}
n &= |S|^2 \, |A| \, t_k' H M_G \\
&\leq S^2 A \log_{\frac{M}{M-1}} \left( \frac{6C}{\epsilon'(1-\gamma)} \right) \log_{\frac{1}{\gamma}} \left( \frac{16M}{(1-\gamma)^2 \epsilon'} \right) 2 \left( \frac{16M}{(1-\gamma)^2 \epsilon'} \right)^2 \log \left( \frac{2t_k' S^2 A}{\delta} \right) \\
&= 512 S^2 A \frac{M^2}{(1-\gamma)^4 \epsilon''^2} \log_{\frac{M}{M-1}} \left( \frac{6C}{\epsilon'(1-\gamma)} \right) \log \left( \frac{16M}{(1-\gamma)^4 \epsilon'} \right) \log \left( \frac{2t_k' S^2 A}{\delta} \right) \\
&= \tilde{\mathcal{O}} \left( \frac{S^2 A M^2}{(1-\gamma)^4 \epsilon'^2} \right) .
\end{aligned}
$$

$\square$

# D    Supporting lemmas and proofs for Robust Sample-based PMD-PD

## D.1    Analysis of the multipliers

To analyse the average regret in terms of the value directly, the lemma below provides an equivalence between the unconstrained value and the Lagrangian value at the optimum.

**Lemma 13** (Complementary slackness). *For any CMDP and any $j \in [m]$, the optimal constrained solution $(\pi^*, \lambda^*)$ has either $\lambda_j^* = 0$ or $V_{\pi^*}^j(\rho) = 0$, such that its Lagrangian value is equal to its unconstrained value:*

$$
\mathbf{V}_{\pi^*}(\rho; \lambda^*) = V_{\pi^*}(\rho) . \tag{58}
$$

**Lemma 14** (Bounds on the multipliers). *The sequence of multipliers produced by Robust Sample-based PMD-PD $\{\lambda_{k,j}\}_{k \geq 0}$, $j \in [m]$ satisfy the following properties:*

1. *Non-negativity: for any macro step $k \geq 0$, $\lambda_{k,j} \geq 0$ for all $j \in [m]$.*

2. *Positive augmented multiplier: for any macro step $k \geq 0$, $\lambda_{k,j} + \eta_\lambda \hat{V}_{\pi_k, p_k}^j(\rho) \geq 0$ for all $j \in [m]$.*

3. *Bounded initial multiplier: for macro step $0$, $|\lambda_{0,j}|^2 \leq |\eta_\lambda \hat{V}_{\pi_0, p_0}^j(\rho)|^2$ for all $j \in [m]$*

4. *Bounded value: for macro step $k > 0$, $|\lambda_{k,j}|^2 \geq |\eta_\lambda \hat{V}_{\pi_k, p_k}^j(\rho)|^2$ for all $j \in [m]$*

*Proof.* 1) Note that $\lambda_{0,j} \geq 0$ is trivially satisfied by the initialisation in Algorithm 3. Via induction, if $\lambda_{k,j} \geq 0$ note that if $\hat{V}_{\pi_{k+1}, p_{k+1}}(\rho) \geq 0$ then $\lambda_k, j + \eta_\lambda \hat{V}_{\pi_{k+1}, p_{k+1}}(\rho) \geq 0$; if it is negative, then $-\eta_\lambda \hat{V}_{\pi_{k+1}, p_{k+1}}(\rho) \geq 0$. 2) Note that $0 \leq \lambda_{k,j} = \max\{\lambda_{k-1,j} + \eta_\lambda \hat{V}_{\pi_k, p_k}(\rho), -\eta_\lambda \hat{V}_{\pi_k, p_k}(\rho)\}$. This implies either a) $\eta_\lambda \hat{V}_{\pi_k, p_k}(\rho) \geq \lambda_{k-1,j} \geq 0$ or b) $\hat{V}_{\pi_k, p_k}(\rho) \leq 0$. In case a),

$$
\lambda_{k,j} + \eta_\lambda \hat{V}_{\pi_k, p_k}^j(\rho) \geq 0 .
$$

In case b),

$$\lambda_{k,j} = -\eta_\lambda \hat{V}^j_{\pi_k,p_k} \geq 0 \,.$$

3) This follows directly from the initialisation to $\lambda_{0,j} = \max\{0, -\eta_\lambda \hat{V}^j_{\pi_0,p_0}(\rho)\}$.

4) Note that

$$
\begin{aligned}
|\lambda_{k,j}|^2 &= |\max\{\lambda_{k-1,j} + \eta_\lambda \hat{V}^j_{\pi_k,p_k}(\rho), -\eta_\lambda \hat{V}^j_{\pi_k,p_k}(\rho)\}|^2 \\
&= \max\{|\lambda_{k-1,j} + \eta_\lambda \hat{V}^j_{\pi_k,p_k}|^2, |\eta_\lambda \hat{V}^j_{\pi_k,p_k}|^2\} \\
&\geq |\eta_\lambda \hat{V}^j_{\pi_k,p_k}(\rho)|^2 \,.
\end{aligned}
$$

$\square$

The analysis of the dual variables focuses on the inner product between the modified Lagrangian multiplier and the constraint-costs, which represents the total constraint-penalty at the end of an iteration. Due to the approximation errors $\epsilon_k = \hat{V}^{1:m}_{\pi_k,p_k}(\rho) - V^{1:m}_{\pi_k,p_k}(\rho)$, the inner product can be written as

$$
\left\langle \lambda_k + \eta_\lambda \hat{V}^{1:m}_{\pi_k,p_k}(\rho), V^{1:m}_{\pi_k^{t_k},p_k}(\rho) \right\rangle = \left\langle \lambda_k, \hat{V}^{1:m}_{\pi_k^{t_k},p_{k+1}}(\rho) \right\rangle + \left\langle \lambda_k, -\epsilon_k \right\rangle + \left\langle \eta_\lambda V^{1:m}_{\pi_k,p_k}(\rho), V^{1:m}_{\pi_k^{t_k},p_{k+1}}(\rho) \right\rangle + \left\langle \eta_\lambda \epsilon_k, V^{1:m}_{\pi_k^{t_k},p_{k+1}}(\rho) \right\rangle \,.
$$

(59)

The inner product can be lower bounded, which leads to a somewhat complex summation but a useful one that can be telescoped in the average regret analysis.

**Lemma 15** (Lower bound on the inner product, Eq. 41 in Liu et al. (2021b)). *For any $k = 0, 1, \ldots, K-1$,*

$$
\left\langle \lambda_k + \eta_\lambda \hat{V}^{1:m}_{\pi_k,p_k}(\rho), V^{1:m}_{\pi_{k+1},p_{k+1}}(\rho) \right\rangle
$$
$$
\geq \frac{1}{2\eta_\lambda} \left( \|\lambda_{k+1}\|^2 - \|\lambda_k\|^2 \right) + \frac{\eta_\lambda}{2} \left( \left\|V^{1:m}_{\pi_k,p_k}(\rho)\right\|^2 - \left\|V^{1:m}_{\pi_{k+1},p_{k+1}}(\rho)\right\|^2 \right)
$$
$$
+ \left\langle \lambda_k, -\epsilon_{k-1} \right\rangle + \|\epsilon_{k+1}\|^2 + \eta_\lambda \left\langle \epsilon_k, V^{1:m}_{\pi_k,p_k}(\rho) \right\rangle - \eta_\lambda \|\epsilon_k\| - 2\eta_\lambda \left\langle V^{1:m}_{\pi_{k+1},p_{k+1}}(\rho), \epsilon_{k+1} \right\rangle - \frac{\gamma^2 \eta_\lambda}{(1-\gamma)^4} B_{d^{\pi_{k+1},p_{k+1}}_\rho}(\pi_{k+1}, \pi_k) \,.
$$

### D.2 Analysis of the Bregman divergence

Another essential part of the average regret analysis is the pushback property, which allows a telescoping sum.

**Lemma 16** (Pushback property, Lemma 2 in Liu et al. (2021b)). *If $x^* = \arg\min_{x \in \Delta} f(x) + B(x, y; h)$ for a fixed $y \in Int(\Delta)$, then for $\alpha > 0$ and any $z \in \Delta$,*

$$f(x^*) + B(x^*, y; h) \leq f(x) + \alpha \left( B(z, y; h) - B(z, x^*; h) \right) \,.$$

By selecting $z = \pi$, $y = \pi_k^*$, and $x^* = \pi^*$, and using the weighted Bregman divergence, it is possible to use the property in the context of policies in the interior of the probability simplex.

**Lemma 17** (Pushback property, Lemma 11 in Liu et al. (2021b) rephrased). *For any policy $\pi \in Int(\Delta(\mathcal{A}))$ and any $p$,*

$$\mathbf{V}_{\pi_k^*,p}(\rho) + \frac{\alpha}{1-\gamma} B_{d^{\pi_k^*,p}_\rho}(\pi_k^*, \pi) \leq \mathbf{V}_{\pi,p}(\rho) + \alpha \left( B_{d^{\pi,p}_\rho}(\pi, \pi_k) - B_{d^{\pi,p}_\rho}(\pi, \pi_k^*) \right) \,.$$

The weighted Bregman divergence under the generating function $\sum_i p(i) \log(p(i))$ is equivalent to the KL-divergence, which is bounded by $\log(A)$ if the reference policy is uniform.

**Lemma 18** (Bound on Bregman divergence). *Let $\pi'$ be a uniform policy, then for any policy $\pi \in \Pi$ and any $p \in \mathcal{P}$,*

$$B_{d^{\pi,p}_\rho}(\pi, \pi') = \sum_{s \in \mathcal{S}} d^{\pi,p}_\rho(s) \sum_{a \in \mathcal{A}} \pi(a|s) \log(A\pi(a|s)) \leq \log(A) \,.$$

### D.3 Convergence of approximate entropy-regularised NPG

Below is the supporting convergence result for approximate entropy-regularised NPG.

**Lemma 19** (Convergence of approximate entropy-regularised NPG, Theorem 2 of Cen et al. (2022)). *Let $\epsilon > 0$. Then if $\left\| \hat{\mathbf{Q}}^{\alpha}_{\pi^t_k,p} - \mathbf{Q}^{\alpha}_{\pi^t_k,p} \right\|_{\infty} \le \epsilon$,*

$$C_k \ge \left\| \mathbf{Q}^{\alpha}_{\pi^*_k,p} - \mathbf{Q}^{\alpha}_{\pi^0_k,p} \right\|_{\infty} + 2\alpha \left( 1 - \frac{\eta\alpha}{1-\gamma} \left\| \log(\pi^*_k) - \log(\pi^0_k)) \right\|_{\infty} \right)$$

*and*

$$C' \ge \frac{2\epsilon}{1-\gamma}(1 + \frac{\gamma}{\eta\alpha}),$$

*it follows that for all $t \ge 0$*

$$\left\| \mathbf{Q}^{\alpha}_{\pi^*_k,p} - \mathbf{Q}^{\alpha}_{\pi^{t+1}_k,p} \right\|_{\infty} \le C_k\gamma(1-\eta\alpha)^t + \gamma C' \tag{60}$$

$$\left\| \log(\pi^*_k) - \log(\pi^{t+1}_k) \right\|_{\infty} \le 2C_k\alpha^{-1}(1-\eta\alpha)^t + 2\alpha^{-1}C' \tag{61}$$

$$\left\| \mathbf{V}^{\alpha}_{\pi^*_k,p} - \mathbf{V}^{\alpha}_{\pi^{t+1}_k,p} \right\|_{\infty} \le 3C_k(1-\eta\alpha)^t + 3C'. \tag{62}$$

The following lemma provides settings for the number, $t_k$, of iterations to optimise the policy such that it has negligible error on the regularised objective (i.e. with KL-divergence and modified Lagrangian multiplier).

**Lemma 20** (Number of inner loop iterations, Lemma 7 in Liu et al. (2021b)). *Let $\epsilon > 0$, $\left\| \hat{\mathbf{Q}}^{\alpha}_{\pi^t_k,p_k} - \mathbf{Q}^{\alpha}_{\pi^t_k,p_k} \right\|_{\infty} \le \epsilon$, $\eta \le (1-\gamma)/\alpha$, $t_k = \frac{1}{\eta\alpha}\log(3C_kK)$, and*

$$C_k = 2\gamma \left( \frac{1 + \sum_{j=1}^m \lambda_j}{1-\gamma} + \frac{m\eta_\lambda}{(1-\gamma)^2} \right).$$

*It follows that*

$$\left\| \mathbf{V}^{\alpha}_{\pi^*_k,p_k} - \mathbf{V}_{\pi_{k+1},p_k}\alpha \right\|_{\infty} \le \frac{1}{K} + \frac{6\epsilon}{(1-\gamma)^2} \tag{63}$$

*and*

$$\left\| \log(\pi^*_k) - \log(\pi_{k+1}) \right\|_{\infty} \le \frac{2}{3\alpha K} + \frac{4\epsilon}{\alpha(1-\gamma)^2}. \tag{64}$$

The following lemma bounds the performance difference of entropy-regularised NPG based on the approximation error.

**Lemma 21** (Performance difference lemma, Lemma 4 in Cen et al. (2022) and Lemma 6 in Liu et al. (2021b)). *For learning rate $\eta \le (1-\gamma)/\alpha$ and any $t \ge 0$, it holds for any starting distribution $\rho$ that under entropy-regularised NPG,*

$$\mathbf{V}^{\alpha}_{\pi^{t+1}_k,p_k}(\rho) - \mathbf{V}^{\alpha}_{\pi^t_k,p_k}(\rho) \le \frac{2}{1-\gamma} \left\| \hat{\mathbf{Q}}^{\alpha}_{\pi^t_k,p_k} - \mathbf{Q}^{\alpha}_{\pi^t_k,p_k} \right\|_{\infty}. \tag{65}$$

### D.4 Performance difference across transition kernels

In the context of robust MDPs, the performance difference lemmas below have been formulated across transition kernels, which help the analysis of changing transition dynamics in RCMDPs.

**Lemma 22** (First performance difference lemma across transition kernels, Lemma 5.2 in Wang et al. (2023)). *For any pair of transition kernels $p, p' \in \mathcal{P}$ and any policy $\pi \in \Pi$, we have for any value function that*

$$V_{\pi,p}(\rho) - V_{\pi,p'}(\rho) = \frac{1}{1-\gamma} \sum_{s \in \mathcal{S}} d^{\pi,p'}_\rho(s) \left( \sum_{a \in \mathcal{A}} \pi(a|s) \sum_{s' \in \mathcal{S}} (p(s'|s,a) - p'(s'|s,a)) \left[ c(s,a,s') + \gamma V_{\pi,p}(s') \right] \right). \tag{66}$$

**Lemma 23** (Second performance difference lemma across transition kernels, Lemma 5.3 in Wang et al. (2023)). *For any pair of transition kernels $p, p' \in \mathcal{P}$ and any policy $\pi \in \Pi$, we have for any value function that*

$$V_{\pi,p}(\rho) - V_{\pi,p'}(\rho) = \frac{1}{1-\gamma} \sum_{s \in \mathcal{S}} d_\rho^{\pi,p}(s) \left( \sum_{a \in \mathcal{A}} \pi(a|s) \sum_{s' \in \mathcal{S}} (p(s'|s,a) - p'(s'|s,a)) \left[ c(s,a,s') + \gamma V_{\pi,p'}(s') \right] \right). \tag{67}$$

### D.5 Value function approximation

A proof of Lemma 9 is provided below to demonstrate its applicability to the robust Sample-based PMD algorithm.

*Proof.* **a): approximation of $V_{\pi_k^t, p_k}^i(\rho)$ for all $i \in [m]$.**

Pick $i \in [m]$ and $k \in [K]$ arbitrarily. Note that since costs and constraint-costs are in $[-1, 1]$, the cumulative discounted constraint-cost is bounded by $\sum_{l=0}^\infty \gamma^t c_i(s_l, a_l) \in [-\frac{1}{1-\gamma}, \frac{1}{1-\gamma}]$. Denoting the $N_{V,k}$-step cumulative discounted constraint-cost as a random variable $X = \sum_{l=0}^{N_{V,k}} \gamma^l c_i(s_l, a_l)$, and $\bar{X} = \frac{1}{M_{V,k}} \sum_{n=1}^{M_{V,k}} X_n$, we have by Hoeffding's inequality that the number of samples required is derived as

$$P(|\bar{X} - \mathbb{E}[\bar{X}]| \geq \epsilon)$$

$$\leq 2 \exp\left( -2 \frac{\epsilon^2}{\sum_{i=1}^{M_{V,k}} \left( \frac{2}{M_{V,k}(1-\gamma)} \right)^2} \right)$$

$$= 2 \exp\left( -\frac{(1-\gamma)^2 M_{V,k} \epsilon^2}{2} \right) = \delta_k$$

$$M_{V,k} = \log\left( \frac{2}{\delta_k} \right) \frac{2}{(1-\gamma)^2 \epsilon^2} = \Theta\left( \log\left( \delta_k^{-1} \right) \frac{1}{\epsilon^2} \right).$$

Further, note that since constraint-costs are in $[-1, 1]$, it follows that the number of time steps is derived as

$$|\mathbb{E}[X] - V_{\pi_k}^i(\rho)| \leq 2 \sum_{t=N_{V,k}}^\infty \gamma^t = 2 \frac{\gamma^{N_{V,k}}}{1-\gamma} = \epsilon$$

$$N_{V,k} \log(\gamma) = \log((1-\gamma)\epsilon/2)$$

$$N_{V,k} = \log_\gamma((1-\gamma)\epsilon/2) = \Theta\left( \log_{1/\gamma}\left( \frac{1}{\epsilon} \right) \right), \tag{68}$$

thereby obtaining an $\epsilon$-precise estimate of $V_{\pi_k}^i(\rho)$ with the chosen settings of $M_{V,k}$ and $N_{V,k}$.

**b): approximation of $\tilde{Q}_{\pi_k^t, p_k}^{\alpha_t}(s, a)$ for all $(s, a) \in \mathcal{S} \times \mathcal{A}$.** Select $(s, a) \in \mathcal{S} \times \mathcal{A}$ arbitrarily. First note that $\tilde{\mathbf{c}}_k(s, a) = \mathcal{O}(\max\{1, \|\lambda_k\|_1\})$ and similarly $\tilde{V}_{\pi_k^t, p_k}(s) = \mathcal{O}(\max\{1, \|\lambda_k\|_1\})$ (again omitting the division by $1 - \gamma$ from the notation).

Define the random variable

$$X(s,a) = \tilde{\mathbf{c}}(s,a) + \alpha \log\left( \frac{1}{\pi_k(a|s)} \right) + \sum_{l=1}^{N_{Q,k}} \gamma^l \left( \tilde{\mathbf{c}}(s_l, a_l) + \alpha \sum_{a'} \pi_k^t(a'|s_l) \log\left( \frac{\pi_k^t(a'|s_l)}{\pi_k(a'|s_l)} \right) \right).$$

For $t = 0$, the regularisation term drops. Defining $\bar{X} = \frac{1}{M_{Q,k}} \sum_{n=1}^{M_{Q,k}} X(s,a)_n$, we obtain

$$P(|\bar{X} - \mathbb{E}[\bar{X}]| \geq \epsilon)$$

$$\leq 2 \exp\left(-2 \frac{\epsilon^2}{M_{Q,k} \left(\frac{\max\{1, \|\lambda_k\|_1\}}{M_{Q,k}}\right)^2}\right) = \delta_k$$

$$M_{Q,k} = \log\left(\frac{2}{\delta_k}\right) \frac{\max\{1, \|\lambda_k\|_1\}^2}{\epsilon^2} = \Theta\left(\log\left(\delta_k^{-1}\right) \frac{\max\{1, \|\lambda_k\|_1\}^2}{\epsilon^2}\right).$$

Therefore, setting $M_{Q,K} = \Theta\left(\log(\delta_k^{-1}) \frac{\max\{1, \|\lambda_k\|_1\}^2 + \epsilon t_k}{\epsilon^2}\right)$ is sufficient for approximating $\tilde{\mathbf{Q}}_{\pi_k^0, p_k}^\alpha(s, a)$.

For any $t > 0$, note that the regularisation term is bounded by

$$\alpha \sum_{a'} \pi_k^t(a'|s_l) \log\left(\frac{\pi_k^t(a'|s_l)}{\pi_k(a'|s_l)}\right)$$

$$= \tilde{\mathbf{V}}_{\pi_k^t, p_k}^\alpha(s) - \tilde{\mathbf{V}}_{\pi_k^t, p_k}(s)$$

$$\leq \tilde{\mathbf{V}}_{\pi_k^t, p_k}^\alpha(s) + \mathcal{O}(\max\{1, \|\lambda_k\|_1\}) \qquad \text{(from range)}$$

$$\leq \tilde{\mathbf{V}}_{\pi_k, p_k}^\alpha(s) + \mathcal{O}(\max\{1, \|\lambda_k\|_1\}) + \sum_{t=0}^{t-1} \frac{2}{1-\gamma} \left\|\hat{\mathbf{Q}}_{\pi_k^t, p_k}^\alpha - \tilde{\mathbf{Q}}_{\pi_k^t, p_k}^\alpha\right\|_\infty \qquad \text{(iteratively applying Lemma 21)}$$

$$= \mathcal{O}(\max\{1, \|\lambda_k\|_1\}) + \sum_{t=0}^{t-1} \frac{2}{1-\gamma} \left\|\hat{\mathbf{Q}}_{\pi_k^t, p_k}^\alpha - \tilde{\mathbf{Q}}_{\pi_k^t, p_k}^\alpha\right\|_\infty \qquad \text{(KL-divergence is zero before updates)}$$

$$\leq \mathcal{O}(\max\{1, \|\lambda_k\|_1\} + 2\epsilon t_k). \tag{69}$$

Since the unregularised Lagrangian is bounded by $\mathcal{O}(\max\{1, \|\lambda_k\|_1\})$, and again applying Hoeffding's inequality, it follows that $M_{Q,K} = \Theta\left(\log(\delta_k^{-1}) \frac{(\max\{1, \|\lambda_k\|_1\} + \epsilon t_k)^2}{\epsilon^2}\right)$ is sufficient for approximating $\tilde{\mathbf{Q}}_{\pi_k^t, p_k}^\alpha(s, a)$.

With similar reasoning as in Eq. 68, but applied to the augmented regularised Lagrangian, we have

$$|\mathbb{E}[X] - \tilde{\mathbf{Q}}_{\pi_k^t, p_k}^\alpha(s, a)| \leq \sum_{l=N_{Q,k}}^{\infty} \gamma^l \left(\mathcal{O}(\max\{1, \|\lambda_k\|_1\}) + \epsilon t_k\right)$$

$$= \frac{\gamma^{N_{Q,k}} \mathcal{O}(\max\{1, \|\lambda_k\|_1\}) + \epsilon t_k}{1 - \gamma} = \epsilon$$

$$N_{Q,k} = \Theta\left(\log_{1/\gamma}\left(\frac{\mathcal{O}(\max\{1, \|\lambda_k\|_1\})}{\epsilon} + t_k\right)\right)$$

Note that $t_k = \Theta\left(\log(\max\{1, \|\lambda_k\|_1\}/\epsilon)\right) = \mathcal{O}\left(\frac{\max\{1, \|\lambda_k\|_1\}}{\epsilon}\right)$. Therefore $N_{Q,k} = \mathcal{O}\left(\log_{1/\gamma}\left(\frac{\max\{1, \|\lambda_k\|_1\}}{\epsilon}\right)\right)$.

By union bound, with probability at least $1 - \sum_{k=0}^{K-1} t_k \delta_k = 1 - \delta$, the statement holds for iteration $K - 1$. $\quad\square$

## D.6 Sample complexity

### D.6.1 Supporting lemmas

The following lemma provides a constant upper bound on the cumulative regret difference induced by the LTMA update under general uncertainty sets (including non-rectangular uncertainty sets), where constant

indicates that it is independent of the number of iterations. With additional $\ell_q$ norm bound assumptions on the uncertainty set, the constant can be reduced in rectangular and non-rectangular uncertainty sets. The improvement factor on the upper bound is $\frac{2S^{1-c(q)}}{\Delta_p}$ for rectangular $\ell_q$ sets and $\frac{2(SA^2)^{1-c(q)}}{\Delta_p}$ for non-rectangular $\ell_q$ sets, where $c(q) = 1 - 1/q$ and $\Delta_p$ is the uncertainty budget (upper bound).

**Lemma 24** (Analysis of the cumulative regret difference). *Let $\pi^* \in \arg\min_{\pi \in \Pi} \Phi(\pi)$, and define $D \geq \sum_{k=0}^{K-1} V_{\pi_{k+1}, p_{k+1}}(\rho) - V_{\pi^*, p_{k+1}}(\rho) - \left( V_{\pi_{k+1}, p_k}(\rho) - V_{\pi^*, p_k}(\rho) \right)$ as the difference in cumulative regret induced by the transition kernel updates in Algorithm 3.*
*a) For any uncertainty set $\mathcal{P}$, $D \leq \frac{4}{(1-\gamma)^2} AS^2$.*
*b) Let $\mathcal{P}$ be an uncertainty set contained in an $\ell_q$ $(s,a)$-rectangular uncertainty set around the nominal $\bar{p} = p_0$ such that*

$$\mathcal{P}_{s,a} \subseteq \left\{ p \in \Delta(\mathcal{S}) : \|p(\cdot|s,a) - \bar{p}(\cdot|s,a)\|_q \leq \psi_{s,a} \right\} \quad \forall (s,a) \in \mathcal{S} \times \mathcal{A}, \tag{70}$$

*for some $q \geq 1$, where $0 < \Delta_p = \max_{s,a} \psi_{s,a}$ is the maximal budget. Then $D \leq \frac{2}{(1-\gamma)^2} AS^{1+c(q)} \Delta_p$ where $c(q) = 1 - 1/q$.*
*c) Let $\mathcal{P}$ be a non-rectangular $\ell_q$ uncertainty set around the nominal $\bar{p} = p_0$ such that*

$$\mathcal{P} = \left\{ p : \|p - \bar{p}\|_q \leq \Delta_p, \sum_{s' \in \mathcal{S}} p(s'|s,a) = 1 \quad \forall (s,a) \in \mathcal{S} \times \mathcal{A} \right\} \tag{71}$$

*for some $q \geq 1$, where $0 < \Delta_p$ is the budget. Then $D \leq \frac{2}{(1-\gamma)^2} (AS^2)^{c(q)} \Delta_p$, where $c(q) = 1 - 1/q$.*

*Proof.* Note that

$$\sum_{k=0}^{K-1} V_{\pi_{k+1}, p_{k+1}}(\rho) - V_{\pi^*, p_{k+1}}(\rho) - \left( V_{\pi_{k+1}, p_k}(\rho) - V_{\pi^*, p_k}(\rho) \right)$$

$$= \sum_{k=0}^{K-1} V_{\pi_{k+1}, p_{k+1}}(\rho) - V_{\pi_{k+1}, p_k}(\rho) + V_{\pi^*, p_k}(\rho) - V_{\pi^*, p_{k+1}}(\rho) \qquad \text{(rearranging)}$$

$$= \sum_{k=0}^{K-1} \frac{1}{1-\gamma} \sum_{s \in \mathcal{S}} d_\rho^{\pi_{k+1}, p_{k+1}}(s) \sum_{a \in \mathcal{A}} \pi(a|s) \sum_{s' \in \mathcal{S}} (p_{k+1}(s'|s,a) - p_k(s'|s,a)) \left[ c(s,a,s') + \gamma V_{\pi_{k+1}, p_k}(s') \right]$$

$$+ \frac{1}{1-\gamma} \sum_{s \in \mathcal{S}} d_\rho^{\pi^*, p_k}(s) \sum_{a \in \mathcal{A}} \pi^*(a|s) \sum_{s' \in \mathcal{S}} (p_k(s'|s,a) - p_{k+1}(s'|s,a)) \left[ c(s,a,s') + \gamma V_{\pi^*, p_{k+1}}(s') \right]$$

$$\text{(via Lemma 23)}$$

$$\leq \frac{1}{(1-\gamma)^2} \left| \sum_{k=0}^{K-1} \sum_{s \in \mathcal{S}} d_\rho^{\pi_{k+1}, p_{k+1}}(s) \sum_{a \in \mathcal{A}} \pi(a|s) \sum_{s' \in \mathcal{S}} (p_{k+1}(s'|s,a) - p_k(s'|s,a)) \right|$$

$$+ \frac{1}{(1-\gamma)^2} \left| \sum_{k=0}^{K-1} \sum_{s \in \mathcal{S}} d_\rho^{\pi^*, p_k}(s) \sum_{a \in \mathcal{A}} \pi^*(a|s) \sum_{s' \in \mathcal{S}} (p_k(s'|s,a) - p_{k+1}(s'|s,a)) \right|$$

$$\text{(Cauchy-Schwarz with the maximum value } \tfrac{1}{1-\gamma})$$

$$\leq \frac{1}{(1-\gamma)^2} \left| \sum_{k=0}^{K-1} \sum_{s \in \mathcal{S}} \sum_{a \in \mathcal{A}} \sum_{s' \in \mathcal{S}} (p_{k+1}(s'|s,a) - p_k(s'|s,a)) \right|$$

$$+ \frac{1}{(1-\gamma)^2} \left| \sum_{k=0}^{K-1} \sum_{s \in \mathcal{S}} \sum_{a \in \mathcal{A}} \sum_{s' \in \mathcal{S}} (p_k(s'|s,a) - p_{k+1}(s'|s,a)) \right| \qquad \text{(bound on probability)}$$

$$\leq \frac{1}{(1-\gamma)^2} \left| \sum_{s \in \mathcal{S}} \sum_{a \in \mathcal{A}} \sum_{s' \in \mathcal{S}} (p_K(s'|s,a) - p_0(s'|s,a)) \right| + \frac{1}{(1-\gamma)^2} \left| \sum_{s \in \mathcal{S}} \sum_{a \in \mathcal{A}} \sum_{s' \in \mathcal{S}} (p_0(s'|s,a) - p_K(s'|s,a)) \right|$$

$$\text{(telescoping)}$$

$$\leq \frac{1}{(1-\gamma)^2} SA \max_{s \in \mathcal{S}, a \in \mathcal{A}} (\|p_K(\cdot|s,a) - p_0(\cdot|s,a)\|_1 + \|p_0(\cdot|s,a) - p_K(\cdot|s,a)\|_1) \qquad \text{(definition of norm)}$$

$$\leq \frac{2}{(1-\gamma)^2} SA \max_{s \in \mathcal{S}, a \in \mathcal{A}} \|p_K(\cdot|s,a) - p_0(\cdot|s,a)\|_1 \qquad \text{(symmetry of the norm distance)}$$

For case **a)**, we have

$$\frac{2}{(1-\gamma)^2} SA \max_{s \in \mathcal{S}, a \in \mathcal{A}} \|p_K(\cdot|s,a) - p_0(\cdot|s,a)\|_1$$
$$\leq \frac{4}{(1-\gamma)^2} AS^2 \,.$$

For case **b)**, we have

$$\frac{2}{(1-\gamma)^2} SA \max_{s \in \mathcal{S}, a \in \mathcal{A}} \|p_K(\cdot|s,a) - p_0(\cdot|s,a)\|_1$$
$$\leq \frac{2}{(1-\gamma)^2} SAS^{c(q)} \max_{s \in \mathcal{S}, a \in \mathcal{A}} \|p_K(\cdot|s,a) - p_0(\cdot|s,a)\|_q$$
$$\leq \frac{2}{(1-\gamma)^2} AS^{1+c(q)} \Delta_p \,, \qquad \text{(definition of } \Delta_p\text{)}$$

where the second inequality follows via Hölder's inequality, since for any $X \in \mathbb{R}^d$, $\|X\|_1 = \|\mathbf{1} \cdot X\|_1 \leq d^{1/p} \|X\|_q$ for $p = (1 - 1/q)^{-1} = c(q)^{-1}$.
For case **c)**, we have

$$\frac{1}{(1-\gamma)^2} \left| \sum_{s \in \mathcal{S}} \sum_{a \in \mathcal{A}} \sum_{s' \in \mathcal{S}} (p_K(s'|s,a) - p_0(s'|s,a)) \right| + \frac{1}{(1-\gamma)^2} \left| \sum_{s \in \mathcal{S}} \sum_{a \in \mathcal{A}} \sum_{s' \in \mathcal{S}} (p_0(s'|s,a) - p_K(s'|s,a)) \right|$$

$$\leq \frac{2}{(1-\gamma)^2} \|p_K - p_0\|_1$$
$$\leq \frac{2}{(1-\gamma)^2} (AS^2)^{c(q)} \|p_K - p_0\|_q$$
$$\leq \frac{2}{(1-\gamma)^2} (AS^2)^{c(q)} \Delta_p \,. \qquad \text{(definition of } \Delta_p\text{)}$$

where the second inequality again makes use of Hölder's inequality analogous to part **b)**. $\qquad \square$

Since the difference of Bregman divergences in the RHS of Lemma 10 has a mismatch in the transition kernel index, its sum cannot be bounded by traditional telescoping. The lemma below provides a bound based on an alternative technique.

**Lemma 25** (Analysis of the Bregman term). *Let $\alpha > 0$ be the Bregman penalty parameter, let $\pi^*$ be the optimal deterministic policy for the RCMDP, let $\pi_0 \in \Pi$ be the uniform random policy, and define $U > 0$ according to Eq. 22. Then under Algorithm 3, the following equation holds:*

$$\sum_{k=1}^{K} \frac{\alpha}{1-\gamma} \left( B_{d_\rho^{\pi^*, p_{k-1}}}(\pi^*, \pi_{k-1}) - B_{d_\rho^{\pi^*, p_{k-1}}}(\pi^*, \pi_k) \right) \leq \frac{\alpha}{1-\gamma} \left( \log(A) + 2 \log \left( \frac{1}{U} \right) \right) \,.$$

*Proof.* Note that

$$\sum_{k=1}^{K} \frac{\alpha}{1-\gamma} \left( B_{d_\rho^{\pi^*,p_{k-1}}}(\pi^*, \pi_{k-1}) - B_{d_\rho^{\pi^*,p_{k-1}}}(\pi^*, \pi_k) \right) \tag{72}$$

$$\leq \frac{\alpha}{1-\gamma} \left( B_{d_\rho^{\pi^*,p_0}}(\pi^*, \pi_0) + \sum_{k=1}^{K-1} B_{d_\rho^{\pi^*,p_k}}(\pi^*, \pi_k) - B_{d_\rho^{\pi^*,p_{k-1}}}(\pi^*, \pi_k) \right)$$

$$\text{(drop negative term } -\alpha B_{d_\rho^{\pi,p_{K-1}}}(\pi^*, \pi_K) \text{ and rearrange)}$$

$$\leq \frac{\alpha}{1-\gamma} \left( \log(A) + \sum_{k=1}^{K-1} B_{d_\rho^{\pi^*,p_k}}(\pi^*, \pi_k) - B_{d_\rho^{\pi^*,p_{k-1}}}(\pi^*, \pi_k) \right). \qquad \text{(by Lemma 18)}$$

The remaining summation in the above is at most $2\log\left(\frac{1}{U}\right)$:

$$\sum_{k=1}^{K-1} \left( B_{d_\rho^{\pi^*,p_k}}(\pi^*, \pi_k) - B_{d_\rho^{\pi^*,p_{k-1}}}(\pi^*, \pi_k) \right)$$

$$= \sum_{k=1}^{K-1} \sum_{s\in\mathcal{S}} \left( d_\rho^{\pi^*,p_k}(s) - d_\rho^{\pi^*,p_{k-1}}(s) \right) \sum_{a\in\mathcal{A}} \pi^*(a|s) \log\left( \frac{\pi^*(a|s)}{\pi_k(a|s)} \right)$$

$$= \sum_{k=1}^{K-1} \sum_{s\in\mathcal{S}} \left( d_\rho^{\pi^*,p_k}(s) - d_\rho^{\pi^*,p_{k-1}}(s) \right) \log\left( \frac{\pi^*(a^*(s)|s)}{\pi_k(a^*(s)|s)} \right) \qquad (\pi^* \text{ is deterministic})$$

$$\leq \log\left(\frac{1}{U}\right) \left| \sum_{k=1}^{K-1} \sum_{s\in\mathcal{S}} \left( d_\rho^{\pi^*,p_k}(s) - d_\rho^{\pi^*,p_{k-1}}(s) \right) \right| \qquad (\text{via Eq. 23, } U \leq \inf_{s\in\mathcal{S},k\geq 1} \pi_k(a^*(s)|s))$$

$$= \log\left(\frac{1}{U}\right) \left| \sum_{s\in\mathcal{S}} \left( d_\rho^{\pi^*,p_{K-1}}(s) - d_\rho^{\pi^*,p_0}(s) \right) \right| \qquad (\text{telescoping})$$

$$\leq 2\log\left(\frac{1}{U}\right).$$

$\square$

The proof that the optimal dual variable is bounded by a constant, potentially much lower than $B_\lambda$, is given in the following lemma.

**Lemma 26** (Lemma 16 of Liu et al. (2021b) rephrased). *Let $\zeta > 0$ be a slack variable for a CMDP with transition dynamics $p \in \mathcal{P}$, and let $(\pi^*, \lambda^*)$ be its optimal solution. Then*

$$\|\lambda^*\| \leq \frac{2}{\zeta(1-\gamma)} \tag{73}$$

*Proof.* Following Assumption 4, there exists some policy $\bar{\pi}$ such that

$$V_{\pi^*,p}(\rho) = V_{\pi^*,p}(\rho) + \sum_{j=1}^{m} \lambda^* V_{\pi^*,p}^j(\rho) \leq V_{\bar{\pi},p}(\rho) + \sum_{j=1}^{m} \lambda^* V_{\bar{\pi},p}^j(\rho) \leq V_{\bar{\pi},p}(\rho) - \zeta \sum_{i=1}^{m} \lambda_i^*,$$

such that $\|\lambda^*\| \leq \|\lambda^*\|_1 \leq \frac{V_{\bar{\pi},p}(\rho) - V_{\pi^*,p}(\rho)}{\zeta} \leq \frac{2}{\zeta(1-\gamma)}$. $\square$

The lemma below provides a way to bound the ratio between the value of two transition kernels.

**Lemma 27** (Performance ratio of transition kernels). *Let $\pi \in \Pi$ and $p, p' \in \mathcal{P}$, and let $V$ be a value function. Moreover, let $M$ upper bound the mismatch coefficient as in Eq. 21. Then it follows that*

$$V_{\pi,p}(\rho) \leq \frac{M}{1-\gamma} V_{\pi,p'}(\rho). \tag{74}$$

*Proof.* For any $\pi \in \Pi, p \in \mathcal{P}$, we have $d_\rho^{\pi,p}(s) \geq (1-\gamma)\rho(s)$, such that using derivations as in Theorem 4.2 of Wang et al. (2024),

$$\frac{M}{1-\gamma} \geq \frac{1}{1-\gamma} \left\| \frac{d_\rho^{\pi,p}}{\rho} \right\|_\infty \geq \left\| \frac{d_\rho^{\pi,p}}{d_\rho^{\pi_k,p'}} \right\|_\infty \geq \frac{d_\rho^{\pi,p}(s)}{d_\rho^{\pi,p'}(s)}. \tag{75}$$

Therefore,

$$
\begin{aligned}
V_{\pi,p}(\rho) &= \frac{1}{1-\gamma} \sum_{s \in \mathcal{S}} d_\rho^{\pi,p}(s) \sum_{a \in \mathcal{A}} \pi(a|s)\, c(s,a) && \text{(definition of value)} \\
&= \frac{1}{1-\gamma} \sum_{s \in \mathcal{S}} d_\rho^{\pi,p'}(s) \frac{d_\rho^{\pi,p}(s)}{d_\rho^{\pi,p'}(s)} \sum_{a \in \mathcal{A}} \pi(a|s)\, c(s,a) && \text{(division and multiplication by the same term)} \\
&\leq \frac{M}{1-\gamma} V_{\pi,p'}(\rho). && \text{(Eq. 75 and definition of value)}
\end{aligned}
$$

$\square$

### D.6.2 Main theorem

The full derivation of Theorem 2 is given below.

*Proof.* **(a)** The proof uses similar derivations as in Theorem 3 of Liu et al. (2021b) and then accounts for the performance difference due to transition kernels.

Note that

$$
\begin{aligned}
& \tilde{\mathbf{V}}_{\pi_{k+1},p_k}^\alpha(\rho) \\
&= V_{\pi_{k+1},p_k}(\rho) + \left\langle \lambda_k + \eta_\lambda \hat{V}_{\pi_k,p_k}^{1:m}(\rho), V_{\pi_{k+1},p_k}^{1:m}(\rho) \right\rangle + \frac{\alpha}{1-\gamma} B_{d_\rho^{\pi_{k+1},p_k}}(\pi_{k+1}, \pi_k) \\
&\leq V_{\pi^*,p_k}(\rho) + \left( \frac{\alpha}{1-\gamma} \left( B_{d_\rho^{\pi^*,p_k}}(\pi^*, \pi_k) - B_{d_\rho^{\pi^*,p_k}}(\pi^*, \pi_{k+1}) \right) + \Theta(\epsilon) \right),
\end{aligned}
$$

where the last step follows from setting $\pi = \pi^*$ in Lemma 10 and noting that the inner product will be negative, since $\lambda_{k,j} + \eta_\lambda \hat{V}_{\pi_k,p_k}^j(\rho) \geq 0$ by property 2 of Lemma 14 and $V_{\pi^*,p_k}^j(\rho) \leq 0$ for any $j \in [m]$.

It then follows that

$$
\begin{aligned}
V_{\pi_{k+1},p_k}(\rho) - V_{\pi^*,p_k}(\rho) &\leq \frac{\alpha}{1-\gamma} \left( B_{d_\rho^{\pi^*,p_k}}(\pi^*, \pi_k) - B_{d_\rho^{\pi^*,p_k}}(\pi^*, \pi_{k+1}) \right) + \Theta(\epsilon) \\
&\quad - \left\langle \lambda_k + \eta_\lambda \hat{V}_{\pi_k,p_k}^{1:m}(\rho), V_{\pi_{k+1},p_k}^{1:m}(\rho) \right\rangle - \frac{\alpha}{1-\gamma} B_{d_\rho^{\pi_{k+1},p_k}}(\pi_{k+1}, \pi_k).
\end{aligned}
$$

Filling in the lower bound on the inner product from Eq. 41 from Liu et al. (2021b) (see also Lemma 15),

$$
\begin{aligned}
& \left\langle \lambda_k + \eta_\lambda \hat{V}_{\pi_k,p_k}^{1:m}(\rho), V_{\pi_{k+1},p_k}^{1:m}(\rho) \right\rangle \geq \frac{1}{2\eta_\lambda} \left( \|\lambda_{k+1}\|^2 - \|\lambda_k\|^2 \right) + \frac{\eta_\lambda}{2} \left( \left\| V_{\pi_k,p_k}^{1:m}(\rho) \right\|^2 - \left\| V_{\pi_{k+1},p_k}^{1:m}(\rho) \right\|^2 \right) \\
& - 2\eta_\lambda \left\langle V_{\pi_{k+1},p_k}^{1:m}(\rho), \epsilon_{k+1} \right\rangle - \frac{\gamma^2 \eta_\lambda}{(1-\gamma)^4} B_{d_\rho^{\pi_{k+1},p_k}}(\pi_{k+1}, \pi_k),
\end{aligned}
$$

we get

$$V_{\pi_{k+1},p_k}(\rho) - V_{\pi^*,p_k}(\rho)$$

$$\leq \frac{\alpha}{1-\gamma}\left(B_{d_\rho^{\pi^*,p_k}}(\pi^*,\pi_k) - B_{d_\rho^{\pi^*,p_k}}(\pi^*,\pi_{k+1})\right) + \Theta(\epsilon) + \frac{1}{2\eta_\lambda}\left(\|\lambda_k\|^2 - \|\lambda_{k+1}\|^2\right) + \frac{\eta_\lambda}{2}\left(\left\|V_{\pi_{k+1},p_k}^{1:m}(\rho)\right\|^2 - \left\|V_{\pi_k,p_k}^{1:m}(\rho)\right\|^2\right)$$

$$+ 2\eta_\lambda\left\langle V_{\pi_{k+1},p_k}^{1:m}(\rho), \epsilon_{k+1}\right\rangle - \frac{\alpha(1-\gamma)^3 - \gamma^2\eta_\lambda}{(1-\gamma)^4}B_{d_\rho^{\pi_{k+1},p_k}}(\pi_{k+1},\pi_k)$$

$$\leq \frac{\alpha}{1-\gamma}\left(B_{d_\rho^{\pi^*,p_k}}(\pi^*,\pi_k) - B_{d_\rho^{\pi^*,p_k}}(\pi^*,\pi_{k+1})\right) + \Theta(\epsilon) + \frac{1}{2\eta_\lambda}\left(\|\lambda_k\|^2 - \|\lambda_{k+1}\|^2\right) + \frac{\eta_\lambda}{2}\left(\left\|V_{\pi_{k+1},p_k}^{1:m}(\rho)\right\|^2 - \left\|V_{\pi_k,p_k}^{1:m}(\rho)\right\|^2\right)$$

$$+ 2\eta_\lambda\left\langle V_{\pi_{k+1},p_k}^{1:m}(\rho), \epsilon_{k+1}\right\rangle . \tag{76}$$

where the last step follows because due to the parameter settings the term $\frac{\alpha(1-\gamma)^3-\gamma^2\eta_\lambda}{(1-\gamma)^4}B_{d_\rho^{\pi_{k+1},p_k}}(\pi_{k+1},\pi_k) \geq 0$.

Due to the approximations and the above, the regret introduces a term

$$\Delta_k = \Theta(\epsilon) + \langle\lambda_{k-1},\epsilon_k\rangle - \eta_\lambda\left\langle\epsilon_{k-1}, V_{\pi_k,p_{k-1}}^{1:m}(\rho)\right\rangle + \eta_\lambda\left\langle\epsilon_k + 2V_{\pi_k,p_{k-1}}^{1:m}(\rho), \epsilon_k\right\rangle \tag{77}$$

for each $k \in [K]$.

It follows that

$$\sum_{k=1}^K (V_{\pi_k,p_k}(\rho) - V_{\pi^*,p_k}(\rho)) \tag{78}$$

$$= \sum_{k=1}^K \left(V_{\pi_k,p_k}(\rho) - V_{\pi^*,p_k}(\rho) - V_{\pi_k,p_{k-1}}(\rho) - V_{\pi^*,p_{k-1}}(\rho)\right) + V_{\pi_k,p_{k-1}}(\rho) - V_{\pi^*,p_{k-1}}(\rho) \qquad \text{(decomposing)}$$

$$\leq \mathcal{O}(1) + \sum_{k=1}^K \left(\frac{\alpha}{1-\gamma}\left(B_{d_\rho^{\pi^*,p_{k-1}}}(\pi^*,\pi_{k-1}) - B_{d_\rho^{\pi^*,p_{k-1}}}(\pi^*,\pi_k)\right) + \Delta_k\right.$$

$$\left. + \frac{\eta_\lambda}{2}\left(\left\|V_{\pi_k,p_{k-1}}^{1:m}(\rho)\right\|^2 - \left\|V_{\pi_{k-1},p_{k-1}}^{1:m}(\rho)\right\|^2\right) + \frac{1}{2\eta_\lambda}\left(\|\lambda_{k-1}\|^2 - \|\lambda_k\|^2\right)\right)$$

$$\text{(Lemma 24 and derivations above)}$$

$$\leq \mathcal{O}(1) + \frac{\alpha}{1-\gamma}\left(\log(A) + 2\log\left(\frac{1}{U}\right)\right) + \sum_{l=1}^K \Delta_k$$

$$+ \frac{\eta_\lambda}{2}\left(\left\|V_{\pi_K,p_{K-1}}^{1:m}(\rho)\right\|^2 - \left\|V_{\pi_0,p_0}^{1:m}(\rho)\right\|^2\right) + \frac{1}{2\eta_\lambda}\left(\|\lambda_0\|^2 - \|\lambda_K\|^2\right) \qquad \text{(telescoping and Lemma 25)}$$

$$\leq \mathcal{O}(1) + \frac{\eta_\lambda}{2}\left\|V_{\pi_K,p_{K-1}}^{1:m}(\rho)\right\|^2 + \frac{1}{2\eta_\lambda}\|\lambda_0\|^2 + \sum_{k=1}^K \Delta_k \qquad \text{(dropping negative terms)}$$

$$\leq \mathcal{O}(1) + \frac{\eta_\lambda m}{(1-\gamma)^2} + \sum_{k=1}^K \Delta_k ,$$

where the last step follows from the fact that $\|\lambda_0\|^2 \leq \frac{m\eta_\lambda^2}{(1-\gamma)^2}$ and $\|V_{\pi_K,p_K}(\rho)\|^2 \leq \frac{m}{(1-\gamma)^2}$. Note further that $\frac{\eta_\lambda m}{(1-\gamma)^2} = \mathcal{O}(1)$. Moreover, note from Eq. 77 that $\Delta_k = \mathcal{O}(\epsilon\max\{1,\|\lambda_{k-1}\|_1\})$. Lemma 28 shows that with probability $1-\delta$, $\|\lambda_k\| \leq \|\lambda_k\|_1 = \mathcal{O}(1)$, such that $\Delta_k = \mathcal{O}(\epsilon)$ and $\sum_{k=1}^K \Delta_k = \mathcal{O}(1)$. All terms then reduce to $\mathcal{O}(\epsilon)$ after division by $K = \Theta(\frac{1}{\epsilon})$.

**(b)** Denoting the approximation error at iteration $k \in [K]$ and constraint $j$ as $\epsilon_{k,j}$, and observing that for

any $j \in [m]$

$$\lambda_{k,j} = \max\left\{-\eta_\lambda \hat{V}^j_{\pi_k,p_k}(\rho), \lambda_{k-1,j} + \eta_\lambda \hat{V}^j_{\pi_k,p_k}(\rho)\right\}$$
$$\geq \lambda_{k-1,j} + \eta_\lambda \hat{V}^j_{\pi_{\theta_k},p_k}(\rho), \tag{79}$$

it follows that

$$\frac{1}{K}\sum_{k=1}^{K} V^j_{\pi_k,p_k}(\rho) = \frac{1}{K}\sum_{k=1}^{K}\left(\hat{V}^j_{\pi_k,p_k}(\rho) - \epsilon_{k,j}\right) \tag{80}$$

$$\leq \frac{1}{K}\sum_{k=1}^{K}\left(\frac{\lambda_{k,j}-\lambda_{k-1,j}}{\eta_\lambda} - \epsilon_{k,j}\right) \qquad \text{(from Eq. 79)}$$

$$\leq \frac{\lambda_{K,j}}{\eta_\lambda K} - \frac{1}{K}\sum_{k=1}^{K}\epsilon_{k,j} \qquad \text{(telescoping and non-negativity condition in Lemma 14)}$$

$$\leq \frac{\|\lambda^*\| + \|\lambda^* - \lambda_K\|}{\eta_\lambda K} - \frac{1}{K}\sum_{k=1}^{K}\epsilon_{k,j} \qquad \text{(since } \lambda_{K,j} \leq \|\lambda_K\| \leq \|\lambda^*\|_+ \|\lambda^* - \lambda_K\|)$$

$$\leq \frac{\|\lambda^*\|_+ \|\lambda^* - \lambda_K\|}{\eta_\lambda K} + \mathcal{O}(\epsilon) \qquad \text{(since by Lemma 9, } K = \Theta\left(\frac{1}{\epsilon}\right) \text{ and } \epsilon_{k,j} = \mathcal{O}(\epsilon))$$

$$= \mathcal{O}(1/K) + \mathcal{O}(\epsilon) \qquad \text{(by Lemma 28)}$$

$$= \mathcal{O}(\epsilon). \qquad \text{(since } K = \Theta\left(\frac{1}{\epsilon}\right))$$

**c)** For any $k \in [K]$, it follows from Lemma 2 that

$$t'_k = \Theta\left(\frac{F_\lambda M}{\epsilon'_k(1-\gamma)}\right)$$

is sufficient to obtain an $\epsilon'_k$-optimal transition kernel from LTMA. Note then that via Lemma 29

$$\frac{1}{K}\sum_{k\in[K]}\Delta_{\lambda_k} = \mathcal{O}(\epsilon).$$

Using Lemma 8, combining the errors from **a)** and **b)** for the cost and constraint-costs, and $\epsilon'_k = \Theta(\epsilon)$, it follows that

$$\frac{1}{K}\sum_{k=1}^{K}(\Phi(\pi_k) - \Phi(\pi^*)) \leq \frac{1}{K}\sum_{k=1}^{K}\left(\mathbf{V}_{\pi_k,p_k}(\rho;\lambda_{k-1}) + \epsilon'_k + \Delta_{\lambda_k} - \mathbf{V}_{\pi^*,p_k;\lambda_{k-1}}(\rho)\right)$$
$$= \mathcal{O}(F_\lambda \epsilon),$$

where $F_\lambda = \max_{k\in[K]} F_{\lambda_k}$. Due to the results in Lemma 28, $\lambda_k = \mathcal{O}(1)$ and $F_\lambda = \mathcal{O}(1)$. Moreover $\eta_\lambda V^j_{\pi_k,p_k} \leq \frac{1}{1-\gamma} = \mathcal{O}(1)$ for all $j \in [m]$ and all $k \in [K]$.

**d)** The settings of Lemma 7 guarantee an $\epsilon'_k$-optimal transition kernel from LTMA. Following steps as in part **c)**, it follows that

$$\frac{1}{K}\sum_{k=1}^{K}(\Phi(\pi_k) - \Phi(\pi^*)) \leq \frac{1}{K}\sum_{k=1}^{K}(\mathbf{V}_{\pi_k,p_k}(\rho;\lambda_{k-1}) + \epsilon'_k + \Delta_{\lambda_k} - \mathbf{V}_{\pi^*,p_k}(\rho;\lambda_{k-1})) = \mathcal{O}(\epsilon),$$

where the factor $F_\lambda$ does not appear since it is already accounted for in the parameter settings.

**Sample complexity:** Following the loop in Algorithm 3, plugging in the settings from Lemma 9 and Lemma 7, and omitting logarithmic factors and constants, the total number of calls to the generative model

is given by

$$T = \sum_{k=1}^{K} \left( M_{V,k} N_{V,k} + \sum_{t=0}^{t_k-1} M_{Q,k} N_{Q,k} + \sum_{t=0}^{t'_k-1} M_{G,t} N_{G,t} \right)$$
$$= \mathcal{O}(\epsilon^{-1}) \left( \epsilon^{-2} \tilde{\mathcal{O}}(1) + \tilde{\mathcal{O}}(1) \epsilon^{-2} \tilde{\mathcal{O}}(1) + \tilde{\mathcal{O}}\left(\epsilon^{-2}\right) \right)$$
$$= \tilde{\mathcal{O}}(\epsilon^{-3}).$$

where the last step follows from $\epsilon'^{-1} = \mathcal{O}(\epsilon^{-1})$. □

The proof that $\|\lambda_k\|_1 = \mathcal{O}(1)$ for all $k \in [K]$ (independent of Assumption 2) is given below.

**Lemma 28.** *Let $K' \geq 1$ be a macro-iteration. Under the event that $|V^i_{\pi_{K'-1}, p_{K'-1}}(\rho) - V^i_{\pi_{K'-1}, p_{K'-1}}(\rho)| \leq \epsilon$ for all $i \in [m]$, it follows that $\|\lambda_{K'}\|_1 = \mathcal{O}(1)$.*

*Proof.* Consider the Lagrangian with optimal parameters $(\pi^*, \lambda^*, p^*)$. Note that

$$K' V_{\pi^*, p^*}(\rho; \lambda^*) = K' \mathbf{V}_{\pi^*, p^*}(\rho; \lambda^*) \qquad \text{(complementary slackness)}$$
$$\leq \sum_{k=1}^{K'} \mathbf{V}_{\pi_k, p^*}(\rho; \lambda^*)$$
$$= \sum_{k=1}^{K'} V_{\pi_k, p^*}(\rho) + \sum_{j=1}^{m} \lambda_j^* V^j_{\pi_k, p^*}(\rho)$$
$$\leq \sum_{k=1}^{K'} V_{\pi_k, p^*_{k-1}}(\rho) + \sum_{j=1}^{m} \lambda_j^* V^j_{\pi_k, p^*_{k-1}}(\rho) \qquad (p^*_{k-1} \text{ maximises the Lagrangian for } \pi_k)$$
$$\leq \frac{M}{1-\gamma} \left( \sum_{k=1}^{K'} V_{\pi_k, p_k}(\rho) + \sum_{j=1}^{m} \lambda_j^* V^j_{\pi_k, p_k}(\rho) \right) \qquad \text{(Lemma 27)}$$
$$\leq \frac{M}{1-\gamma} \left( \mathcal{O}(1) + \sum_{k=1}^{K'} V_{\pi_k, p_k}(\rho) + \sum_{j=1}^{m} \lambda_j^* \left( \frac{1}{\eta_\lambda} \lambda_{K',j} - \sum_{k=1}^{K'} \epsilon_{k,j} \right) \right) \qquad \text{(derivations in Eq. 80)}$$
$$\leq \frac{M}{1-\gamma} \left( \mathcal{O}(1) + \sum_{j=1}^{m} \lambda_j^* \frac{1}{\eta_\lambda} \lambda_{K',j} - \sum_{k=1}^{K} \sum_{j=1}^{m} \lambda_j^* \epsilon_{k,j} \right.$$
$$+ \sum_{k=1}^{K'} \frac{\alpha}{1-\gamma} \left( B_{d_\rho^{\pi^*, p_{k-1}}}(\pi^*, \pi_{k-1}) - B_{d_\rho^{\pi^*, p_{k-1}}}(\pi^*, \pi_k) \right) + \Delta_k$$
$$\left. + \frac{\eta_\lambda}{2} \left( \left\| V^{1:m}_{\pi_k, p_{k-1}}(\rho) \right\|^2 - \left\| V^{1:m}_{\pi_{k-1}, p_{k-1}}(\rho) \right\|^2 \right) + \frac{1}{2\eta_\lambda} \left( \|\lambda_{k-1}\|^2 - \|\lambda_k\|^2 \right) \right). \qquad \text{(following Eq. 78)}$$

Applying Lemma 26 to $p^*$, it follows that $\|\lambda^*\| = \mathcal{O}(1)$ such that $\sum_{j=1}^{m} \lambda_j^* \mathcal{O}(\epsilon) = \mathcal{O}(\epsilon)$. Note then that via Lemma 25, $\frac{\alpha}{1-\gamma} \sum_{k \in [K]} \left( B_{d_\rho^{\pi^*, p_{k-1}}}(\pi^*, \pi_{k-1}) - B_{d_\rho^{\pi^*, p_{k-1}}}(\pi^*, \pi_k) \right) \leq \frac{\alpha}{1-\gamma} \left( \log(A) + 2 \log(\frac{1}{U}) \right)$. Moreover, the preconditions for $\alpha$ in the further derivations in Lemma 21 of Liu et al. (2021b) are satisfied. The remainder of the proof removes the constant $\frac{M}{1-\gamma} = \mathcal{O}(1)$ and then follows the derivations in Lemma 21 of Liu et al. (2021b). □

The lemma below bounds the error induced by the dual variable taking suboptimal values during policy optimisation and LTMA. The result bounds the instantaneous additional error by a constant, and the average error as $\mathcal{O}(\epsilon)$.

**Lemma 29** (Bound on dual variable induced error)**.** *Let $k = 1, \ldots, K$ define a sequence of macro-iterations with the preconditions of Theorem 2. Let $(p_{k-1}^*, \lambda_{k-1}^*)$ be the optimal solutions for $\lambda$ and $p$ in the problem*

$$\max_{\lambda \geq 0} \sup_{p \in \mathcal{P}} \left( V_{\pi_k, p}(\rho) + \sum_{j=1}^{m} \lambda_j V_{\pi_k, p}^j(\rho) \right).$$

*Then the error induced by the dual variable, as defined in Lemma 8 according to $\Delta_{\lambda_k} = \sum_{j=1}^{m} (\lambda_{k-1,j}^* - \lambda_{k-1,j}) V_{\pi_k, p_{k-1}^*}^j(\rho)$ satisfies*

$$\frac{1}{K} \sum_{k=1}^{K} \Delta_{\lambda_k} = \mathcal{O}(\epsilon) \tag{81}$$

*Proof.* Note that

$$
\begin{aligned}
\frac{1}{K} \sum_{k=1}^{K} \Delta_{\lambda_k} &= \frac{1}{K} \sum_{k=1}^{K} \sum_{j=1}^{m} (\lambda_{k-1,j}^* - \lambda_{k-1,j}) V_{\pi_k, p_{k-1}^*}^j(\rho) \\
&\leq \frac{1}{K} \sum_{k=1}^{K} \sum_{j=1}^{m} B_\lambda V_{\pi_k, p_{k-1}^*}^j(\rho) && \text{(non-negativity in Lemma 14 and Assumption 2)} \\
&\leq \frac{M}{1-\gamma} B_\lambda \frac{1}{K} \sum_{k=1}^{K} \sum_{j=1}^{m} V_{\pi_k, p_k}^j(\rho) && \text{(Lemma 27)} \\
&= \mathcal{O}(\epsilon). && \text{(Theorem 2\textbf{b}))}
\end{aligned}
$$

$\square$

### D.6.3 Non-rectangular uncertainty sets

The global performance guarantee of CPI can be expressed in terms of the degree of non-rectangularity (Li et al., 2023). Note that we slightly modify the original lemma by removing the irreducibility assumption since the mismatch coefficient can also be limited using Assumption 3.

**Lemma 30** (Global performance guarantee of CPI, Theorem 3.8 in Li et al. (2023))**.** *Let $\epsilon > 0$, let $\pi \in \Pi$ and let $V : \mathcal{S} \to \mathbb{R}$ be a value function. Then Algorithm 2 returns within $\mathcal{O}(1/\epsilon^2)$ iterations a solution $\hat{p}$ that satisfies*

$$V_{\pi, p^*}(\rho) - V_{\pi, \hat{p}}(\rho) \leq M(2\epsilon + \delta_{\mathcal{P}}), \tag{82}$$

*where $\delta_{\mathcal{P}}$ is the degree of non-rectangularity defined in Eq 44.*

## E   Algorithm implementation details

All the algorithms are implemented in Pytorch. The code for RMCPMD is based on the original implementation, which can be found at `https://github.com/JerrisonWang/JMLR-DRPMD`. The code for PPO is taken from the StableBaselines3 class. Both are modified to fit our purposes by allowing to turn off and on robust training and constraint-satisfaction. Similar to PPO, MDPO is also implemented following StableBaselines3 class conventions. All the environments are implemented in Gymnasium. For simplicity, the updates with LTMA are based on the original TMA code in `https://github.com/JerrisonWang/JMLR-DRPMD`, which performs pure Monte Carlo based TMA rather than additional function approximation as proposed in our Approximate TMA algorithm. For MDPO and PPO experiments, we use four parallel environments with four CPUs. Our longest experiments typically take no more than two hours to complete. The remainder of the section describes domain-specific hyperparameter settings.

### E.1 Hyperparameters for Cartpole experiments

Hyperparameters settings for the Carpole experiments can be found in 8. Settings for robust optimisation, including the policy architecture, transition kernel architecture, and TMA learning rate are taken from Wang et al. (2023) with the exception that we formulate a more challenging uncertainty set with a 5 times larger range. The policy learning rate and GAE lambda is typical for PPO based methods so we apply these for PPO and MDPO methods. The number of policy epochs and early stopping KL target is based for PPO methods on the standard repository for PPO-Lag (`https://github.com/openai/safety-starter-agents/`) and for MDPO it is based on the original paper's settings (Tomar et al., 2022). PPO obtained better results without entropy regularisation and value function clipping on initial experiments, so for simplicity we disabled these for all algorithms. The batch size, multiplier initialisation, and multiplier learning rate and scaling are obtained via a limited tuning procedure. For the tuning, the batch size tried was in $\{100, 200, 400, 1000, 2000\}$ and the multiplier learning rate was set to $\{1e^{-1}, 1e^{-2}, 1e^{-3}, 1e^{-4}\}$ on partial runs. Then the use of a softplus transformation of the multiplier was compared to the direct (linear) multiplier, and for the linear multiplier we tested initialisations in $\{1, 5, 10\}$.

Table 8: Hyperparameter settings for the Cartpole experiments.

| Hyperparameter | Setting |
|---|---|
| Policy architecture | MLP 4 Inputs – Linear(128) – Dropout(0.6) – Linear(128) – Softmax(2) |
| Critic architecture | MLP 4 Inputs – Linear(128) – Dropout(0.6) – ReLU – Linear(1+m) |
| Policy learning rate ($\eta$) | $3e^{-4}$ |
| Policy optimiser | Adam |
| GAE lambda ($\lambda_{\mathrm{GAE}}$) | 0.95 |
| Discount factor ($\gamma$) | 0.99 |
| Batch and minibatch size | PPO/MDPO policy update: $400 \times 4$ time steps per batch, minibatch 32, episode steps at most 100 |
| | PPO/MDPO multiplier update: $100 \times 4$ time steps per batch, minibatch 32, episode steps at most 100 |
| | LTMA: $10 \times$ factor[1] Monte Carlo updates, episode steps at most 10 |
| Policy epochs | MCPMD: 1 |
| | PPO: 50, early stopping with KL target 0.01 |
| | MDPO: 5 |
| Transition kernel architecture | multi-variate Gaussian with parametrised mean in $(1 + \delta)\mu_c(s)$ |
| | where $\delta \in (\pm 0.005, \pm 0.05, \pm 0.005, \pm 0.05)$ |
| | and covariance $\sigma \mathbb{I}$ where $\sigma = 1e^{-7}$ |
| LTMA learning rate ($\eta_\xi$) | $1e^{-7}$ |
| Dual learning rate ($\eta_\lambda$) | $1e^{-3}$ |
| Dual epochs | MCPMD: 1 |
| | PPO: 50 with early stopping based on target-kl 0.01 |
| | MDPO: 5 |
| Multiplier | initialise to 5, linear, clipping to $\lambda_{\max} = 50$ for non-augmented algorithms |

### E.2 Hyperparameters for Inventory Management experiments

Hyperparameters settings for the unidimensional Inventory Management (see Table 9) are similar to the Cartpole experiments. The key differences are the output layer, the discount factor, the transition kernel, and the TMA parameters, which follow settings of the domain in Wang et al. (2024). Based on initial tuning experiments with PPO and MDPO, the learning rates are set to $\eta = 1\mathrm{e}^{-3}$ and $\eta_\lambda = 1\mathrm{e}^{-2}$, a lower GAE lambda of 0.50 was chosen and PPO is implemented without early stopping and with the same number of (5) epochs as MDPO.

Table 9: Hyperparameter settings for Inventory Management experiments.

| Hyperparameter | Setting |
| --- | --- |
| Policy architecture | MLP 1 Input – Linear(64) – Dropout(0.6) – ReLU – Softmax(4) |
| Critic architecture | MLP 3 Inputs – Linear(128) – Dropout(0.6) – ReLU – Linear(1+m) |
| Policy learning rate ($\eta$) | $1\mathrm{e}^{-3}$ |
| Policy optimiser | Adam |
| GAE lambda ($\lambda_{\mathrm{GAE}}$) | 0.50 |
| Discount factor ($\gamma$) | 0.95 |
| Batch and minibatch size | PPO/MDPO policy update: $400 \times 4$ time steps per batch, minibatch 32, episode steps at most 80 |
| | PPO/MDPO multiplier update: $100 \times 4$ time steps per batch, minibatch 32, episode steps at most 80 |
| | LTMA: $20 \times$ factor Monte Carlo updates, episode steps at most 40 |
| Policy epochs | MCPMD: 1 |
| | PPO: 5 |
| | MDPO: 5 |
| Transition kernel architecture | Radial features with Gaussian mixture parametrisation |
| LTMA learning rate ($\eta_\xi$) | $1\mathrm{e}^{-1}$ |
| Dual learning rate ($\eta_\lambda$) | $1\mathrm{e}^{-2}$ |
| Dual epochs | MCPMD: 1 |
| | PPO and MDPO: 5 |
| Multiplier | initialise to 5, linear, clipping to $\lambda_{\max} = 500$ for non-augmented algorithms |

### E.3 Hyperparameters for 3-D Inventory Management experiments

Hyperparameters settings for the 3-D Inventory Management (see Table 9) are the same as in the IM domain with a few exceptions. The policy architecture is now a Gaussian MLP. The standard deviation of the policy is set to the default with Log std init equal to 0.0 albeit after some tuning effort. The optimiser is set to SGD instead of Adam based on improved preliminary results.

Table 10: Hyperparameter settings for Inventory Management experiments.

| Hyperparameter | Setting |
|---|---|
| Policy architecture | Gaussian MLP 3 Inputs – Linear(128) – Dropout(0.6) – ReLU – Linear(128) – ReLU – Linear(3 × 2) |
| Critic architecture | MLP 3 Inputs – Linear(128) – Dropout(0.6) – ReLU – Linear(1+m) |
| Policy learning rate ($\eta$) | $1e^{-3}$ |
| Policy optimiser | SGD |
| Log std init | 0.0 |
| GAE lambda ($\lambda_{\mathrm{GAE}}$) | 0.50 |
| Discount factor ($\gamma$) | 0.95 |
| Batch and minibatch size | PPO/MDPO policy update: 400 × 4 time steps per batch, minibatch 32, episode steps at most 100 |
| | PPO/MDPO multiplier update: 100 × 4 time steps per batch, minibatch 32, episode steps at most 100 |
| | LTMA: 20 × factor Monte Carlo updates, episode steps at most 40 |
| Policy epochs | MCPMD: 1 |
| | PPO: 5 |
| | MDPO: 5 |
| Transition kernel architecture | Radial features with Gaussian mixture parametrisation |
| LTMA learning rate ($\eta_\xi$) | $1e^{-1}$ |
| Dual learning rate ($\eta_\lambda$) | $1e^{-2}$ |
| Dual epochs | MCPMD: 1 |
| | PPO and MDPO: 5 |
| Multiplier | initialise to 5, linear, clipping to $\lambda_{\max} = 500$ for non-augmented algorithms |

## F  Training performance plots

While the experiments with small and large training time steps were run independently, we report here the training development under the large training data regime (i.e. between 200,000 and 400,000 time steps) since this gives a view of both the early and late stages of training.

### F.1  Cartpole

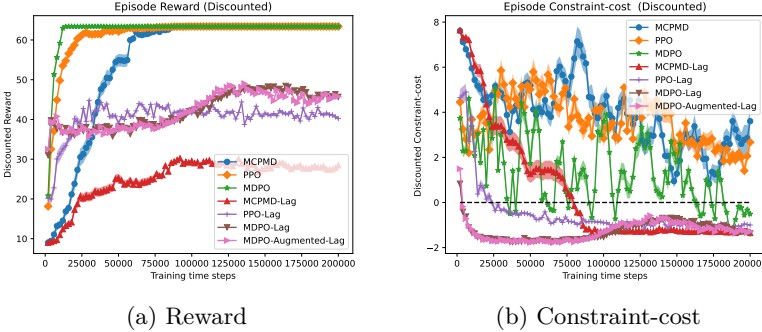

(a) Reward          (b) Constraint-cost

Figure 7: Constrained MDP training development plots in the Cartpole domain, where each sample in the plot is based on 20 evaluations of the deterministic policy. The line and shaded area represent the mean and standard error across 10 seeds.

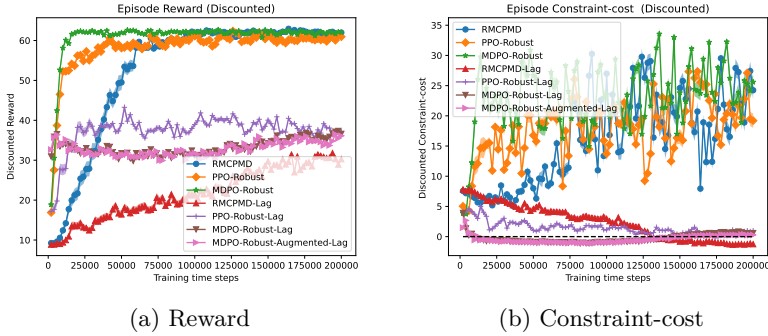

(a) Reward          (b) Constraint-cost

Figure 8: RCMDP training development plots in the Cartpole domain, where each sample in the plot is based on 20 evaluations of the deterministic policy as it interacts with the adversarial environment from that iteration. The line and shaded area represent the mean and standard error across 10 seeds.

## F.2 Inventory Management

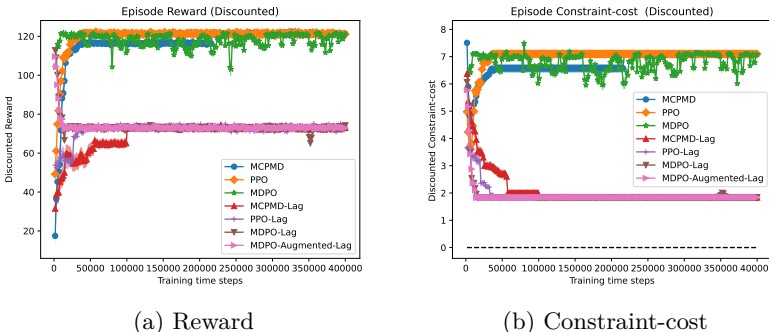

(a) Reward      (b) Constraint-cost

Figure 9: CMDP training development plots in the Inventory Management domain, where each sample in the plot is based on 20 evaluations of the deterministic policy. The line and shaded area represent the mean and standard error across 10 seeds.

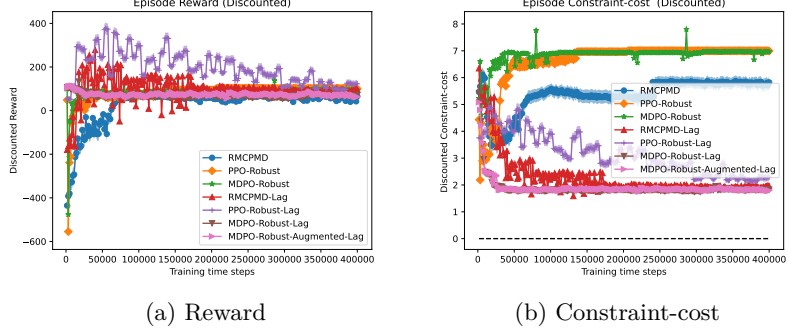

(a) Reward      (b) Constraint-cost

Figure 10: RCMDP training development plots in the Inventory Management domain, where each sample in the plot is based on 20 evaluations of the deterministic policy as it interacts with the adversarial environment from that iteration. The line and shaded area represent the mean and standard error across 10 seeds.

## F.3   3-D Inventory Management

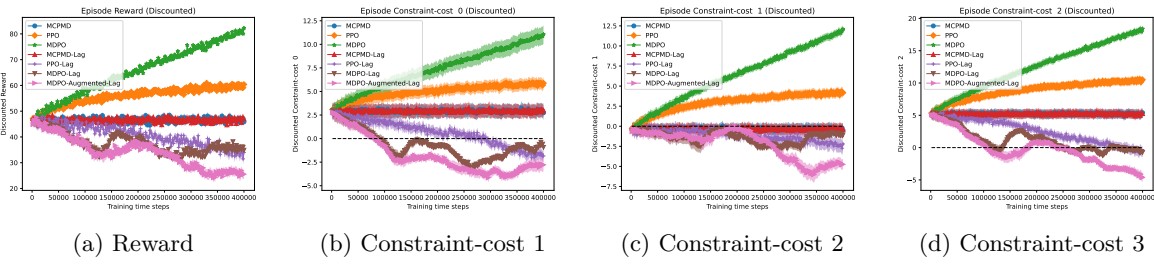

(a) Reward     (b) Constraint-cost 1     (c) Constraint-cost 2     (d) Constraint-cost 3

Figure 11: MDP training development plots in the 3-D Inventory Management domain, where each sample in the plot is based on 20 evaluations of the deterministic policy. The line and shaded area represent the mean and standard error across 10 seeds.

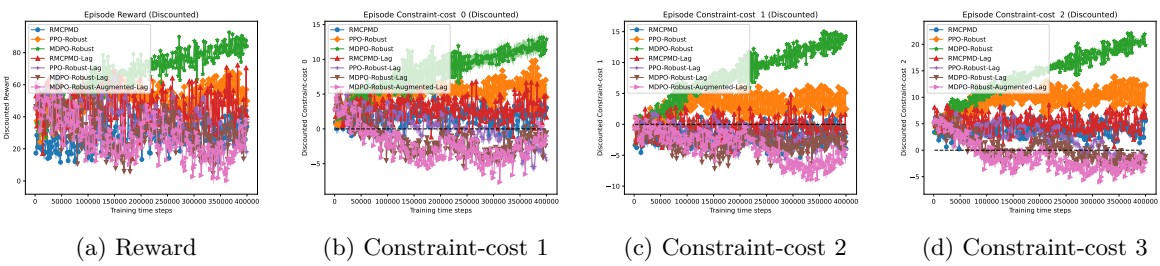

(a) Reward     (b) Constraint-cost 1     (c) Constraint-cost 2     (d) Constraint-cost 3

Figure 12: RCMDP training development plots in the 3-D Inventory Management domain, where each sample in the plot is based on 20 evaluations of the deterministic policy as it interacts with the adversarial environment from that iteration. The line and shaded area represent the mean and standard error across 10 seeds.

# G    Test performance plots with large sample budget

While the main text presents the test performance plots with small sample budgets, between 16,000 and 50,000 time steps, the section below presents the test performance plots after the larger sample budget of 200,000 to 500,000 time steps.

## G.1    Cartpole

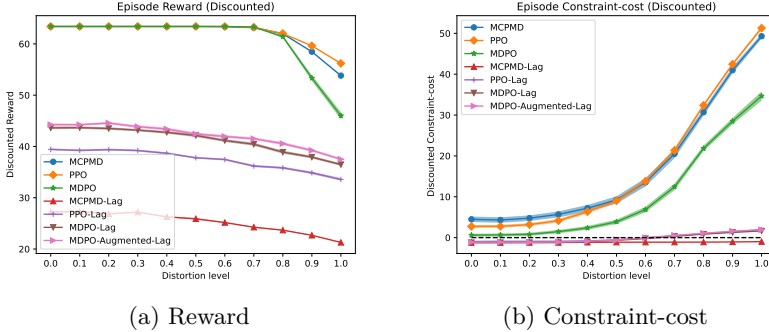

(a) Reward                (b) Constraint-cost

Figure 13: Test performance of MDP and CMDP algorithms obtained by applying the learned deterministic policy from the Cartpole domain after 200,000 time steps of training. The line and shaded area represent the mean and standard error across the perturbations for the particular distortion level.

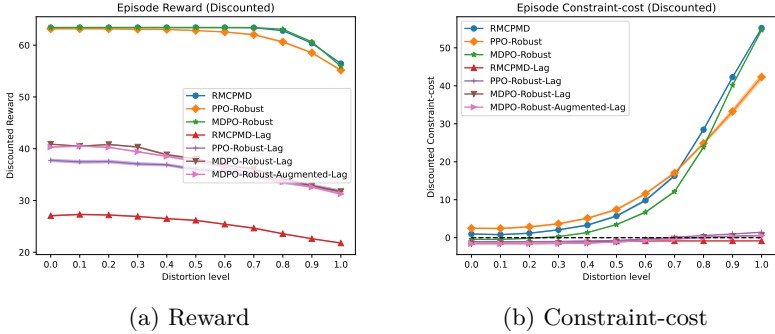

(a) Reward                (b) Constraint-cost

Figure 14: Test performance of RMDP and RCMDP algorithms obtained by applying the learned deterministic policy from the Cartpole domain after 200,000 time steps of training. The line and shaded area represent the mean and standard error across the perturbations for the particular distortion level.

## G.2 Inventory Management

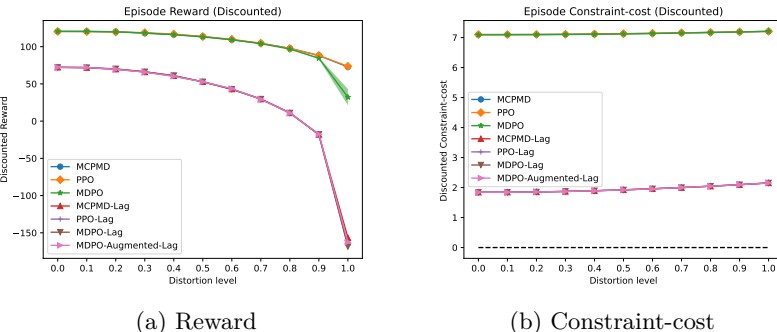

(a) Reward        (b) Constraint-cost

Figure 15: Test performance of MDP and CMDP algorithms obtained by applying the learned deterministic policy from the Inventory Management domain after 400,000 time steps of training. The line and shaded area represent the mean and standard error across the perturbations for the particular distortion level.

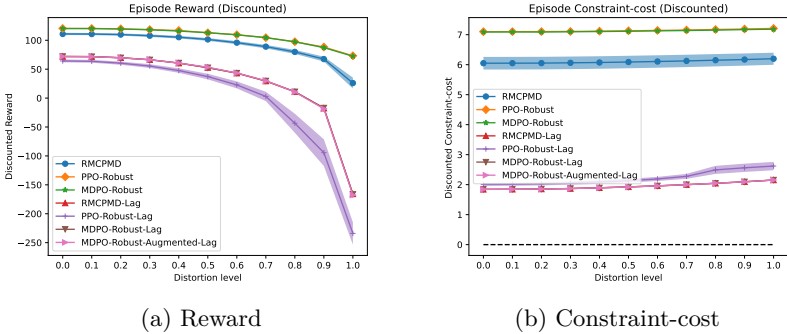

(a) Reward        (b) Constraint-cost

Figure 16: Test performance of RMDP and RCMDP algorithms obtained by applying the learned deterministic policy from the Inventory Management domain after 400,000 time steps of training. The line and shaded area represent the mean and standard error across the perturbations for the particular distortion level.

## G.3  3-D Inventory Management

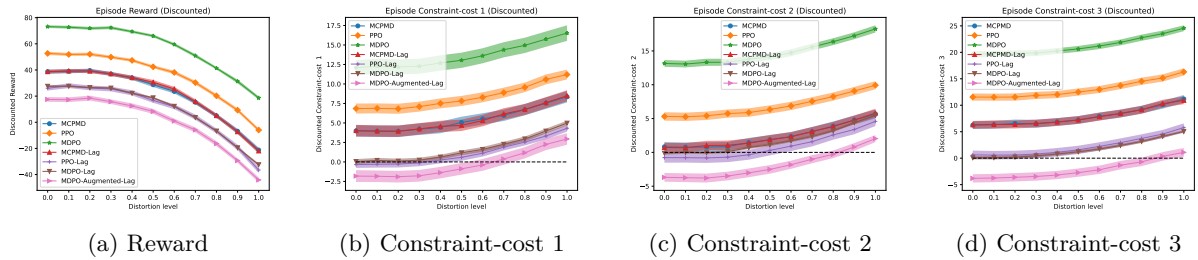

(a) Reward      (b) Constraint-cost 1      (c) Constraint-cost 2      (d) Constraint-cost 3

Figure 17: Test performance of RMDP and RCMDP algorithms obtained by applying the learned deterministic policy from the 3-D Inventory Management domain after 400,000 time steps of training. The line and shaded area represent the mean and standard error across the perturbations for the particular distortion level.

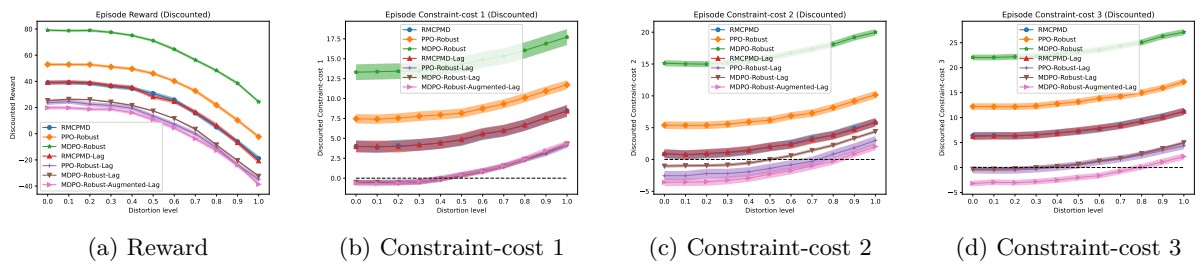

(a) Reward      (b) Constraint-cost 1      (c) Constraint-cost 2      (d) Constraint-cost 3

Figure 18: Test performance of RMDP and RCMDP algorithms obtained by applying the learned deterministic policy from the 3-D Inventory Management domain after 400,000 time steps of training. The line and shaded area represent the mean and standard error across the perturbations for the particular distortion level.

# H   Comparison of schedules

Tomar et al. (2022) propose either a fixed $\alpha$ or a linear increase across iterations to give a larger penalty at the end of the iterations (inspired by theoretical works, e.g. Beck & Teboulle (2003)). A potential problem with such a linear scheme is that if the LTMA update is strong and the policy cannot take sufficiently large learning steps, it may sometimes lead to a gradual loss of performance and premature convergence, especially in the context of contrained optimisation where the objective is often subject to large changes due to constraints becoming active or inactive. As an alternative, we also consider a schedule where the penalty parameter schedule is restarted every time the LTMA update to $\xi$ is larger than a particular threshold. The schedule then follows the process $\alpha_k = \frac{1}{1-(k-k')/(K-k')}$, where $k'$ indicates the restart time and $K$ is the final time. In the experiments, the restart is done every time any dimension $i$ of the update is larger than half the maximal distance from nominal, i.e. larger than $0.5 \max_{\xi_i \in \mathcal{U}_{\xi_i}} |\xi_i - \bar{\xi}_i|$. A last included schedule is the geometrically decreasing schedule $\alpha_k = \alpha_{k-1} * \gamma$ based on the discount factor $\gamma$ as proposed in earlier work (Xiao, 2022). We experiment with both epoch-based and batch-based schedules. While in the batch-based schedules we additionally experimented with the more aggressive update rule $\alpha_k = \alpha_{k-1} \frac{1-\gamma}{M}$ proposed in Wang et al. (2024). However, the results under the scheme were poor, and so these results are not included.

Table 11 and 12 summarise the experiments comparing fixed schedules, restart schedules, linear schedules, and the geometric schedules for MDPO-Robust and MDPO-Robust-Lag variants. For the fixed schedule, the parameter is set to $\alpha = 2.0$. For the geometric schedule, the starting parameter is set to $\alpha = 5.0$ for epoch-based scheduling and to $\alpha = 2.0$ for the less frequent batch-based scheduling. The fixed setting works reasonably well while in particular cases there may be benefits from time-varying schedules. The best overall performance for robust-constrained RL is obtained by the batch-based restart schedule with MDPO-Robust-Augmented-Lag. As a second top performer, the geometric schedule with MDPO-Robust-Lag also performs consistently high.

Table 11: Return and penalised return statistics comparing **per-epoch** schedules for the penalty parameter $\alpha$ under the short runs (200-500 episodes $\times$ maximal number of time steps) and the long runs (2000-5000 episodes $\times$ maximal number of time steps).

| Algorithm | Return | | $R_{\text{pen}}^{\pm}$ (signed) | | $R_{\text{pen}}$ (positive) | |
|---|---|---|---|---|---|---|
| | Mean $\pm$ SE | Min | Mean $\pm$ SE | Min | Mean $\pm$ SE | Min |
| **Cartpole (short runs)** | | | | | | |
| MDPO-Robust (fixed) | $63.1_{\pm 0.1}$ | 56.4 | $-229.7_{\pm 179.6}$ | -2610.6 | $-294.1_{\pm 171.4}$ | -2610.8 |
| MDPO-Robust (linear) | $62.9_{\pm 0.1}$ | 55.0 | $-482.5_{\pm 182.3}$ | -2487.1 | $-514.6_{\pm 177.4}$ | -2487.1 |
| MDPO-Robust (restart) | $63.1_{\pm 0.1}$ | 55.2 | $-252.8_{\pm 179.1}$ | -2520.5 | $-304.8_{\pm 172.0}$ | -2520.6 |
| MDPO-Robust (geometric) | $63.1_{\pm 0.1}$ | 55.8 | $-148.0_{\pm 173.5}$ | -2480.3 | $-219.3_{\pm 164.7}$ | -2481.5 |
| MDPO-Robust-Lag (fixed) | $34.4_{\pm 0.3}$ | 24.9 | $108.4_{\pm 4.4}$ | 65.9 | $34.2_{\pm 1.2}$ | 24.9 |
| MDPO-Robust-Lag (linear) | $35.3_{\pm 0.3}$ | 25.2 | $107.1_{\pm 5.0}$ | 53.0 | $34.5_{\pm 1.2}$ | 25.0 |
| MDPO-Robust-Lag (restart) | $34.9_{\pm 0.3}$ | 25.7 | $107.2_{\pm 4.9}$ | 53.1 | $34.3_{\pm 1.1}$ | 25.4 |
| MDPO-Robust-Lag (geometric) | $34.8_{\pm 0.3}$ | 25.0 | $109.4_{\pm 4.8}$ | 60.2 | $34.5_{\pm 1.2}$ | 25.0 |
| MDPO-Robust-Augmented-Lag (fixed) | $34.6_{\pm 0.3}$ | 24.7 | $110.8_{\pm 4.9}$ | 61.6 | $34.6_{\pm 1.2}$ | 24.7 |
| MDPO-Robust-Augmented-Lag (linear) | $35.2_{\pm 0.3}$ | 25.4 | $106.6_{\pm 5.0}$ | 52.6 | $34.2_{\pm 1.2}$ | 22.9 |
| MDPO-Robust-Augmented-Lag (restart) | $35.4_{\pm 0.3}$ | 25.7 | $107.1_{\pm 5.4}$ | 45.6 | $34.4_{\pm 1.3}$ | 21.7 |
| MDPO-Robust-Augmented-Lag (geometric) | $35.1_{\pm 0.3}$ | 24.6 | $109.7_{\pm 5.1}$ | 56.4 | $34.7_{\pm 1.2}$ | 24.6 |
| **Cartpole (long runs)** | | | | | | |
| MDPO-Robust (fixed) | $63.2_{\pm 0.1}$ | 56.1 | $-109.7_{\pm 176.0}$ | -2647.1 | $-185.5_{\pm 167.1}$ | -2648.3 |
| MDPO-Robust (linear) | $33.7_{\pm 0.1}$ | 29.2 | $-145.3_{\pm 24.1}$ | -526.4 | $-165.0_{\pm 23.0}$ | -541.6 |
| MDPO-Robust (restart) | $63.1_{\pm 0.1}$ | 55.8 | $-173.6_{\pm 174.9}$ | -2537.4 | $-237.1_{\pm 167.0}$ | -2538.1 |
| MDPO-Robust (geometric) | $63.2_{\pm 0.1}$ | 56.1 | $-120.4_{\pm 179.5}$ | -2662.1 | $-193.7_{\pm 170.7}$ | -2662.7 |
| MDPO-Robust-Lag (fixed) | $41.6_{\pm 0.2}$ | 32.1 | $105.1_{\pm 9.5}$ | 9.6 | $38.6_{\pm 2.5}$ | 2.6 |
| MDPO-Robust-Lag (linear) | $32.3_{\pm 0.3}$ | 22.9 | $72.0_{\pm 4.1}$ | 38.6 | $15.4_{\pm 1.8}$ | -4.8 |
| MDPO-Robust-Lag (restart) | $41.8_{\pm 0.2}$ | 32.2 | $107.6_{\pm 9.3}$ | 14.5 | $39.2_{\pm 2.3}$ | 5.8 |
| MDPO-Robust-Lag (geometric) | $41.4_{\pm 0.2}$ | 31.7 | $103.7_{\pm 9.4}$ | 8.0 | $38.6_{\pm 2.5}$ | 1.3 |
| MDPO-Robust-Augmented-Lag (fixed) | $41.2_{\pm 0.2}$ | 31.4 | $106.2_{\pm 9.1}$ | 13.7 | $38.6_{\pm 2.2}$ | 5.0 |
| MDPO-Robust-Augmented-Lag (linear) | $30.0_{\pm 0.3}$ | 21.5 | $32.8_{\pm 2.8}$ | 12.2 | $-18.8_{\pm 1.3}$ | -34.6 |
| MDPO-Robust-Augmented-Lag (restart) | $40.5_{\pm 0.2}$ | 30.5 | $108.7_{\pm 8.2}$ | 24.1 | $38.3_{\pm 1.9}$ | 10.7 |
| MDPO-Robust-Augmented-Lag (geometric) | $40.9_{\pm 0.2}$ | 31.4 | $102.1_{\pm 8.9}$ | 9.2 | $37.6_{\pm 2.7}$ | -2.1 |
| **Inventory Management (short runs)** | | | | | | |
| MDPO-Robust (fixed) | $118.0_{\pm 1.8}$ | 72.3 | $-3306.4_{\pm 9.7}$ | -3350.0 | $-3306.4_{\pm 9.7}$ | -3350.0 |
| MPDO-Robust (linear) | $98.4_{\pm 5.7}$ | -59.6 | $-2764.0_{\pm 20.5}$ | -2808.5 | $-2764.2_{\pm 20.2}$ | -2808.5 |
| MPDO-Robust (restart) | $105.1_{\pm 6.3}$ | -20.2 | $-3404.8_{\pm 32.9}$ | -3634.3 | $-3404.8_{\pm 32.9}$ | -3634.3 |
| MPDO-Robust (geometric) | $114.6_{\pm 2.3}$ | 49.7 | $-3166.4_{\pm 6.9}$ | -3207.6 | $-3166.5_{\pm 6.9}$ | -3207.6 |
| MPDO-Robust-Lag (fixed) | $60.7_{\pm 10.7}$ | -228.5 | $-930.9_{\pm 32.3}$ | -1118.2 | $-932.2_{\pm 30.8}$ | -1118.2 |
| MPDO-Robust-Lag (linear) | $93.2_{\pm 4.7}$ | -55.0 | $-2186.8_{\pm 6.9}$ | -2248.8 | $-2187.3_{\pm 6.4}$ | -2248.8 |
| MPDO-Robust-Lag (restart) | $88.3_{\pm 4.8}$ | -61.3 | $-1880.7_{\pm 16.5}$ | -2003.7 | $-1881.1_{\pm 16.1}$ | -2003.7 |
| MPDO-Robust-Lag (geometric) | $67.2_{\pm 9.0}$ | -156.8 | $-1046.3_{\pm 10.2}$ | -1122.4 | $-1048.1_{\pm 8.0}$ | -1122.4 |
| MPDO-Robust-Augmented-Lag (fixed) | $60.9_{\pm 10.6}$ | -221.4 | $-938.6_{\pm 34.4}$ | -1172.2 | $-940.1_{\pm 32.9}$ | -1172.2 |
| MPDO-Robust-Augmented-Lag (linear) | $91.8_{\pm 4.4}$ | -47.8 | $-2033.0_{\pm 8.0}$ | -2054.6 | $-2033.3_{\pm 7.6}$ | -2054.6 |
| MPDO-Robust-Augmented-Lag (restart) | $85.6_{\pm 5.9}$ | -94.9 | $-1902.5_{\pm 6.5}$ | -1954.2 | $-1902.7_{\pm 6.2}$ | -1954.2 |
| MPDO-Robust-Augmented-Lag (geometric) | $60.0_{\pm 9.3}$ | -206.9 | $-1095.9_{\pm 18.3}$ | -1128.7 | $-1097.2_{\pm 16.3}$ | -1128.7 |
| **Inventory Management (long runs)** | | | | | | |
| MDPO-Robust (fixed) | $119.7_{\pm 1.8}$ | 72.4 | $-3416.7_{\pm 4.1}$ | -3429.7 | $-3416.7_{\pm 4.1}$ | -3429.7 |
| MPDO-Robust (linear) | $114.8_{\pm 2.3}$ | 53.6 | $-3166.7_{\pm 6.8}$ | -3208.2 | $-3166.7_{\pm 6.8}$ | -3208.2 |
| MPDO-Robust (restart) | $119.9_{\pm 1.8}$ | 74.4 | $-3425.2_{\pm 5.4}$ | -3460.2 | $-3425.2_{\pm 5.4}$ | -3460.2 |
| MPDO-Robust (geometric) | $114.7_{\pm 2.3}$ | 51.0 | $-3166.5_{\pm 6.9}$ | -3207.6 | $-3166.6_{\pm 6.9}$ | -3207.6 |
| MPDO-Robust-Lag (fixed) | $66.9_{\pm 7.1}$ | -166.5 | $-837.4_{\pm 21.4}$ | -939.3 | $-838.0_{\pm 20.4}$ | -939.3 |
| MPDO-Robust-Lag (linear) | $66.8_{\pm 7.1}$ | -165.7 | $-837.3_{\pm 21.4}$ | -939.1 | $-838.0_{\pm 20.3}$ | -939.1 |
| MPDO-Robust-Lag (restart) | $66.8_{\pm 7.2}$ | -169.1 | $-837.3_{\pm 21.6}$ | -939.8 | $-838.0_{\pm 20.5}$ | -939.8 |
| MPDO-Robust-Lag (geometric) | $66.8_{\pm 7.2}$ | -168.9 | $-837.3_{\pm 21.6}$ | -939.7 | $-837.9_{\pm 20.6}$ | -939.7 |
| MPDO-Robust-Augmented-Lag (fixed) | $66.8_{\pm 7.1}$ | -167.7 | $-837.2_{\pm 21.5}$ | -939.1 | $-837.9_{\pm 20.5}$ | -939.1 |
| MPDO-Robust-Augmented-Lag (linear) | $66.8_{\pm 7.1}$ | -163.4 | $-837.4_{\pm 21.3}$ | -939.2 | $-838.0_{\pm 20.3}$ | -939.2 |
| MPDO-Robust-Augmented-Lag (restart) | $66.8_{\pm 7.1}$ | -162.8 | $-837.4_{\pm 21.2}$ | -939.0 | $-837.9_{\pm 20.4}$ | -939.0 |
| MPDO-Robust-Augmented-Lag (geometric) | $66.9_{\pm 7.1}$ | -162.6 | $-837.4_{\pm 21.2}$ | -939.2 | $-838.1_{\pm 20.2}$ | -939.2 |
| **3-D Inventory Management (short runs)** | | | | | | |
| MDPO-Robust (fixed) | $54.9_{\pm 0.4}$ | -9.6 | $-7956.9_{\pm 457.5}$ | -14747.8 | $-9376.8_{\pm 330.2}$ | -14836.3 |
| MDPO-Robust (linear) | $56.5_{\pm 0.4}$ | -8.1 | $-8716.0_{\pm 458.1}$ | -15404.0 | $-9918.5_{\pm 343.1}$ | -15469.0 |
| MDPO-Robust (restart) | $56.3_{\pm 0.4}$ | -8.7 | $-8693.9_{\pm 455.4}$ | -15503.7 | $-9866.2_{\pm 333.9}$ | -15531.6 |
| MPDO-Robust (geometric) | $53.6_{\pm 0.4}$ | -11.5 | $-7337.6_{\pm 459.1}$ | -14074.2 | $-8846.6_{\pm 315.8}$ | -14173.9 |
| MPDO-Robust-Lag (fixed) | $41.4_{\pm 0.5}$ | -24.6 | $-1416.9_{\pm 446.1}$ | -7726.9 | $-5088.9_{\pm 221.7}$ | -8560.6 |
| MPDO-Robust-Lag (linear) | $41.8_{\pm 0.5}$ | -25.5 | $-1584.6_{\pm 447.5}$ | -8169.0 | $-5178.0_{\pm 222.3}$ | -8982.4 |
| MPDO-Robust-Lag (restart) | $41.9_{\pm 0.5}$ | -24.9 | $-1648.0_{\pm 446.1}$ | -8138.0 | $-5308.1_{\pm 223.1}$ | -8995.2 |
| MPDO-Robust-Lag (geometric) | $41.1_{\pm 0.5}$ | -28.2 | $-1278.3_{\pm 449.6}$ | -7571.7 | $-5060.9_{\pm 219.1}$ | -8659.8 |
| MPDO-Robust-Augmented-Lag (fixed) | $39.5_{\pm 0.5}$ | -29.6 | $-525.0_{\pm 441.2}$ | -6861.9 | $-4957.0_{\pm 211.2}$ | -8289.3 |
| MPDO-Robust-Augmented-Lag (linear) | $42.4_{\pm 0.5}$ | -23.7 | $-1929.8_{\pm 446.9}$ | -8420.5 | $-5311.8_{\pm 228.7}$ | -9203.4 |
| MPDO-Robust-Augmented-Lag (restart) | $40.7_{\pm 0.5}$ | -26.8 | $-1070.1_{\pm 441.9}$ | -7347.8 | $-5155.6_{\pm 216.3}$ | -8671.2 |
| MPDO-Robust-Augmented-Lag (geometric) | $42.8_{\pm 0.5}$ | -25.1 | $-2123.5_{\pm 450.2}$ | -8567.3 | $-5507.8_{\pm 228.9}$ | -9303.0 |
| **3-D Inventory Management (long runs)** | | | | | | |
| MDPO-Robust (fixed) | $87.0_{\pm 0.4}$ | 24.4 | $-23312.6_{\pm 525.6}$ | -31314.1 | $-23459.7_{\pm 499.3}$ | -31314.1 |
| MPDO-Robust (linear) | $96.3_{\pm 0.4}$ | 36.0 | $-27752.5_{\pm 547.7}$ | -35936.5 | $-27902.6_{\pm 525.3}$ | -35936.5 |
| MPDO-Robust (restart) | $90.8_{\pm 0.4}$ | 27.4 | $-25169.7_{\pm 523.8}$ | -32979.5 | $-25265.0_{\pm 504.2}$ | -32979.5 |
| MPDO-Robust (geometric) | $82.2_{\pm 0.4}$ | 20.0 | $-21022.5_{\pm 487.2}$ | -28373.0 | $-21098.2_{\pm 469.0}$ | -28373.0 |
| MPDO-Robust-Lag (fixed) | $32.8_{\pm 0.5}$ | -32.4 | $2617.9_{\pm 378.6}$ | -2601.7 | $-1011.4_{\pm 169.9}$ | -3767.4 |
| MPDO-Robust-Lag (linear) | $33.2_{\pm 0.5}$ | -31.9 | $2538.8_{\pm 368.0}$ | -2793.7 | $-1353.1_{\pm 197.6}$ | -4804.2 |
| MPDO-Robust-Lag (restart) | $34.0_{\pm 0.5}$ | -31.2 | $2062.3_{\pm 375.3}$ | -3333.9 | $-1317.1_{\pm 191.5}$ | -4687.1 |
| MPDO-Robust-Lag (geometric) | $31.7_{\pm 0.5}$ | -34.9 | $3269.5_{\pm 371.3}$ | -2036.2 | $-1097.3_{\pm 173.3}$ | -4299.3 |
| MPDO-Robust-Augmented-Lag (fixed) | $26.8_{\pm 0.4}$ | -38.6 | $5312.6_{\pm 340.2}$ | 758.7 | $-1219.2_{\pm 150.4}$ | -3867.7 |
| MPDO-Robust-Augmented-Lag (linear) | $23.5_{\pm 0.4}$ | -41.1 | $6993.3_{\pm 325.6}$ | 2682.2 | $-963.4_{\pm 122.0}$ | -3066.4 |
| MPDO-Robust-Augmented-Lag (restart) | $24.2_{\pm 0.5}$ | -41.3 | $6811.0_{\pm 333.5}$ | 2581.0 | $-1043.3_{\pm 159.2}$ | -3775.2 |
| MPDO-Robust-Augmented-Lag (geometric) | $26.5_{\pm 0.5}$ | -40.3 | $5494.4_{\pm 348.4}$ | 824.2 | $-1644.2_{\pm 167.2}$ | -4586.5 |

Table 12: Return and penalised return statistics comparing **per-batch** schedules for the penalty parameter $\alpha$ under the short runs (200-500 episodes × maximal number of time steps) and the long runs (2000-5000 episodes × maximal number of time steps).

| Algorithm | Return | | $R^{\pm}_{\text{pen}}$ (signed) | | $R_{\text{pen}}$ (positive) | |
|---|---|---|---|---|---|---|
| | Mean ± SE | Min | Mean ± SE | Min | Mean ± SE | Min |
| **Cartpole (short runs)** | | | | | | |
| MDPO-Robust (fixed) | $63.1_{\pm 0.1}$ | 56.4 | $-229.7_{\pm 179.6}$ | -2610.6 | $-294.1_{\pm 171.4}$ | -2610.8 |
| MDPO-Robust (linear) | $50.7_{\pm 0.1}$ | 44.4 | $-183.8_{\pm 58.8}$ | -831.2 | $-214.9_{\pm 54.7}$ | -833.9 |
| MDPO-Robust (restart) | $62.5_{\pm 0.1}$ | 55.6 | $-440.3_{\pm 173.8}$ | -2450.3 | $-471.9_{\pm 169.0}$ | -2450.5 |
| MDPO-Robust (geometric) | $63.1_{\pm 0.1}$ | 55.6 | $-286.5_{\pm 178.9}$ | -2504.6 | $-339.3_{\pm 171.8}$ | -2504.8 |
| MDPO-Robust-Lag (fixed) | $34.4_{\pm 0.3}$ | 24.9 | $108.4_{\pm 4.4}$ | 65.9 | $34.2_{\pm 1.2}$ | 24.9 |
| MDPO-Robust-Lag (linear) | $35.3_{\pm 0.3}$ | 25.2 | $107.1_{\pm 5.0}$ | 53.0 | $34.5_{\pm 1.2}$ | 25.0 |
| MDPO-Robust-Lag (linear) | $34.7_{\pm 0.3}$ | 24.7 | $108.7_{\pm 4.7}$ | 62.3 | $34.4_{\pm 1.2}$ | 24.7 |
| MDPO-Robust-Lag (restart) | $35.4_{\pm 0.3}$ | 25.1 | $111.0_{\pm 5.1}$ | 59.1 | $35.1_{\pm 1.2}$ | 25.1 |
| MDPO-Robust-Lag (geometric) | $34.6_{\pm 0.3}$ | 24.3 | $109.4_{\pm 4.7}$ | 64.0 | $34.3_{\pm 1.1}$ | 24.3 |
| MDPO-Robust-Augmented-Lag (fixed) | $34.6_{\pm 0.3}$ | 24.7 | $110.8_{\pm 4.9}$ | 61.6 | $34.6_{\pm 1.2}$ | 24.7 |
| MDPO-Robust-Augmented-Lag (linear) | $34.5_{\pm 0.3}$ | 24.4 | $110.2_{\pm 4.7}$ | 64.7 | $34.4_{\pm 1.2}$ | 24.4 |
| MDPO-Robust-Augmented-Lag (restart) | $34.4_{\pm 0.3}$ | 24.6 | $108.2_{\pm 4.4}$ | 67.6 | $34.0_{\pm 1.2}$ | 24.6 |
| MDPO-Robust-Augmented-Lag (geometric) | $35.0_{\pm 0.3}$ | 25.0 | $109.6_{\pm 5.1}$ | 57.1 | $34.6_{\pm 1.2}$ | 25.0 |
| **Cartpole (long runs)** | | | | | | |
| MDPO-Robust (fixed) | $63.2_{\pm 0.1}$ | 56.1 | $-109.7_{\pm 176.0}$ | -2647.1 | $-185.5_{\pm 167.1}$ | -2648.3 |
| MDPO-Robust (linear) | $18.6_{\pm 0.1}$ | 15.6 | $-275.5_{\pm 30.8}$ | -630.8 | $-282.6_{\pm 30.8}$ | -640.4 |
| MDPO-Robust (restart) | $63.0_{\pm 0.1}$ | 55.2 | $-246.4_{\pm 174.0}$ | -2510.7 | $-298.2_{\pm 167.1}$ | -2510.9 |
| MDPO-Robust (geometric) | $63.1_{\pm 0.1}$ | 56.2 | $-151.9_{\pm 176.3}$ | -2639.6 | $-219.7_{\pm 168.1}$ | -2640.4 |
| MDPO-Robust-Lag (fixed) | $41.6_{\pm 0.2}$ | 32.1 | $105.1_{\pm 9.5}$ | 9.6 | $38.6_{\pm 2.5}$ | 2.6 |
| MDPO-Robust-Lag (linear) | $22.2_{\pm 0.2}$ | 16.4 | $-104.5_{\pm 5.2}$ | -165.9 | $-137.5_{\pm 5.1}$ | -199.0 |
| MDPO-Robust-Lag (restart) | $41.1_{\pm 0.2}$ | 30.9 | $112.4_{\pm 8.6}$ | 30.1 | $39.6_{\pm 1.7}$ | 16.6 |
| MDPO-Robust-Lag (geometric) | $41.0_{\pm 0.2}$ | 30.9 | $106.3_{\pm 8.8}$ | 16.2 | $38.5_{\pm 2.2}$ | 7.1 |
| MDPO-Robust-Augmented-Lag (fixed) | $41.2_{\pm 0.2}$ | 31.4 | $106.2_{\pm 9.1}$ | 13.7 | $38.6_{\pm 2.2}$ | 5.0 |
| MDPO-Robust-Augmented-Lag (linear) | $25.8_{\pm 0.2}$ | 19.2 | $-21.6_{\pm 3.3}$ | -46.8 | $-66.9_{\pm 2.8}$ | -102.0 |
| MDPO-Robust-Augmented-Lag (restart) | $40.3_{\pm 0.2}$ | 30.3 | $111.7_{\pm 8.0}$ | 33.5 | $39.0_{\pm 1.5}$ | 20.0 |
| MDPO-Robust-Augmented-Lag (geometric) | $40.9_{\pm 0.2}$ | 31.1 | $105.6_{\pm 8.8}$ | 16.5 | $38.2_{\pm 2.2}$ | 6.3 |
| **Inventory Management (short runs)** | | | | | | |
| MDPO-Robust (fixed) | $118.0_{\pm 1.8}$ | 72.3 | $-3306.4_{\pm 9.7}$ | -3350.0 | $-3306.4_{\pm 9.7}$ | -3350.0 |
| MDPO-Robust (linear) | $120.0_{\pm 1.8}$ | 73.6 | $-3425.4_{\pm 5.4}$ | -3460.1 | $-3425.4_{\pm 5.4}$ | -3460.1 |
| MDPO-Robust (restart) | $116.4_{\pm 2.1}$ | 55.3 | $-3240.2_{\pm 9.3}$ | -3293.8 | $-3240.2_{\pm 9.2}$ | -3293.8 |
| MDPO-Robust (geometric) | $119.0_{\pm 1.8}$ | 72.3 | $-3361.8_{\pm 7.1}$ | -3382.9 | $-3361.8_{\pm 7.1}$ | -3382.9 |
| MPDO-Robust-Lag (fixed) | $60.7_{\pm 10.7}$ | -228.5 | $-930.9_{\pm 32.3}$ | -1118.2 | $-932.2_{\pm 30.8}$ | -1118.2 |
| MPDO-Robust-Lag (linear) | $56.3_{\pm 9.8}$ | -226.8 | $-939.1_{\pm 35.3}$ | -1175.6 | $-940.5_{\pm 33.8}$ | -1175.6 |
| MPDO-Robust-Lag (restart) | $68.2_{\pm 8.8}$ | -168.2 | $-964.1_{\pm 19.3}$ | -1117.6 | $-964.5_{\pm 18.8}$ | -1117.6 |
| MPDO-Robust-Lag (geometric) | $61.9_{\pm 8.2}$ | -198.7 | $-885.0_{\pm 24.6}$ | -981.9 | $-886.9_{\pm 21.7}$ | -981.9 |
| MPDO-Robust-Augmented-Lag (fixed) | $60.9_{\pm 10.6}$ | -221.4 | $-938.6_{\pm 34.4}$ | -1172.2 | $-940.1_{\pm 32.9}$ | -1172.2 |
| MPDO-Robust-Augmented-Lag (linear) | $68.9_{\pm 8.9}$ | -180.3 | $-932.6_{\pm 28.3}$ | -1118.2 | $-934.2_{\pm 26.7}$ | -1118.2 |
| MDPO-Robust-Augmented-Lag (restart) | $66.9_{\pm 7.1}$ | -165.4 | $-837.6_{\pm 21.1}$ | -939.1 | $-838.2_{\pm 20.2}$ | -939.1 |
| MDPO-Robust-Augmented-Lag (geometric) | $66.8_{\pm 7.1}$ | -166.0 | $-837.3_{\pm 21.4}$ | -938.6 | $-837.9_{\pm 20.4}$ | -938.6 |
| **Inventory Management (long runs)** | | | | | | |
| MDPO-Robust (fixed) | $119.7_{\pm 1.8}$ | 72.4 | $-3416.7_{\pm 4.1}$ | -3429.7 | $-3416.7_{\pm 4.1}$ | -3429.7 |
| MDPO-Robust (linear) | $109.2_{\pm 2.9}$ | 21.5 | $-2907.0_{\pm 8.5}$ | -2950.6 | $-2907.1_{\pm 8.3}$ | -2950.6 |
| MDPO-Robust (restart) | $120.0_{\pm 1.8}$ | 73.0 | $-3421.4_{\pm 4.7}$ | -3441.4 | $-3421.4_{\pm 4.7}$ | -3441.4 |
| MDPO-Robust (geometric) | $114.0_{\pm 2.3}$ | 50.3 | $-3121.5_{\pm 5.2}$ | -3134.9 | $-3121.5_{\pm 5.2}$ | -3134.9 |
| MPDO-Robust-Lag (fixed) | $66.9_{\pm 7.1}$ | -166.5 | $-837.4_{\pm 21.4}$ | -939.3 | $-838.0_{\pm 20.4}$ | -939.3 |
| MPDO-Robust-Lag (linear) | $60.7_{\pm 10.8}$ | -234.4 | $-930.9_{\pm 32.6}$ | -1118.1 | $-932.4_{\pm 30.9}$ | -1118.1 |
| MPDO-Robust-Lag (restart) | $66.8_{\pm 7.1}$ | -162.6 | $-837.4_{\pm 21.2}$ | -939.0 | $-837.9_{\pm 20.4}$ | -939.0 |
| MPDO-Robust-Lag (geometric) | $66.8_{\pm 7.1}$ | -163.7 | $-837.4_{\pm 21.3}$ | -939.4 | $-837.9_{\pm 20.5}$ | -939.4 |
| MPDO-Robust-Augmented-Lag (fixed) | $66.8_{\pm 7.1}$ | -167.7 | $-837.2_{\pm 21.5}$ | -939.1 | $-837.9_{\pm 20.5}$ | -939.1 |
| MPDO-Robust-Augmented-Lag (linear) | $66.8_{\pm 7.2}$ | -168.8 | $-837.3_{\pm 21.5}$ | -939.1 | $-837.8_{\pm 20.6}$ | -939.1 |
| MPDO-Robust-Augmented-Lag (restart) | $66.7_{\pm 7.2}$ | -170.1 | $-837.1_{\pm 21.6}$ | -939.5 | $-837.9_{\pm 20.5}$ | -939.5 |
| MDPO-Robust-Augmented-Lag (geometric) | $66.7_{\pm 7.2}$ | -168.4 | $-837.2_{\pm 21.5}$ | -939.6 | $-837.9_{\pm 20.5}$ | -939.6 |
| **3-D Inventory Management (short runs)** | | | | | | |
| MDPO-Robust (fixed) | $54.9_{\pm 0.4}$ | -9.6 | $-7956.9_{\pm 457.5}$ | -14747.8 | $-9376.8_{\pm 330.2}$ | -14836.3 |
| MDPO-Robust (linear) | $55.1_{\pm 0.4}$ | -9.5 | $-8112.0_{\pm 455.7}$ | -14726.5 | $-9507.9_{\pm 328.3}$ | -14830.6 |
| MDPO-Robust (restart) | $54.5_{\pm 0.4}$ | -8.4 | $-7809.5_{\pm 453.2}$ | -14454.0 | $-9408.3_{\pm 318.5}$ | -14561.6 |
| MDPO-Robust (geometric) | $55.4_{\pm 0.4}$ | -9.3 | $-8215.7_{\pm 454.4}$ | -14973.8 | $-9588.4_{\pm 332.3}$ | -15084.5 |
| MPDO-Robust-Lag (fixed) | $41.4_{\pm 0.5}$ | -24.6 | $-1416.9_{\pm 446.1}$ | -7726.9 | $-5088.9_{\pm 221.7}$ | -8560.6 |
| MPDO-Robust-Lag (linear) | $41.7_{\pm 0.5}$ | -26.1 | $-1573.9_{\pm 446.2}$ | -8108.3 | $-5164.3_{\pm 222.7}$ | -8973.5 |
| MPDO-Robust-Lag (restart) | $41.0_{\pm 0.5}$ | -27.1 | $-1239.8_{\pm 440.8}$ | -7650.5 | $-5350.6_{\pm 217.9}$ | -8969.6 |
| MPDO-Robust-Lag (geometric) | $40.2_{\pm 0.5}$ | -26.7 | $-778.5_{\pm 441.4}$ | -7152.1 | $-5115.9_{\pm 212.1}$ | -8602.1 |
| MDPO-Robust-Augmented-Lag (fixed) | $39.5_{\pm 0.5}$ | -29.6 | $-525.0_{\pm 441.2}$ | -6861.9 | $-4957.0_{\pm 211.2}$ | -8289.3 |
| MDPO-Robust-Augmented-Lag (linear) | $41.5_{\pm 0.5}$ | -25.6 | $-1474.0_{\pm 443.6}$ | -7706.7 | $-5299.4_{\pm 218.6}$ | -8798.9 |
| MDPO-Robust-Augmented-Lag (restart) | $40.0_{\pm 0.5}$ | -28.7 | $-754.8_{\pm 438.8}$ | -7087.5 | $-4950.3_{\pm 210.5}$ | -8389.0 |
| MDPO-Robust-Augmented-Lag (geometric) | $41.6_{\pm 0.5}$ | -24.6 | $-1582.8_{\pm 445.8}$ | -8124.9 | $-5316.9_{\pm 218.7}$ | -9080.0 |
| **3-D Inventory Management (long runs)** | | | | | | |
| MDPO-Robust (fixed) | $87.0_{\pm 0.4}$ | 24.4 | $-23312.6_{\pm 525.6}$ | -31314.1 | $-23459.7_{\pm 499.3}$ | -31314.1 |
| MDPO-Robust (linear) | $100.1_{\pm 0.4}$ | 41.1 | $-29655.3_{\pm 540.5}$ | -37776.0 | $-29778.8_{\pm 520.6}$ | -37776.0 |
| MDPO-Robust (restart) | $93.9_{\pm 0.4}$ | 32.7 | $-26709.0_{\pm 534.6}$ | -34493.9 | $-26815.8_{\pm 514.0}$ | -34493.9 |
| MDPO-Robust (geometric) | $89.8_{\pm 0.4}$ | 28.8 | $-24688.1_{\pm 520.7}$ | -32646.2 | $-24771.7_{\pm 503.7}$ | -32648.9 |
| MPDO-Robust-Lag (fixed) | $32.8_{\pm 0.5}$ | -32.4 | $2617.9_{\pm 378.6}$ | -2601.7 | $-1011.4_{\pm 169.9}$ | -3767.4 |
| MDPO-Robust-Lag (linear) | $39.3_{\pm 0.4}$ | -24.4 | $-427.1_{\pm 372.9}$ | -5798.5 | $-2588.9_{\pm 233.2}$ | -6431.7 |
| MDPO-Robust-Lag (restart) | $39.1_{\pm 0.4}$ | -27.4 | $-274.3_{\pm 379.8}$ | -5788.0 | $-2356.2_{\pm 228.9}$ | -6272.1 |
| MDPO-Robust-Lag (geometric) | $36.7_{\pm 0.4}$ | -29.0 | $602.2_{\pm 373.5}$ | -4823.9 | $-2245.8_{\pm 207.3}$ | -5945.4 |
| MPDO-Robust-Augmented-Lag (fixed) | $26.8_{\pm 0.4}$ | -38.6 | $5312.6_{\pm 340.2}$ | 758.7 | $-1219.2_{\pm 150.4}$ | -3867.7 |
| MDPO-Robust-Augmented-Lag (linear) | $22.8_{\pm 0.5}$ | -43.5 | $7297.2_{\pm 334.4}$ | 3001.8 | $-997.1_{\pm 133.8}$ | -3293.2 |
| MDPO-Robust-Augmented-Lag (restart) | $22.8_{\pm 0.5}$ | -42.8 | $7290.6_{\pm 347.8}$ | 2672.1 | $-679.9_{\pm 136.7}$ | -3116.2 |
| MDPO-Robust-Augmented-Lag (geometric) | $30.8_{\pm 0.4}$ | -33.8 | $3642.3_{\pm 357.3}$ | -1105.7 | $-1898.7_{\pm 200.9}$ | -5429.6 |

