# OpenReview forum: "Mirror Descent Policy Optimisation for Robust Constrained Markov Decision Processes"
_TMLR — Accepted by TMLR_

### Review · Reviewer_kh7f · 2025-08-11

**Summary Of Contributions:**

This paper introduces Mirror Descent Policy Optimization for RCMDPs, combining primal dual policy gradient updates with robust transition kernel optimisation using Transition Mirror Ascent. It establishes convergence rates in oracle and sample based settings, with theoretical guarantees for robustness and constraint satisfaction. This paper also includes rigorous proofs and empirical evaluation against baseline robust policy optimization methods.

**Audience:**

Yes

**Broader Impact Concerns:**

There is no ethical concern is this paper.

**Claims And Evidence:**

Yes

**Requested Changes:**

Questions and requested changes:

1. Can the Transition Mirror Ascent step be adapted to a fully sample based setting without degrading the stated convergence rates?

2. Could you provide a more detailed discussion of Slater’s condition, for example, whether this assumption is overly strict and how it compares with related assumptions in the literature? The current discussion below Assumption 3 is rather brief. An open question is: what modifications would be required to handle weaker feasibility assumptions than Slater’s condition in RCMDPs?

3. An open question: Can this method be extended to uncertainty sets beyond the convex or rectangular structure, or to rectangular uncertainty sets defined using alternative divergences such as $\chi^2$ or total variation divergence? Can you discuss the potential challenges for the extension?

4. Explicitly acknowledge that some convergence results assume oracle access to transition dynamics and discuss potential approaches for making the Transition Mirror Ascent step fully sample based — **critical**.

5. Extend the theoretical analysis to cover uncertainty sets beyond the convex or rectangular structure, or to rectangular uncertainty sets defined using alternative divergences. If this is hard and beyond the scope of this paper, you can discuss the potential challenges — **would strengthen**.

6. Analyze the effect of estimation error propagation between policy and transition updates in the sample based regime — **would strengthen**.

**Strengths And Weaknesses:**

Strengths:
1. This paper proposes a novel integration of mirror descent based primal dual policy optimization with adversarial transition kernel updates via Transition Mirror Ascent in the RCMDP setting, extending theory from both CMDPs and RMDPs.

2. This paper provides rigorous and complete convergence proofs for oracle and sample based regimes.

3. This paper clearly states and justifies methodological assumptions such as bounded costs, convex and rectangular uncertainty sets, and Slater’s condition. This ensures the theoretical results are mathematically sound.


Weaknesses:
1. This method relies on oracle access for transition kernel updates in key theoretical results, limiting the generality of the strongest guarantees.

2. This paper restricts analysis to convex and rectangular uncertainty sets.

3. This paper does not fully analyze how estimation errors from the sample based setting propagate between policy and transition updates.

---

> ### Author Response · Authors · 2025-10-04
>
> We thank the reviewer for recognising the strengths of our work in terms of rigour, complete convergence proofs, and clearly stating and justifying methodological assumptions. The reviewer also mentions limitations which we discuss and address point by point below.
>
> **Q1.** *Can TMA be adapted to the sample-based setting without degrading the convergence rates?*
>
> We have now designed an algorithm for Approximate TMA and have analysed its properties in Section 4.4 of the updated paper. In addition to the usual updates of TMA, the algorithm also considers the problem of approximating the Q-function. Following our sample-based analysis, we find that it satisfies the requirement of $\tilde{O}(1/\epsilon^2)$ for each of the macro-iterations, as required by our main theorem for the sample-based average regret.
>
> **Q2.** *A more detailed discussion of Slater’s condition (how strict is it, related assumptions in the literature, and potential alternative assumptions).*
>
> The condition is an extension of the traditional Slater's condition for CMDPs and many constrained RL algorithms do mention the assumption explicitly. Approaches without Slater's condition are possible using primal linear programming but are less practical due to requiring estimates of the occupancy measure. The condition of having a subset of the policies being able to strictly satisfy all the constraints for all the transition dynamics reflects the RCMDP objective since the worst-case dynamic w.r.t. the constraints should be solvable. A further discussion can be found in the extended discussion of Assumption 4 in the updated paper.
>
> **Q3.** *Possibility of using uncertainty sets beyond the convex or rectangular structure*
>
> We have discussed this assumption (Assumption 5 in the updated paper) in more detail now. The convex set assumption ensures updates to the transition kernel will remain in the interior so long as it is on a line between two interior points, a common assumption gradient descent/ascent type algorithms. The rectangular uncertainty set assumption is satisfied for most common uncertainty sets, e.g. state-action conditioned $\ell_1$ and $\ell_2$ sets centred around nominal probability distributions. In our main theorem, we initially made use of $\ell_1$ sets to provided a lower constant; in the updated paper, Lemma 28 provides constants for different kinds of uncertainty sets, namely general, $\ell_q$ rectangular, and $\ell_q$ non-rectangular uncertainty sets.
>
> While the TMA algorithm was previously derived only for rectangular uncertainty sets and to supplement our analysis, we have now performed an additional analysis (Section 5.5 in the updated paper), where we consider the sample-based convergence rate for non-rectangular uncertainty sets by considering policy iteration as a special case of Approximate TMA, and then apply the conservative policy iteration technique and convergence analysis of Li et al.
>
> **Q4.** *Possibility of using uncertainty sets with alternative divergences such as $\chi^2$ or total variation divergence.*
>
> We have updated the paper with a discussion of various uncertainty sets in the related work section. For direct parametrisations, one may consider the uncertainty set as similar to estimating the expected value from a limited number of samples from the true environment. This can be related to student-t and $\chi^2$ statistics, Hoeffding or Chebyshev inequalities, and the total variation distance, which can be related to the $\ell_1$ norm over the probability distribution, a case covered by our theoretical analysis. The KL divergence is another distance metric with known analytical form for the transition kernel (see Appendix A) and with a direct connection to the $\ell_1$ norm. As shown in Lemma~24, general uncertainty sets and $\ell_p$ uncertainty set can be related to the $\ell_1$ norm, which appears naturally in the cumulative regret computation, and in this sense our main theorem is flexible to different divergences.
>
> **Q5.** *Error propagation between policy and transition updates in the sample based regime.*
>
> With regard to the main theorem (Theorem 2 in the updated paper), we note the following parts of our analysis. First, part a) and b) only make statements about the regret with respect to the actual transition kernel at macro-iteration $k$. After this, part c) and part d) obtain the final regret with respect to the robust objective, using Lemma 8 to account for errors due to applying the robust objective. The telescoping sum of Liu et al. cannot be obtain directly for part a) and part b) since the transition dynamics change after the policy update. Therefore, additional corrections to the summation are applied using Lemmas 24 and 25.

---

### Review · Reviewer_brpZ · 2025-08-23

**Summary Of Contributions:**

This paper addresses the problem of safe and robust reinforcement learning by extending mirror descent policy optimisation (MDPO) to the robust constrained Markov decision process (RCMDP) framework. The paper proposes a new algorithm, MDPO-Robust-Lagrangian, which simultaneously optimises the policy (maximiser) and the adversarial transition kernel (minimiser) over the Lagrangian. They provide theoretical convergence guarantees both in oracle-based and sample-based settings. Empirical evaluations show consistent improvements in robustness and constraint satisfaction compared to baselines.

**Audience:**

Yes

**Broader Impact Concerns:**

This paper focuses on RL theory, and I do not see ethical concerns.

**Claims And Evidence:**

Yes

**Requested Changes:**

Please see the last section.

**Strengths And Weaknesses:**

Strengths:
1. The paper tackles safety and robustness in RL, two crucial aspects for real-world deployment, especially in continuous control and high-stakes applications.
2. The paper extends existing algorithms in purely constrained MDP or purely robust MDP to the RCMDP setting. It provides provable convergence guarantees for both oracle-based and sample-based RCMDPs and achieves competitive convergence rates.
3. Experiments confirm theoretical insights and demonstrate robustness improvements.

Weaknesses:
1. The paper focuses heavily on technical developments of mirror descent for RCMDPs, but does not provide a sufficiently clear high-level summary of the conceptual challenges in extending from CMDP and RMDP to RCMDP. For instance, what specific difficulties arise when simultaneously handling constraints and robustness, and how the proposed approach overcomes them, could be discussed more explicitly to highlight the novelty.
2. While the introduction cites relevant work, the paper lacks a dedicated related work section that systematically positions this work in the broader literature. In particular, although mirror descent is the main focus, the RCMDP setting has been studied with other algorithmic approaches, and these should be discussed in more detail to give readers a complete picture of existing results and limitations.
3. The current textual summaries and tables do not clearly convey the landscape of results. A more intuitive presentation would be to provide a single structured table comparing algorithms across RMDP, CMDP, and RCMDP settings, summarising key aspects such as: algorithm type, convergence point, convergence rate, and assumptions. Using concise keywords rather than long text would make the comparisons more direct and reader-friendly.
4. Presentation: Some technical details (e.g., long proofs) could be moved to the appendix, leaving the main text to emphasise conclusions and high-level proof sketches for accessibility. The formatting of algorithms and figures (e.g., on pages 6, 18, 21) is uneven, with some blocks occupying an entire line unnecessarily; these should be compacted for better readability and layout.

---

> ### Author Response · Authors · 2025-10-04
>
> The reviewer highlights the importance of safety and robustness in RL as crucial for real-world deployment and particularly in continuous control and high-stakes applications, and notes the theoretical guarantees and high empirical performance in robustness and constraint satisfaction. We thank the reviewer for recognising the importance of the problem setting and our results. The reviewer also provides several suggested changes for improving the paper, especially in the presentation of the paper, which we have agreed to implement.
>
> **Q1.** *The paper focuses heavily on technical developments of mirror descent for RCMDPs, but does not provide a sufficiently clear high-level summary of the conceptual challenges in extending from CMDP and RMDP to RCMDP. The paper lacks a dedicated related work section.*
>
> The authors have updated the paper to provide a broader overview the challenges and related work, particularly in the introduction and a newly dedicated related work section. The introduction more clearly summarises the challenges, design choices, and technical novelty of our solution. The newly added related work section (Section 2 in the updated paper) discusses challenges such as scalable solutions to and parametrisations of the inner problem, the design of objectives in RCMDPs, the lack of (or otherwise poor) convergence guarantees, sample-based approximate policy gradient techniques, and a discussion of MDPO as a practical algorithm compared to related policy optimisers.
>
> **Q2.** *The current textual summaries and tables do not clearly convey the landscape of results. A more intuitive presentation would be to provide a single structured table comparing algorithms across RMDP, CMDP, and RCMDP settings, summarising key aspects such as: algorithm type, convergence point, convergence rate, and assumptions. Using concise keywords rather than long text would make the comparisons more direct and reader-friendly.*
>
> The authors have changed the introduction and added a related work section, where the rates, assumptions, and challenges are discussed. The table has been merged into a single, reader-friendly table with concise keywords.
>
> **Q3.** *Presentation: Some technical details (e.g., long proofs) could be moved to the appendix, leaving the main text to emphasise conclusions and high-level proof sketches for accessibility.*
>
> The authors have now moved extensive proofs in the appendix, and have instead added brief proof sketches in the main text.
>
> **Q4.** *The formatting of algorithms and figures (e.g., on pages 6, 18, 21). These should be compacted for better readability and layout.*
>
> The authors have made the formatting of the figures uniform. The algorithm formatting has also been compacted by using Approximate TMA as a subroutine with its own algorithm box and by removing where possible too wordy explanations. As the algorithms possibly detract from the flow of the text, they have been deferred to the appendix.

---

### Review · Reviewer_Yek2 · 2025-10-01

**Summary Of Contributions:**

This paper proposes a policy-gradient framework for (RCMDPs) with formal convergence guarantees. It extends the primal-mirror-descent approach of Liu et al. (2021) and Wang et al. (2023) to jointly optimise the policy as a maximizer and the transition kernel as an adversarial minimiser over the Lagrangian of a constrained MDP. Theoretical results show an **$\mathcal{O}(1/T)$** convergence rate for the squared Bregman distance and **$\mathcal{O}(e^{-T})$** for entropy-regularised objectives in the oracle-based setting, and an **$\mathcal{O}(1/T^{1/3})$** rate in the sample-based RCMDP. For practical use (continuous state action space), the authors introduce MDPO-Robust-Lagrangian, a robust extension of MDPO [Tomaretal., et al., 2022] that incorporates a robust Lagrangian.

**Audience:**

Yes

**Claims And Evidence:**

Yes

**Requested Changes:**

See the weakness section

**Strengths And Weaknesses:**

**Strengths**

1. The paper studies a safe RL problem, which is relatively novel and it is relevant to practical applications.

2. The paper establishes state-of-the-art global convergence rates for RCMDPs in both the oracle-based and sample-based settings.

3. It covers both discrete and continuous action spaces, which enhances the generality of the proposed approach and makes the algorithm more practical for implementation across diverse environments.

4. The empirical evaluation spans three real-world applications offering a relatively comprehensive assessment that supports the effectiveness and robustness of the proposed method.

**Weaknesses**

1. The core methodology appears to be mainly a combination and limited extension of existing techniques from Liu et al. (2021), Tomar et al. (2022), and Wang & Petrik (2024). The paper does not convincingly demonstrate strong technical novelty beyond these prior works.  Although TMLR does not strictly require strong technical novelty, the reviewer would still encourage the authors to include more discussion on the paper’s technical novelty to better highlight its unique contributions compared to existing work. (My main concern)

2. The paper’s use of abbreviations is inconsistent. For example, *Robust Constrained Markov Decision Processes* is sometimes written as **RCMDP**, sometimes as **Robust Constrained MDP**. Besides, many abbreviations are redefined multiple times in different sections of the paper

3. The paper suffers from missing hyperlinks for lemmas, theorems, equations, and references.

4. The paper lacks a dedicated Related Work section, and the discussion of prior studies is somewhat unclear and inconsistent. For example, Table 1 briefly mentions Wang & Petrik (2024), but the introduction does not; conversely, Wang et al. (2023) is discussed in the introduction but omitted from the table. I recommend adding a clear Related Work section and improving Table 1 by restructuring it with articles as rows and key elements (e.g., type of MDP, algorithmic design technique, convergence rate, etc.) as columns, which would improve readability and comparison across methods.

5. The Preliminaries section does not sufficiently define the paper’s key learning objectives and concepts. Terms such as regret are used without formal definitions, which may confuse non-expert readers.

---

> ### Author Response · Authors · 2025-10-04
>
> As our main strengths, the reviewer highlights the novelty and relevance of our safe RL setting, the state-of-the-art convergence rates, the generalisation in action spaces, and the comprehensiveness of the empirical evaluation. The authors greatly appreciate these compliments and are pleased of the positive reception of the paper. The reviewer also mentions a few weaknesses which have been addressed as discussed below.
>
> **Q1**. *The paper does not convincingly demonstrate strong technical novelty beyond Liu et al. (2021), Tomar et al. (2022), and Wang & Petrik (2024) and the reviewer encourages more discussion on the paper’s technical novelty.*
>
> The updated paper now includes additional proofs and further highlights the key novelty in the introduction, which we summarise here. From a theoretical perspective, we emphasise the following key technical novelties:
> - **Bounding the error due to sub-optimal dual variables:** we provide an upper bound on the robust Lagrangian regret (Lemma 8 in the updated paper) which in addition to the two terms considered by Wang et al. (2024) also considers the suboptimality of the dual variables. This additional error is then shown to vanish in an average regret sense due to the constraint-costs averaging to $\mathcal{O}(\epsilon)$ across iterations (Lemma~29 in the updated paper).
> - **Bounding errors due to changing transition kernels:** a challenge in applying the telescoping analysis of Liu et al. to RCMDPs is that some of the terms no longer cancel since the transition kernel changes after the policy update. To enable such a telescoping analysis, we bound the error introduced by changes in the transition kernel in Lemma 24 and 25.
> - **Approximate TMA:** the TMA algorithm has been extended to the sample-based setting, introducing the Approximate TMA algorithm, which comes with an $\tilde{\mathcal{O}}(1/\epsilon^2)$ sample complexity.
> - **Continuous pseudo-KL divergence of occupancy:** we present a pseudo-KL divergence for continuous state-action spaces and show it is a Bregman divergence, making PMD-PD techniques applicable to continuous state-action spaces.
> - **Non-rectangular RCMDPs:** while Wang et al. provides an analysis of TMA only for $s$-rectangular RMDPs and derives oracle-based convergence rates for the policy optimiser, our work applies the Appproximate TMA algorithm to non-rectangular RCMDPs and then derives a full sample-based analysis of the full algorithm optimising all three parameters $(\pi,p,\lambda)$.
>
> From an experimental perspective, we highlight the following novelties:
> - **Comprehensive ablation study:** our study compares MDPO, PPO, and MCPMD algorithms for MDPs, CMDPs, RMDPs, and RCMDPs across three problem domains.
> - **Scalable implementation:** the practical algorithm is highly scalable, making use of function approximators and other tools of deep RL, as opposed to the pure Monte Carlo considered in \cite{Wangd}.
> - **Learning rate schedules:** we also present an empirical comparison of learning rate schedules, including traditional schedules (fixed, linear, and geometric) as well as a new restart schedule. We show that for RCMDPs, the best performance is obtained by the MDPO-Robust-Augmented-Lag variant under batch-based restart schedules.
>
> **Q2.** *Consistent use of abbreviations.*
>
> The text has been corrected to remove inconsistencies in the abbreviations.
>
> **Q3.** *The paper lacks a dedicated Related Work section, and the discussion of prior studies is somewhat unclear and inconsistent. For example, Table 1 briefly mentions Wang & Petrik (2024), but the introduction does not; conversely, Wang et al. (2023) is discussed in the introduction but omitted from the table.*
>
> The updated paper has now a dedicated related work section and the 2023 reference has been replaced by the intended 2024 reference.
>
> **Q4.** *The paper suffers from missing hyperlinks for lemmas, theorems, equations, and references.*
>
> The issue with missing hyperlinks has been resolved.
>
> **Q5.** *I recommend adding a clear Related Work section and improving Table 1 by restructuring it with articles as rows and key elements (e.g., type of MDP, algorithmic design technique, convergence rate, etc.) as columns, which would improve readability and comparison across methods.*
>
> The updated paper includes a related work section discussing the design of RCMDP objectives, the design of adversarial transition kernels, and policy optimisation in the context of CMDPs, RMDPs, and RCMDPs. The table has been made according to the mentioned template also following the recommendations of Reviewer brpZ.
>
> **Q6.** *The Preliminaries section does not sufficiently define the paper’s key learning objectives and concepts. Terms such as regret are used without formal definitions.*
>
> We have improved the preliminaries section to clarify the key objectives and concepts. This includes the various terms of regret, cumulative, and average regret, and clarifying how these relate to the aims of the paper.

---

> > ### Comment · Reviewer_Yek2 · 2025-11-05
> > **Thank you for your response**
> >
> > Thank you for your detailed response. I have carefully read the revised version of the paper, and I am pleased to see that all my previous concerns have been thoroughly addressed. I have no further comments.

---

### Author Response · Authors · 2025-10-04

The authors thank the reviewers for their positive comments as well as their constructive feedback on the limitations of the paper. During the review period, we made a few improvements in the writing. Apart from improving typos, inconsistencies, and grammar, we have also provided more context to the paper:
- We have expanded the introduction to better highlight the key challenges and how we address them.
- We have added a dedicated related work section (Section 2).
- We have provided additional context in the general assumptions (Section 5.1).
- Long proofs and algorithms have been put in the appendix to improve readability of the text. For proofs, we have also provided brief proof sketches as a replacement.

In addition to the writing, we have also introduced additional results to the paper:
- Since TMA previously used an oracle for the computation of its gradient, we have designed an algorithm for Approximate TMA (see Section 5.3 in the updated paper), which now allows a full sample-based analysis in our main theorem due to accounting for the samples required to compute the gradient. Since it comes with $\tilde{\mathcal{O}}(1/\epsilon^2)$ sample complexity, we maintain the $\tilde{\mathcal{O}}(1/\epsilon^3)$ convergence rate in the overall Robust Sample-based PMD-PD algorithm.
- In Section 5.5 of the updated paper, we provide an extension to non-rectangular uncertainty sets by comparing Approximate TMA to Conservative Policy Iteration over transition kernels. The convergence result implies an $\tilde{\mathcal{O}}(1/\epsilon^5)$ convergence rate to a sub-optimal solution, where the sub-optimality is determined by the degree of non-rectangularity.
- We have performed additional experiments to compare batch-based and epoch-based schedules for the penalty parameter, with results being especially favourable for batch-based MDPO-Robust-Augmented-Lag with restart schedule. These experiments are found in Appendix H, while the results presented in the Experiments section stay the same since they are based on the fixed schedule.

In addition to writing changes and additional results, the following technical aspects of the paper have also been revised:
- We have concluded that Theorem 4.5 of Wang et al. (2024) is incorrect since its use of Lemma B.5, which allows one to select the starting point and to reach any point at any iteration, does not allow one to make any inference about the regret of another process simply by virtue of passing through the same point. Consequently, since our oracle-based analysis relied on this reasoning, we have removed this part from our paper. Since the analysis was not so extensive and we consider the sample-based setting to be more realistic and important, we believe it is not such a great loss to the technical contribution of our paper.
- The derivations to account for the transition kernel changes in the telescoping sum of part a) in the main theorem previously used a term $\Delta_p \alpha \frac{\log(A)}{1-\gamma}$, which was not correct. The full derivation of the additional terms is now given in Lemmas 24 and 25.
- To evaluate a particular policy for the robust Lagrangian objective, it needs to be considered at the maximising settings of not only the transition kernel but also the dual variable. Therefore, we now include the sub-optimality of the dual variable as an additional term in the robust Lagrangian regret upper bound (Lemma 8 in the updated paper). It is shown to vanish in an average regret sense due to the averaged constraint-costs being $\mathcal{O}(\epsilon)$.

---

### Decision · Action_Editor_UW1h · 2025-11-15

**Recommendation:** Accept with minor revision

**Additional Comments:**

The authors are encouraged to embed the answers provided in the rebuttal in the final version.

**Audience:**

Yes

**Audience Explanation:**

The reviewers have raised some concerns about the scope of the paper, although this does not compromise the potential interest.

**Claims And Evidence:**

Yes

**Claims Explanation:**

The reviewers agree on the soundness of the paper and on the fact that the provided claims are accurate and evident.